# The Implicit Bias of Depth: From Neural Collapse to Softmax Codes

**Connall Garrod** [1]   **Jonathan P. Keating** [1]   **Christos Thrampoulidis** [2]

## Abstract

Neural collapse (NC) describes the structured geometry that emerges in the features and weights of trained classifiers. Recent theory suggests NC can be suboptimal in deep architectures, attributing this to an explicit low-rank bias from $L_2$ regularization. We study the deep unconstrained feature model (UFM)—equivalent to a deep linear network with orthogonal inputs—trained *without* regularization, to isolate how gradient descent and depth alone shape NC. We show that depth induces an implicit low-rank bias: low-rank matrices propagate norm more efficiently through successive multiplications, promoting low-rank alternatives to NC. These alternatives, we argue, correspond to softmax codes: max-margin solutions previously found in width-bottlenecked networks. Analyzing training dynamics under spectral initialization, we identify an early-time repulsion among singular values that drives low-rank emergence, and characterize how depth shrinks NC's basin of attraction. Finally, we show that some effects act in the opposite direction: for randomly initialized networks, increasing width biases training toward higher-rank solutions. Our results provide the first asymptotic and dynamic characterization of implicit bias in deep UFMs trained with unregularized multiclass cross-entropy.

## 1. Introduction

Deep neural-network classifiers, when trained beyond interpolation, converge to a highly structured geometric arrangement termed neural collapse (Papyan et al., 2020): class means form a simplex equiangular tight frame (ETF), and both features and classifier weights align with this frame.

Interestingly, this configuration arises without any explicit regularization or architectural constraint. *What biases inherent to gradient-descent dynamics or the architecture give rise to such geometric regularities?*

A substantial body of work approaches this question using the unconstrained features model (UFM) (Mixon et al., 2020; Fang et al., 2021), which treats final-layer features as free optimization variables jointly optimized with the classifier weights, abstracting away the preceding layers. The majority of these studies analyze the UFM with $L_2$ regularization on both features and weights, primarily under mean squared error (MSE) loss (Mixon et al., 2020; Han et al., 2022; Zhou et al., 2022a; Tirer & Bruna, 2022; Dang et al., 2023; Tirer et al., 2023), and less often under the more practically relevant cross-entropy (CE) loss (Zhu et al., 2021; Li et al., 2023; Zhao et al., 2024). In this framework, regularization becomes essential for NC to emerge. However, when the UFM is extended to multiple layers—yielding the *deep UFM*—NC and its generalization, deep neural collapse (DNC), are no longer globally optimal under $L_2$ regularization (Súkeník et al., 2024; Garrod & Keating, 2026): $L_2$ regularization induces a low-rank bias that suppresses NC in deep architectures.

Analyses based on $L_2$ regularization face key limitations. First, it is not clear whether the regularized model informs us about structures that emerge in unregularized training (Thrampoulidis et al., 2022). Second, in the deep UFM, despite NC being suboptimal at finite regularization, large regularization values are empirically needed to prevent it from emerging (Súkeník et al., 2024; Garrod & Keating, 2026). Third, and perhaps more concerning, the way $L_2$ regularization is applied in the UFM—directly on the features—differs markedly from standard training, where regularization acts on network weights.

These concerns are mitigated when examining UFMs without explicit regularization. Garrod et al. (2025) shows that the UFM trained with CE loss *without* explicit regularization still converges to NC, demonstrating that NC arises from the implicit bias of CE training alone. Yet, their analysis is restricted to a single hidden layer. It remains open how depth impacts the implicit bias: *How do the inductive biases induced by depth affect convergence to NC, both asymptotically and at finite training time? If depth favors alternatives*

[1]Mathematical Institute, University of Oxford [2]Department of Electrical and Computer Engineering, University of British Columbia. Correspondence to: Connall Garrod <connall.garrod@maths.ox.ac.uk>, Christos Thrampoulidis <cthrampo@ece.ubc.ca>.

*Proceedings of the 43rd International Conference on Machine Learning*, Seoul, South Korea. PMLR 306, 2026. Copyright 2026 by the author(s).

*to NC, what geometric structures replace it, and why is NC nonetheless pervasive in practical deep networks?*

## 1.1. Contributions

We provide the first comprehensive characterization of how implicit biases from depth, optimization dynamics, and initialization interact to determine whether NC emerges. Our setting is the deep UFM trained with CE loss *without* explicit regularization—a model where the loss minimum is achieved only at infinite parameter norm, and many geometric configurations attain it. The structure that emerges thus reflects purely the implicit biases of optimization and model depth, which we analyze from two complementary perspectives:

**Asymptotics: depth induces low-rank bias.** Asymptotically in time, gradient-flow (GF) dynamics reduce to a non-convex max-margin problem (Lyu & Li, 2019). By analyzing its landscape, we show that, unlike the shallow case, the deep landscape admits multiple stable local minima, among which NC is strictly suboptimal. We identify the mechanism behind this non-benign landscape: low-rank structures propagate norms more effectively through matrix multiplications, achieving larger logits for a fixed parameter norm. In the large-depth limit, we prove the global optimum corresponds to the $d = 2$ softmax code (Jiang et al., 2023), connecting depth-induced bias to explicit width constraints. Empirical evidence suggests this connection extends to higher-rank softmax codes at finite depths.

**Dynamics: depth reshapes the trajectory.** Building on the Hadamard framework of Garrod et al. (2025), we analyze GF dynamics at finite time. Unlike the shallow UFM, where NC is the unique stable direction throughout the trajectory, in the multilayer setting we prove this no longer holds: NC becomes unstable near the origin, with the unstable region growing with the number of layers, while alternative low-rank directions become stable. We show the mechanism is a "rich-get-richer" effect in which larger singular values grow faster, promoting low-rank structures before exiting the linearized regime. We further show that the KL divergence to NC, a Lyapunov function for the shallow UFM, can diverge under depth. Finally, we prove that random initialization induces concentration-of-measure effects that, for sufficiently large network widths, drive early dynamics toward NC. This provides a justification for why low-rank structures—though theoretically favored by depth—are not a certainty in practice.

We validate our results experimentally on deep UFMs and neural networks. Beyond NC, the deep UFM, which is equivalent to a linear network with orthogonal inputs, is the minimal model in which depth-induced implicit bias can be isolated from explicit regularization. Our analysis of this canonical setting thus informs the broader question of how depth shapes implicit bias in deep learning.

## 1.2. Related Work

Here, we review the most closely related works. See App. A for a full review and additional references.

Deep NC has been explored empirically by He & Su (2022); Parker et al. (2023); Rangamani et al. (2023), but theoretical efforts have been mainly based on deep UFMs trained with *MSE loss* (Tirer & Bruna, 2022; Dang et al., 2023; Súkeník et al., 2023; Garrod & Keating, 2024; Súkeník et al., 2024). Again under MSE loss, a parallel direction examines implicit regularization in more general linear networks, studying the interplay between GD and depth (Arora et al., 2018a;b; Bah et al., 2022; Yaras et al., 2023; Tu et al., 2024) and matrix factorization dynamics (Gunasekar et al., 2017; Arora et al., 2019; Li et al., 2020; Razin & Cohen, 2020). The more practically relevant CE setting is significantly less studied due to its added complexity. Only recently, Garrod & Keating (2026) showed that *explicit $L_2$* regularization induces a low-rank bias in the deep UFM. Ours is the first direct analysis of the deep UFM with CE loss and without explicit regularization, focusing instead on the implicit biases induced by GF dynamics.

Within the implicit-bias literature, the most closely related study is Ji & Telgarsky (2019), which analyzed GF dynamics in deep linear networks under binary logistic loss. While the deep UFM is a special case of a deep linear network with orthogonal inputs, our results are significantly more general as they: *(i)* directly apply to *multiclass* settings; *(ii)* are conclusive about GF convergence to *explicit geometric configurations* rather than implicit non-convex characterizations via KKT points (Lyu & Li, 2019); *(iii)* go beyond asymptotic convergence to study the *full dynamics*. GF dynamics for CE loss are significantly more challenging than for MSE: even in shallow UFM, it was only recently that Garrod et al. (2025) introduced a spectral-initialization-based framework to study these dynamics, extending prior work by Saxe et al. (2013) for MSE. We generalize this framework to deep models.

Prior work has shown that depth can bias optimization toward low-rank solutions, especially in matrix factorization (Gunasekar et al., 2017; Arora et al., 2019; Li et al., 2020; Chou et al., 2020), with related evidence in nonlinear and homogeneous networks (Huh et al., 2021; Jacot, 2023; Timor et al., 2023). Our work differs by focusing on a model of CE-trained *overparameterized* classifiers, showing that depth in this setting induces low-rank representations, reduces normalized margins, and drives a transition from neural collapse to softmax codes.

## 2. Background

**(Deep) UFM.** We study gradient flow (GF) on the deep unconstrained features model (UFM) with CE loss for $K$-class classification with $n$ samples per class. The model parameters are matrices $W_L \in \mathbb{R}^{K \times d}$, $W_{L-1}, \ldots, W_1 \in \mathbb{R}^{d \times d}$, and $H_1 \in \mathbb{R}^{d \times nK}$, where $d$ is the embedding dimension. The columns of $H_1$ are *trainable* feature vectors $h_{ic} \in \mathbb{R}^d$ for example $i$ of class $c$, ordered by class. The loss is

$$\mathcal{L}(Z) = -\sum_{c=1}^{K}\sum_{i=1}^{n}\log\left(\frac{\exp((z_{ic})_c)}{\sum_{c'=1}^{K}\exp((z_{ic})_{c'})}\right), \quad (1)$$

where $z_{ic} = W_L W_{L-1} \cdots W_1 h_{ic}$ are logit vectors forming the columns of the logit matrix $Z \in \mathbb{R}^{K \times nK}$. When $L = 1$ (single hidden layer), this reduces to the (shallow) UFM; our focus is on the deep case $L > 1$.

The deep UFM serves as an abstraction of overparameterized DNN classifiers, where the features $h_{ic}$ represent unconstrained embeddings trained jointly with the weights (Mixon et al., 2020; Fang et al., 2021; Han et al., 2022). This models a highly expressive feature map, for example a ResNet, with a linear head. We leave consideration of homogeneous activations in the head for future work. See App. B for more details and Garrod & Keating (2024, App. A) for justification of the UFM as a modeling tool of deep learning. That aside, since the deep UFM is equivalent to an $L$-layer linear network with orthogonal inputs (i.e., the simplest architecture exhibiting depth-dependent dynamics), it serves as the minimal setting for disentangling explicit regularization from implicit biases induced by optimization and depth. Linear networks have been studied extensively in deep-learning theory, but almost exclusively under MSE loss rather than the more practically relevant CE loss.

**(Deep) neural collapse.** DNC refers to geometric regularities observed empirically in the final layers of trained networks (Papyan et al., 2020; Parker et al., 2023). The key property is that the Gram matrix of centered class mean embeddings becomes proportional to the simplex ETF matrix $S = I_K - \frac{1}{K}1_K 1_K^T$. We defer a detailed definition of DNC to App. B; here we work with the following well-established equivalent characterization (Garrod & Keating, 2026). Let $\otimes$ denote the Kronecker product.

**Definition 2.1 (DNC in the deep UFM).** A DNC solution occurs when the normalized parameter matrices $\hat{W}_L, \ldots, \hat{W}_1, \hat{H}_1$ (where $\hat{X} = X/\|X\|_F$) satisfy the balancedness relations

$$\hat{W}_l^T \hat{W}_l = \hat{W}_{l-1}\hat{W}_{l-1}^T, \quad l = 1, \ldots, L, \quad (2)$$

with $\hat{W}_0 := \hat{H}_1$, and the normalized logit matrix is proportional to the simplex: $\hat{W}_L \cdots \hat{W}_1 \hat{H}_1 \propto S \otimes 1_n^T$.

Balancedness emerges asymptotically: the quantity $W_l^T W_l - W_{l-1}W_{l-1}^T$ is conserved under GF, and since parameter norms diverge, the normalized matrices satisfy (2) in the limit (Ji & Telgarsky, 2019).

For $L = 1$, Ji et al. (2022) showed that GF converges in direction to a KKT point of a non-convex max-margin problem (Eq. (3)), with NC as the unique global minimum and all other critical points being saddles. Garrod et al. (2025) further proved convergence to NC under spectral initialization, showing that the KL divergence from the parameters to the ETF structure decreases monotonically throughout training. For $L > 1$, both questions—whether the asymptotic landscape remains benign and how the dynamics evolve at finite time—have remained open.

## 3. Implicit Bias in the deep UFM

Here, we characterize the geometric structures that arise as limiting configurations of GF in the deep UFM. The CE objective admits no finite minimizers: the infimum of zero is achieved only at infinity, and by a wide variety of parameter configurations. This is the essence of implicit bias: although many directions asymptotically attain zero training loss, the optimization dynamics select structured ones from among them. We proceed in three steps: we first show that DNC is no longer the unique optimum once $L > 1$, then explain the mechanism behind this—a low-rank bias induced by depth in such models—and finally obtain an explicit characterization of the optimal low-rank structures in the large-depth limit.

### 3.1. Depth Breaks the Benign Landscape

*Does the benign landscape of the shallow UFM persist with depth?* To answer this we apply the framework of Lyu & Li (2019) to the deep UFM (Eq. 1). This shows that GF on CE loss implicitly performs (non-convex) margin maximization. Concretely, if there exists time $t_0$ such that the loss $\mathcal{L}(t_0) < \log(2)$, then any limit point of the normalized parameters $(\hat{W}_L, ..., \hat{W}_1, \hat{H}_1)$ lies in the direction of a Karush-Kuhn-Tucker (KKT) point of the constrained optimization problem

$$\min_{W_L, ..., W_1, H_1} \|H_1\|_F^2 + \sum_{l=1}^{L}\|W_l\|_F^2 \quad (3)$$

s.t. $(z_{ic})_c - (z_{ic})_{c'} \geq 1, \quad c' \neq c = 1, ..., K, \ i = 1, ..., n,$

where $z_{ic}$ are the columns in class order of the logit matrix $Z = W_L ... W_1 H_1$. Since this problem is non-convex, KKT points may be global minima, local minima, or saddles. Our first contribution is to characterize this landscape; the following theorem shows the benign structure of the shallow case shown by Ji et al. (2022) does *not* persist.

**Theorem 3.1.** *Consider the optimization problem in Eq. (3) with $d \geq K$. For $L = 2, K \geq 6$ or $L \geq 3, K \geq 4$, DNC is not globally optimal, although it remains a local optimum.*

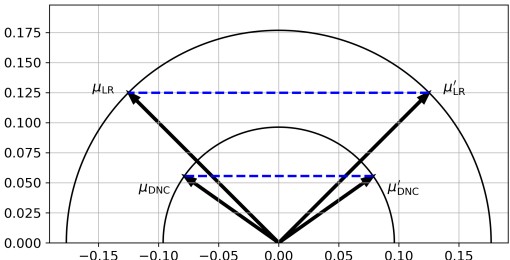

Figure 1. Illustration of low-rank bias for $K = 4, L = 3, n = 1$, with all parameter matrices normalized to unit Frobenius norm. $\mu_{\text{DNC}}, \mu'_{\text{DNC}}$ denote two feature embeddings of a logit matrix $Z$ under DNC, while $\mu_{\text{LR}}, \mu'_{\text{LR}}$ denote two feature embeddings of a logit matrix $Z$ under the low-rank geometry of Eq. (14). Although DNC yields the larger angle between features, the low-rank geometry lies on a large-radius hypersphere and therefore attains a larger Euclidean separation.

*Consequently, the landscape is no longer benign: multiple distinct locally optimal structures exist.*

The proof appears in App. D.1. This reveals a fundamental distinction: once $L > 1$, the landscape admits multiple stable configurations and GF may converge to structures other than DNC—bringing it qualitatively closer to practical deep-learning settings.

A qualitatively similar phenomenon, i.e., suboptimality of DNC at depth, was observed in regularized deep UFMs (Súkeník et al., 2024; Garrod & Keating, 2026), leading to interpretations that $L_2$ regularization is the cause. From a technical standpoint, the two settings differ fundamentally: regularized analyses study finite global minimizers, whereas Theorem 3.1 characterizes the asymptotic hard-margin regime. From a conceptual standpoint, our analysis shows the interpretation of previous work is incomplete: Theorem 3.1 establishes that depth alone, without any explicit regularization, suffices to induce the suboptimality of DNC. The next subsection explains why.

### 3.2. The Mechanism: Low-Rank Structures Propagate Norm More Efficiently

The mechanism by which depth alters the loss surface in such models is a low-rank bias: low-rank matrices propagate norm more effectively through a sequence of matrix multiplications. To make this precise, consider the logit matrix $Z$ formed as a product of parameter matrices all having the same Frobenius norm, which is true under the balancedness condition (Eq. (2)). Writing $W_0 := H_1$, we decompose

$$Z = \Big( \prod_{l=0}^{L} \|W_l\|_F \Big) \cdot \tilde{Z}, \text{ where } \tilde{Z} = \hat{W}_L \cdots \hat{W}_1 \hat{W}_0 . \quad (4)$$

Once positive margins are achieved, the loss decreases monotonically as the norms grow, and each $\|W_l\|_F$ in-

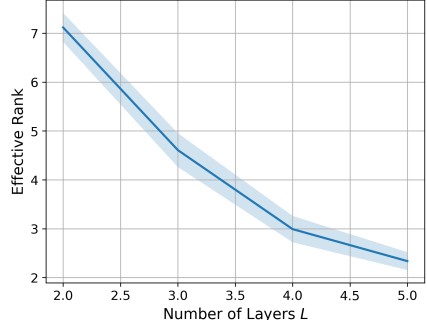

Figure 2. Effective rank (Eq. (60), App. E) of the logit matrix at GF convergence for $K = 10$ classes and varying network depth $L$. For each value of $L$ we trained deep UFM for five random initializations and computed the effective rank after training. Mean $\pm$ one standard deviation is shown. Rank decreases with depth.

creases monotonically under gradient flow (Lyu & Li, 2019). Therefore, differences between asymptotic structures are governed by $\tilde{Z}$. Crucially, it is clear from Eq. (4) that $\|\tilde{Z}\|_F$ contributes to the overall scale of $Z$, so maximizing $\|\tilde{Z}\|_F$ is advantageous for reducing the loss by widening the margins. We thus compare $\|\tilde{Z}\|_F$ under DNC versus low-rank alternatives.

Under DNC, and taking $n = 1$ for simplicity, using Definition 2.1 and a small calculation shows $\tilde{Z}_{\text{DNC}} = \hat{S}^{L+1}$, yielding $\|\tilde{Z}_{\text{DNC}}\|_F = (K - 1)^{-L/2}$. By contrast, consider solutions with positive margins whose normalized logits have equal singular values but rank $r < K - 1$ [1]. For such low-rank structures, we can compute that $\|\tilde{Z}_{\text{LR}}\|_F = r^{-\frac{L}{2}}$, and hence

$$\frac{\|\tilde{Z}_{\text{DNC}}\|_F}{\|\tilde{Z}_{\text{LR}}\|_F} = \exp\left( -\frac{L}{2} \log\left( \frac{K-1}{r} \right) \right) .$$

The logit norm achievable under DNC is thus exponentially suppressed in depth relative to lower-rank alternatives. This occurs because of the basic linear-algebra effect that low-rank matrices "propagate norm" more efficiently through repeated matrix multiplication. This is since fewer singular vector directions means larger individual singular values for a given Frobenius norm, causing less decay under matrix powers, with the effect strengthening with increasing $L$.

**Angular versus Euclidean separation.** For fixed $\|Z\|_F$, DNC is the maximum-margin solution under direct logit optimization (Thrampoulidis et al., 2022; Garrod et al., 2025), and is known to maximize the angular separation between class means (Papyan et al., 2020)—a property closely related to margin maximization (Jiang et al., 2023). However, depth distinguishes angular separation from Euclidean separation. Low-rank structures can attain larger logit norms, and thus larger Euclidean margins, simply by lying on a

---

[1]Concrete examples with $r = K/2$ are given in App. D.1, and $r = 2$ in Theorem 3.2.

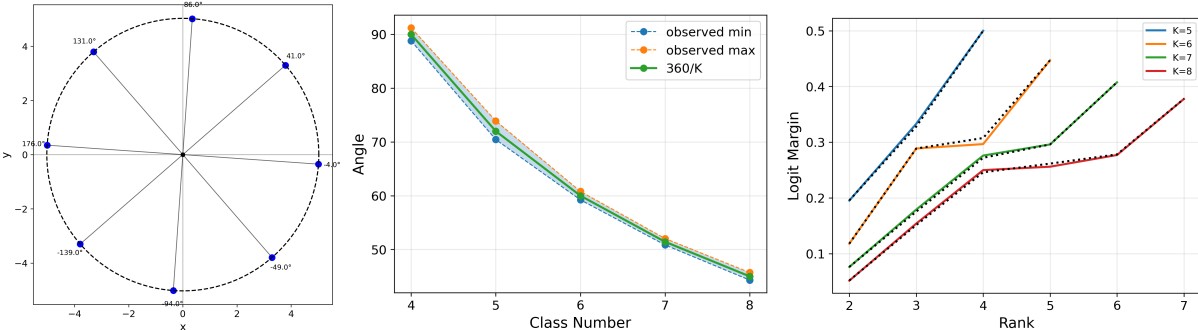

*Figure 3.* Optimal low-rank structures correspond to softmax codes. **Left:** Columns of the gram factor $X$ defined in Eq. (5) for one run with $L = 10$ that converged to rank 2. **Middle:** For each $K$, we trained five deep UFMs that converged to rank 2; shown are the largest and smallest adjacent angle gaps of the columns $x_i$ of the Gram factor $X$ over all five runs. The angle $360/K$ corresponds to the $d = 2$ softmax code. **Right:** Normalized margins for low-rank solutions versus architecturally bottlenecked solutions. Dotted lines: $L = 1$ UFM with width $d = r$ which is known to yield softmax codes (Jiang et al., 2023). Solid lines: networks with $L = 4$ and $d \geq K$ that converged to rank $r$, for varying $K$.

larger hypersphere (even though their angular separation is inferior). Figure 1 illustrates this geometry: DNC maximizes the angle between logit means, but the low-rank configuration achieves greater Euclidean separation by residing on a hypersphere of larger radius. Figure 2 provides empirical confirmation: as depth increases, the effective rank of the converged logit matrix decreases systematically.

**Discrimination versus confidence.** Low-rank solutions can achieve lower training loss than DNC despite being worse normalized classifiers. Since diverging logit norms drive softmax outputs toward one-hot vectors, larger $\|Z\|_F$ contributes only by increasing prediction confidence. Maximal angular separation, by contrast, accounts for the discrimination between classes. Low-rank solutions thus trade discrimination quality (encoded by $\hat{Z}$, which determines the predicted label) for increased confidence (encoded by $\|Z\|_F$). This suggests they may overfit more and generalize worse than DNC, connecting their rarity in standard training (Papyan et al., 2020) to the broader question of why deep networks generalize despite sufficient capacity to overfit (Zhang et al., 2017).

### 3.3. Characterizing the Optimal Low-Rank Structures

Having established that low-rank structures can outperform DNC and explained the mechanism, we now ask a more challenging question: *what are these optimal structures?* Since explicit characterizations of non-convex landscapes are difficult to obtain, we focus on a large-depth limit.

**Theorem 3.2.** *Consider the optimization problem in Eq. (3), with class number $K > 2$ and network width $d \geq K$. In the limit $L \to \infty$, the globally optimal solutions among the set of positive semi-definite logit matrices satisfying NC1 (meaning $Z$ can be written as $Z = \bar{Z} \otimes 1_n^T$) and equal norm*

*features is of the form*

$$Z = X^T X \otimes 1_n^T, \quad X \in \mathbb{R}^{2 \times K}, \quad (5)$$

*where the columns $x_i \in \mathbb{R}^2$ of $X$ are uniformly spaced on a circle, meaning for some $\alpha, \mu \in \mathbb{R}$, the set $\{x_i : i = 1, ..., K\}$ is equal to $\left\{ \left( \mu \cos \left( \frac{2\pi i}{K} + \alpha \right), \mu \sin \left( \frac{2\pi i}{K} + \alpha \right) \right) : i \in [K] \right\}$.*

The proof appears in App. D.2. Figure 3 confirms the prediction: for large $L$, when GD converges to rank two solutions they always correspond to the theorem's structure: class means arranged as vertices of a regular $K$-gon.

**Connection to softmax codes.** The structure identified in Theorem 3.2 coincides with the optimal solution of the single-layer UFM under an extreme width bottleneck $d = 2$ (Jiang et al., 2023). More generally, Jiang et al. (2023) show that when $d < K - 1$, NC is geometrically infeasible and alternative structures—which they term *softmax codes*—must arise. Theorem 3.2 demonstrates that large depths select precisely the $d = 2$ softmax code, even when no width constraint is present. Figure 3 (right) suggests this connection extends beyond the $L \to \infty$ limit: the normalized margin achieved by GF on a depth-$L$ network that converges to a rank-$r$ logit matrix closely matches that of a single-layer network with width $r$, suggesting these alternative optima are softmax codes more generally. This confirms that depth acts as an implicit rank constraint, and intriguingly suggests it selects solutions that would otherwise require explicit width bottlenecks. Extending Theorem 3.2 to finite $L$ is an exciting direction for future work.

## 4. A Dynamical Analysis of Implicit Bias

The previous section characterized implicit bias asymptotically: we identified the KKT points to which GF converges as $t \to \infty$ and studied the landscape of the correspond-

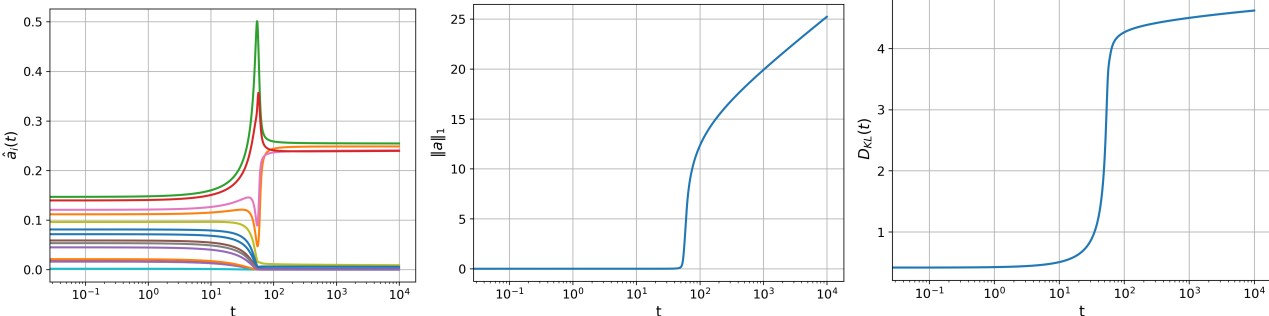

*Figure 4.* Experiments under Hadamard initialization. **Left:** Evolution of the normalized logit singular values $\hat{a}_i$ under Eq. (6) for depth $L = 2$ and $K = 16$ classes; singular values initialized with uniform entries in $[0, 1]$, then rescaled to have $L_1$ norm $10^{-3}$. Note convergence to low-rank solution where many modes approach zero. **Middle:** Corresponding evolution of the logit norm $\|a\|_1 = \sum_i a_i$. By the time norm increases and system exits the linearized regime, the low-rank structure has already taken hold. **Right:** The corresponding KL divergence (definition in Eq. (52).).

ing max-margin problem. We now take a more ambitious step: understanding the dynamics that GF traverses *before* reaching the asymptotic regime. This is particularly important because convergence to KKT points of Eq. (3) only becomes accurate at exponentially large times—the convergence rate is $O(1/\log t)$ (Lyu & Li, 2019). Several natural questions arise: *When does DNC become a stable direction along the trajectory? Are there competing stable directions at finite times? What happens near initialization?*

For $L = 1$, Garrod et al. (2025) recently provided complete answers under spectral initialization: NC is the unique stable direction throughout the entire trajectory, and the KL divergence to NC decreases monotonically for any positive initialization. In other words, the dynamics not only converge to NC but approach it monotonically (in KL distance) at all times.

For $L > 1$, we prove here that this picture changes entirely.

### 4.1. The Hadamard Framework

To obtain these results, we employ Hadamard initialization (Garrod et al., 2025), which provides a tractable framework for studying GF dynamics in CE models. Under this initialization, the singular vectors of all parameter matrices remain fixed throughout training; evolution occurs only in the singular values. The following theorem shows this remains true for arbitrary depth. Although this is a special initialization, it suffices for our purposes: it reveals structure in the loss surface, and for instability questions it is enough to exhibit one trajectory along which a given structure is unstable. Moreover, convergence to non-DNC solutions under Hadamard initialization implies such alternatives exist in the full landscape.

We further provide empirical evidence in App. E.4 that both Hadamard initialization and small random display the same qualitative phenomenology. This is a common observation

in the spectral dynamics literature, where structured analytic trajectories serve as indicative paths of a much wider set of initializations (Saxe et al., 2013; 2019; Gidel et al., 2019; Garrod et al., 2025).

**Definition 4.1** (Sylvester Hadamard Matrix). The Sylvester Hadamard matrices $\{\Phi_{2^m} : m \in \mathbb{N}\}$ are defined recursively by

$$\Phi_1 = 1, \quad \Phi_{2^m} = \begin{bmatrix} \Phi_{2^{m-1}} & \Phi_{2^{m-1}} \\ \Phi_{2^{m-1}} & -\Phi_{2^{m-1}} \end{bmatrix} \in \mathbb{R}^{2^m \times 2^m}.$$

**Theorem 4.2.** *Let $K = 2^m$ for $m \in \mathbb{N}$, and let $U = \frac{1}{\sqrt{K}}\Phi$, where $\Phi$ is the $K \times K$ Sylvester Hadamard matrix. Consider the deep UFM in Eq. (1), with balanced classes of size $n$. Initialize the parameter matrices as $W_L = UD_LR_L^T$, $W_l = R_{l+1}D_lR_l^T$ for $l = 1, ..., L-1$, and $H_1 = R_1D_0V^T$, where $R_L, ..., R_1 \in \mathbb{R}^{d \times d}$ are orthogonal, and $V = U \otimes Q$ with $Q$ a right singular matrix of $1_n^T$. Assume the only non-zero singular values of $D_l$ are $\alpha_0^{(l)}, ..., \alpha_{K-1}^{(l)}$ for $l = 0, ..., L+1$. Then under balancedness, and absorbing constants into the time and singular value variables, the gradient flow equations reduce to*

$$\frac{da_i}{dt} = \frac{1}{D}b_i a_i^{\frac{2L}{L+1}}, \quad i = 1, ..., K-1, \quad (6)$$

*where $a_i = \prod_{l=0}^{L} \alpha_i^{(l)}$ are the logit singular values, $b_i = \sum_{j=1}^{K-1} \Psi_{ij}e^{-(\Psi a)_j}$, $D = 1 + \sum_{j=1}^{K-1} e^{-(\Psi a)_j}$, and $\Psi \in \mathbb{R}^{K-1 \times K-1}$ is the core of the matrix $1_K 1_K^T - \Phi$, constructed by deleting its first column and row.*

The proof appears in App. D.3. As in the $L = 1$ case (Garrod et al., 2025), the first singular value $a_0$ of the logit matrix does not influence the evolution of the remaining singular values nor evolves itself and can be set to zero; we henceforth drop it from our notation, so that $a \in \mathbb{R}^{K-1}$.

We are ultimately interested in the normalized logit matrix (the direction). Thus, normalizing with respect to the $L_1$

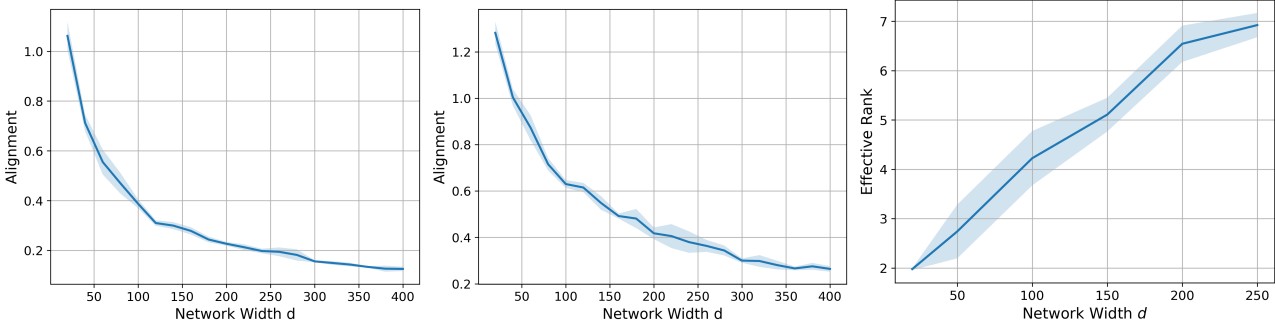

*Figure 5.* Impact of network width on network training. Depicted are the mean and one-standard-deviation error bars over five runs. **Left:** Frobenius distance metric $M$ (see (61) in App.) between the logit-matrix derivative $\frac{dZ}{dt}(0)$ and the simplex ETF. **Middle:** Same metric for the logits after training has induced the logit norm to increase by a scale factor. **Right:** Effective rank of the logit matrix after training.

norm yields dynamics for $\hat{a}_i = a_i/\|a\|_1$:

$$\frac{d\hat{a}_i}{dt} = \frac{1}{D\|a\|_1}\left[b_i a_i^{\frac{2L}{L+1}} - \hat{a}_i \sum_j b_j a_j^{\frac{2L}{L+1}}\right].$$

In this context, DNC corresponds to $\hat{a} \propto 1_{K-1}$, since $\Phi\text{diag}(0,1,...,1)\Phi^T = S$, where we recall that $a_0 = 0$. Moreover, we say a direction $\hat{a}^*$ is *stable* if the Euclidean distance from $\hat{a}$ to $\hat{a}^*$ decreases when sufficiently small.

### 4.2. Instability of Neural Collapse

For $L = 1$, Garrod et al. (2025) proved that the KL divergence $D_{\text{KL}}(\frac{1}{K-1}1_{K-1} \| \hat{a})$ decreases monotonically, serving as a Lyapunov function that forces all strictly positive initializations to converge to NC. Since this KL distance diverges at any low-rank solution, the dynamics must consistently move away from such solutions. They also showed NC is locally stable throughout the trajectory. We prove (see App. D.4) that both properties fail for $L > 1$:

**Theorem 4.3.** *Let $K = 2^m$ for $m \in \mathbb{N}$ with $m > 1$. Let $a(t) \in \mathbb{R}^{K-1}$ evolve according to Eq. (6) with initialization $a_i(0) > 0$ for all $i$, and let $L \in \mathbb{N}$. Then:*

*(i) The direction $\hat{a} = \frac{1}{K-1}1_{K-1}$ is stable when $\|a\|_1 > \frac{L-1}{L+1}(K-1)$, but unstable below this threshold.*
*(ii) When $L > 1$, there exists alternative low-rank structures that are also stable above a threshold in $\|a\|_1$.*
*(iii) For $L > 1$, the KL-divergence $D_{\text{KL}}(\frac{1}{K-1}1_{K-1} \| \hat{a})$ is no longer monotonically decreasing, and there exists gradient-flow trajectories along which it diverges.*

For $L = 1$ the NC direction is stable for all $\|a\|_1$, but for $L > 1$ it is unstable near the origin, with the unstable region growing monotonically in $L$. While DNC becomes stable at larger scales, other stable directions, corresponding to low-rank structures, coexist. Thus, in the context of the model, depth induces a low-rank bias not only in the asymptotic landscape (Section 3) but also along finite-time trajectories. Crucially, for $L > 1$ the KL divergence is not only non-monotone but can diverge along certain GF

paths. This occurs when trajectories converge towards low-rank solutions, confirming that DNC is not guaranteed as an outcome of training.

### 4.3. The "Rich-get-Richer" in the Linearized Regime

To understand *why* DNC is unstable near the origin for $L > 1$, we analyze the small-norm regime $\|a\|_1 \ll 1$.

**Theorem 4.4.** *Let $K = 2^m$ for $m \in \mathbb{N}$. Let $a(t) \in \mathbb{R}^{K-1}$ evolve according to Eq. (6) with initialization $a_i(0) > 0$ for all $i = 1, ..., K-1$, and let $L \in \mathbb{N}$. To linear order in the scale $\|a\|_1$, the dynamics reduce to:*

$$\frac{da_i}{dt} = a_i^{\frac{2L}{L+1}}(1 - a_i) + O\left(\|a\|_1^{2+\frac{2L}{L+1}}\right).$$

*As a consequence, when $\|a\|_1$ is small:*

- *If $L = 1$, then $\frac{d}{dt}\left(\frac{\hat{a}_i}{\hat{a}_j}\right)$ has the opposite sign to $\hat{a}_i - \hat{a}_j$. Consequently, unequal modes are driven toward each other, and DNC is the only stable direction under the linearized dynamics.*
- *If $L > 1$, then $\frac{d}{dt}\left(\frac{\hat{a}_i}{\hat{a}_j}\right)$ has the same sign as $\hat{a}_i - \hat{a}_j$. Consequently, DNC is a critical direction, but is not stable under the linearized dynamics.*

The proof appears in App. D.5. The theorem identifies the mechanism behind DNC instability: A "rich-get-richer" effect that emerges for $L > 1$. Near initialization, larger singular values grow relatively faster than smaller ones, amplifying initial disparities. This promotes the development of approximate low-rank structure in the logit matrix. By the time the dynamics exit the linearized regime, the system is more likely to lie in the basin of attraction of a low-rank direction than of DNC. These small-norm effects are particularly relevant for CE models due to the lazy dynamics that emerge as the norm grows (Garrod et al., 2025).

Figure 4 illustrates this mechanism. Initially, singular values evolve under this "rich-get-richer" effect, and the logit matrix becomes low-rank. Around $t \approx 5 \times 10^1$, the norm $\|a\|_1$ grows large enough that the linearized dynamics no longer

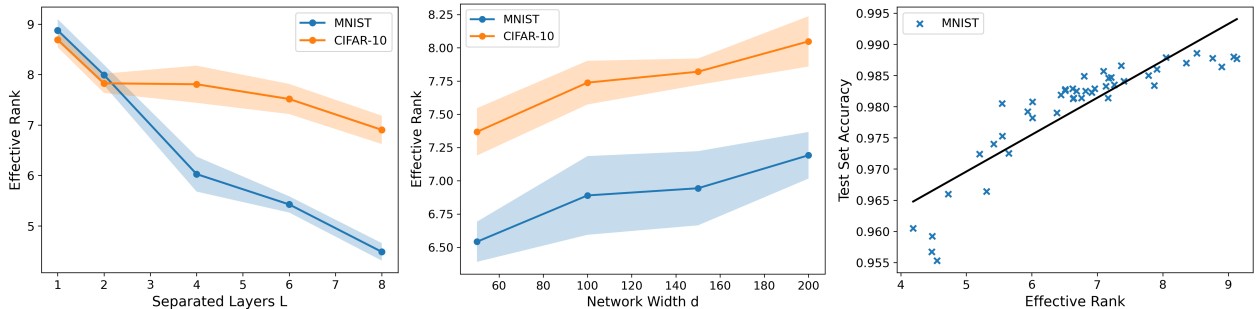

*Figure 6.* Confirmation of theory on the MNIST and CIFAR-10 datasets using the ResNet-20 architecture with a ReLU head. We report the average effective rank of the mean logit matrix at the end of training and one-standard-deviation error bars over five runs. **Left:** Here the ReLU head has width $d = 50$ and a variety of depths $L$. **Middle:** Here the ReLU head has depth $L = 3$ for a variety of widths. **Right:** Scatter plot of the generalization performance against the effective rank of the mean logit matrix across the forty five MNIST models of the previous two plots. We also show the line of best fit to show the general trend.

apply, but by then the low-rank structure has already taken hold, and the trajectory settles into a low-rank optimum.

### 4.4. Random Initialization and Large Width is Biased Towards Neural Collapse

The preceding sections established that depth alone—without $L_2$ regularization—suffices to make DNC suboptimal and to create stable low-rank alternatives. Yet empirically, DNC is observed ubiquitously in standard training, even with little or no regularization (Papyan et al., 2020; Parker et al., 2023). Here we highlight how random initialization combines with large network width actually encourages higher rank outcomes.

Garrod & Keating (2026) conjecture that DNC becomes more prevalent with increasing width, offering a heuristic explanation for the impact of width. However, they do not study optimization trajectories or whether DNC is dynamically reachable from typical initializations. Here we give a dynamical explanation: *standard random initialization, together with large width, biases early training toward DNC*, steering optimization away from regions of the loss surface where low-rank optima are attractive.

**Theorem 4.5.** *Consider the deep UFM described in Eq. (1) with parameter matrices initialized as $W_l = \epsilon B_l$ for $l = 1, ..., L$, and $H_1 = \epsilon B_0$, where each $B_l$ has entries sampled i.i.d. from a Gaussian distribution with mean $0$ and variance $\frac{1}{d}$. Then, in the joint limits $\epsilon \to 0$ and $d \to \infty$ we have the following convergence in probability*

$$\dot{Z}(0) \to \epsilon^{2L}(L+1)S \otimes 1_n^T.$$

The proof appears in App. D.6. The theorem shows that under random initialization and large network width, the initial velocity of the logits is biased toward alignment with the simplex ETF—the signature of DNC. This is due to a concentration of measure among the initial eigenvalues of the logit derivative. Although the result is stated for small $\epsilon$,

the result holds whenever the softmax outputs are approximately uniform at initialization, which does not require vanishingly small logit norms.

By continuity of GF, the velocity remains aligned with $S$ for a nontrivial interval after initialization. Since $Z(0)$ has small norm and balancedness is preserved (keeping layers approximately aligned), the singular values develop nearly equally, up to $\epsilon$- and $d$-suppressed corrections. Figure 5 confirms this: increasing width strengthens the alignment of the initial logit derivative with the simplex ETF, and after brief training the logits inherit this alignment. Correspondingly, the effective rank at convergence increases, indicating that initialization meaningfully shapes the final outcome.

Note that this initialization bias directly counteracts the rich-get-richer effect from Section 4.3. While that effect amplifies disparities among singular values, its strength depends on how non-uniform they are at initialization. Because DNC remains a critical point of the linearized dynamics (Theorem 4.4), nearly equal singular values drift only weakly. Thus, if initialization places the network sufficiently close to DNC, the linearized dynamics will not move it far before norm growth pushes the system out of this regime. The network then enters the large-norm regime with high effective rank, where DNC is locally stable (Theorem 4.3) and hence can converge to DNC. This gives a dynamical explanation for why width affects the empirical persistence of DNC under standard training, and highlights the role of concentration-of-measure at initialization.

## 5. Empirical Verification of Implicit Bias

We evaluate our theory on MNIST (Lecun et al., 1998) and CIFAR-10 (Krizhevsky, 2009). In each experiment, we train a ResNet-20 with a fully connected ReLU head of varying depth and width (see App. E for full details). The first panel of Figure 6 shows that the effective rank of the mean logit matrix at convergence decreases with head depth,

matching the deep UFM: depth induces low-rank structure. We also include the $L = 1$ case, where all effective ranks are approximately $K - 1$, which corresponds to NC. The low-rank bias effect is stronger on MNIST than on CIFAR-10, suggesting dataset complexity influences convergence behavior. The second panel shows that increasing width generally increases the effective rank on both datasets.

The final panel connects generalization with effective rank at convergence. In general, lower effective rank is associated with poorer generalization, consistent with the idea that lower-rank solutions achieve smaller margins and therefore generalize less well. That said, the relationship is not absolute: randomly initialized networks can exhibit high effective rank while still generalizing poorly, whereas NC solutions have low rank yet generalize best among the solutions we observe. The key conclusion is that generalization deteriorates when optimization favors low-rank structure at the expense of the margins the network can achieve.

Overall, these results suggest our theory extends beyond the modeling setting to more general deep classification networks.

## 6. Conclusion

We have shown that depth alone induces an implicit low-rank bias in deep UFMs trained with CE loss, producing alternative low-rank structures, which we partially characterize. Our analysis provides the first rigorous account of implicit bias in these models. This analysis opens several directions. Our infinite-depth theorem and empirics establish correspondence between low-rank optima and softmax codes; extending the theory to finite depth remains open. The low-rank alternatives also provide a natural control group for studying NC's role in generalization—unlike prior comparisons against less-trained models (Papyan et al., 2020), they offer a cleaner baseline (Figure 6 provides preliminary evidence). Finally, the absence of these alternatives in Papyan et al. (2020)'s experiments—most of which use residual connections, with VGG (closest to our setup) showing the weakest NC convergence—suggests skip connections may play a key role in NC's prevalence. Clarifying this mechanism is a promising direction for future work.

**Limitations:** Our work studies deep neural networks through a principled abstraction of the overparameterized limit. Although this model is empirically well supported in that regime (see App. A of Garrod & Keating (2024)), its applicability may weaken in underparameterized or highly constrained settings. Moreover, the UFM is primarily a model of optimization and therefore does not directly address generalization. We formulate and empirically validate hypotheses about generalization via the correlated notion of dataset margin, but a formal link would require additional

modeling. Our real-network experiments further indicate that dataset complexity affects the observed phenomena, and other practical factors—such as batch size, learning rate, and initialization scale—may also influence how the theory manifests in practice. Finally, Theorems 3.2 and 4.5 are asymptotic; obtaining rates or finite analogues is an important direction for future work.

Our finite-time training dynamics require Hadamard initialization, which is substantially more structured than random initialization and requires $K = 2^m$ for some $m \in \mathbb{N}$. We provide evidence in App. E.4 that the same phenomenology persists under small random initialization, even for general $K$. Since Hadamard-initialized paths are genuine trajectories of the loss surface, they remain informative about stability. Nevertheless, formally extending these arguments to random initialization is an important open direction.

## Acknowledgments

We thank the anonymous reviewers for their thoughtful feedback and constructive suggestions, which helped improve the clarity and presentation of the paper. CG is supported by the Charles Coulson Scholarship. The authors also acknowledge support from His Majesty's Government in the development of this research. CT acknowledges support by the NSERC Discovery Grant No. 2021-03677, the Alliance Grant ALLRP 581098-22 and a gift from Google. For the purpose of Open Access, the authors have applied a CC BY public copyright license to any Author Accepted Manuscript (AAM) version arising from this submission.

## Impact Statement

This paper presents work whose goal is to advance the field of machine learning, and more specifically, the impact of implicit biases on the representations of deep neural networks. There are many potential societal consequences of our work, none which we feel must be specifically highlighted here.

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

## A. Detailed Discussion on Related Work

**Neural collapse:** First introduced by Papyan et al. (2020), NC has evolved into a central object of study in the theory of deep classification models. Much of the subsequent analysis has been conducted within the unconstrained feature model (UFM) (Mixon et al., 2020; Fang et al., 2021) with explicit $L_2$ regularization, where a broad range of loss functions has been examined (Zhou et al., 2022b; Mixon et al., 2020; Ji et al., 2022; Han et al., 2022; Ma et al., 2025). Across these settings, NC consistently emerges as the globally optimal geometric configuration, with all other critical points being non-degenerate saddles (Zhou et al., 2022a; Zhu et al., 2021; Ji et al., 2022). Follow up research has since extended the original NC observation to imbalanced data (Thrampoulidis et al., 2022; Behnia et al., 2023; Fang et al., 2021; Dang et al., 2023; Hong & Ling, 2023) and to regimes with many classes (Jiang et al., 2023). Broader generalizations of the phenomenon have also been established in multilabel settings (Li et al., 2023), soft-label language models (Zhao et al., 2024), and multi-class regression (Andriopoulos et al., 2024). The relevance of the UFM for real neural networks has further been supported by Súkeník et al. (2025), who show its equivalence for deep architectures with skip connections.

Complementary perspectives on NC have emerged outside the UFM as well, including analyses in alternative theoretical frameworks (Jacot et al., 2025; Zangrando et al., 2024), as well as treatments via mean-field methods (Wu & Mondelli, 2025) and NTK-based approaches (Kothapalli & Tirer, 2024; Seleznova et al., 2023). Recent works additionally highlight the role of dataset structure in shaping NC behavior (Hong & Ling, 2024; Kothapalli & Tirer, 2024). Beyond theoretical significance, NC has been linked to a number of practical properties in deep learning, such as adversarial robustness (Su et al., 2023), transfer learning (Galanti et al., 2021; Li et al., 2022), generalization (Hui et al., 2022; Gao et al., 2023), and out-of-distribution detection (Wu et al., 2024; Zhang et al., 2024b). Its dependence on architecture has been examined in the context of ResNets (Li & Papyan, 2024; Súkeník et al., 2025; Wang et al., 2024) and large language models (Wu & Papyan, 2024; Zhao & Thrampoulidis, 2025a). An overview of developments surrounding NC can be found in the survey by Kothapalli et al. (2022).

DNC has also been explored empirically (Parker et al., 2023; Rangamani et al., 2023; He & Su, 2022). Efforts to theoretically characterize DNC have relied predominantly on deep UFMs trained with MSE loss, including two-layer constructions (Tirer & Bruna, 2022), linear layers (Dang et al., 2023; Garrod & Keating, 2024), binary classification (Súkeník et al., 2023) and more general settings (Súkeník et al., 2024). In the CE setting, Garrod & Keating (2026) show that $L_2$ regularization induces an explicit low-rank bias that influences the emergence of DNC. Additional perspectives outside the UFM—such as layer-wise training schemes—have also been explored (Beaglehole et al., 2024).

**Implicit bias:** A substantial literature investigates how gradient-based optimization shapes model structure in the absence of explicit regularization. Soudry et al. (Soudry et al., 2018) established that gradient descent converges in direction to the max-margin classifier for logistic regression. Researchers have also related the implicitly regularized path to explicitly regularized solutions in binary classification (Ji et al., 2020). Subsequent studies have broadened our understanding of implicit bias to homogeneous (Lyu & Li, 2019; Ji & Telgarsky, 2020; Vardi et al., 2022; Timor et al., 2023) and non-homogeneous networks (Cai et al., 2025), to alternative optimization algorithms (Gunasekar et al., 2018a; Azizan & Hassibi, 2018; Sun et al., 2022; Pesme et al., 2024; Zhang et al., 2024a; Fan et al., 2025), to large step-size regimes (Even et al., 2023; Wu et al., 2023), and to specific architectural families such as convolutional networks (Gunasekar et al., 2018b; Lawrence et al., 2021) and transformers (Tarzanagh et al., 2023; Vasudeva et al., 2024; Julistiono et al., 2024). A parallel direction examines implicit regularization in linear networks trained with MSE loss, studying the interplay between gradient descent and depth (Arora et al., 2018b; Yaras et al., 2023; Tu et al., 2024; Bah et al., 2022; Arora et al., 2018a), including classical matrix factorization dynamics (Li et al., 2020; Arora et al., 2019; Razin & Cohen, 2020; Gunasekar et al., 2017).

Implicit bias has additionally served as a direct theoretical tool for studying NC within the unregularized CE UFM. Ji et al. (Ji et al., 2022) established convergence to KKT points of a constrained norm-minimization formulation. Thrampoulidis et al. (2022) provided a characterization of the implicit solution path under class imbalance. Garrod et al. (2025) proved convergence to NC under Hadamard-style initializations. Our work extends this line of inquiry by showing how depth-induced implicit bias qualitatively alters the nature of convergence.

**Depth & low-rank bias:** Several lines of work have studied how network depth affects the rank of the solutions reached by optimization. Much of this literature originates in matrix factorization (Gunasekar et al., 2017; Arora et al., 2019; Li et al., 2020; Chou et al., 2020), where low-rank bias is induced in the absence of nonlinearities and is therefore conceptually distinct from the cross-entropy classification setting studied here. More recent work has explored the relationship between depth and rank in nonlinear architectures, both empirically (Huh et al., 2021) and in homogeneous networks (Jacot, 2023; Timor et al., 2023). In contrast, we analyze the representations that emerge in highly overparameterized classifiers, showing

that depth induces a low-rank bias that is reflected in smaller normalized margins and in the transition from neural collapse to softmax codes.

**Spectral initialization:** Spectral initialization analyses aim to understand learning behavior by reducing gradient-flow evolution to closed-form differential equations governing the singular values of the weight matrices, usually for linear networks with MSE loss. This began with work by Saxe et al. (2013; 2019), who established the foundational framework. Subsequent research extended these tools to explore discrete-time gradient descent (Gidel et al., 2019), alternative initializations (Kunin et al., 2024; Braun et al., 2022; Tarmoun et al., 2021) and to consider lazy vs rich learning regimes (Braun et al., 2022; Domin'e et al., 2024). More recent work has attempted to push these techniques beyond the purely linear setting. Mainali & Teixeira (2025) applied similar analyses to transformer architectures with linearized attention. Zhao & Thrampoulidis (2025b) used these tools in the UFM to study semantic structure in language data within the context of MSE loss. Garrod et al. (2025) advanced this line by establishing spectral initialization for the UFM with CE loss for a single hidden layer via Hadamard initialization. To date, however, no prior work has applied these techniques to deep networks trained with CE loss.

# B. Further background on deep UFM and NC

Here we provide more details about NC and the UFM construction. We study classification with $K$ classes and $n$ samples per class. We denote data point $i$ of class $c$ by $x_{ic} \in \mathbb{R}^{d_0}$, with corresponding one-hot encoded label $y_c \in \mathbb{R}^K$. A deep neural network $f(x) : \mathbb{R}^{d_0} \to \mathbb{R}^K$ models the relationship between inputs and labels. We decompose the network as $f_\theta(x) = W_L \sigma(W_{L-1}\sigma(...\sigma(W_1 h_{\bar{\theta}}(x))...))$, where $W_L \in \mathbb{R}^{K \times d}$ and $W_{L-1}, \ldots, W_1 \in \mathbb{R}^{d \times d}$ are weight matrices, and $h_{\bar{\theta}} : \mathbb{R}^{d_0} \to \mathbb{R}^d$ is a highly expressive feature map representing the remainder of the network. The activation function $\sigma : \mathbb{R} \to \mathbb{R}$ is applied elementwise. We denote the image of the data under $h_{\bar{\theta}}$ by $h_{\bar{\theta}}(x_{ic}) = h_{ic}$, and define the feature matrix $H_1 = [h_{1,1}, ..., h_{n,1}, h_{1,2}, ..., h_{n,K}] \in \mathbb{R}^{d \times Kn}$ as the collection of all features in class order.

The parameters $\{W_L, ..., W_1, \bar{\theta}\}$ are trained using a variant of gradient descent on CE loss without explicit regularization

$$\mathcal{L}(\theta) = \sum_{i,c} g(f_\theta(x_{ic}), y_c),$$

where $g$ implements the CE loss function. The matrix $H_1$ denotes the features entering the first separated layer of the fully connected head. Similarly, define $H_l$ for $l = 2, ..., L$ to be the features entering the $l^{\text{th}}$ layer of the fully connected head. DNC then refers to the following observations, which are found to approximately hold in overparameterized neural networks as training continues (Parker et al., 2023). For $L = 1$, this coincides with NC (Papyan et al., 2020).

**Definition B.1 (Deep Neural Collapse).** The last $L$ layers obey DNC if, for $l = 1, ..., L$, they satisfy:

- **DNC1**: Features collapse to their means, meaning there is a matrix $M_l \in \mathbb{R}^{K \times K}$ such that $H_l = M_l \otimes 1_n^T$.
- **DNC2**: The feature mean matrices $M_l$, after global centering, align with a simplex ETF, meaning $(M_l - \mu_G^{(l)} 1_K^T)^T (M_l - \mu_G^{(l)} 1_K^T) \propto S$, where $\mu_G = \frac{1}{K} \sum_j M_{ij}$.
- **DNC3**: The rows of the weight matrices $W_l$ are linear combinations of the columns of the matrix $M_l - \mu_G^{(l)} 1_K^T$.

It is shown by Garrod & Keating (2026) that Definition 2.1 naturally implies all of these properties.

## B.1. The Deep UFM Construction

To formally define the deep UFM, we approximate the feature map $h_{\bar{\theta}}(x)$ as being capable of mapping the training data to arbitrary points in feature space. Accordingly, we treat the feature vectors $h_{ic}$ as freely optimized variables and abstract away the parameters $\bar{\theta}$. This is motivated by the fact that $\bar{\theta}$ interact with the loss only through the features $h_{ic}$; therefore, high performing $\bar{\theta}$ should correspond to high performing feature vectors. In particular, when the feature map is sufficiently expressive, it should be able to map the data close to the optimal features recovered by the deep UFM. Consequently, the deep UFM is expected to describe deep neural networks in the overparametrization limit.

The loss function is then parameterized by $W_L, ..., W_1$ and $H_1$, and becomes

$$\mathcal{L} = g(Z) = -\sum_{c=1}^{K} \sum_{i=1}^{n} \log \left( \frac{\exp((z_{ic})_c)}{\sum_{c'=1}^{K} \exp((z_{ic})_{c'})} \right), \tag{7}$$

where $z_{ic} = W_L\sigma(W_{L-1}\sigma(...W_2\sigma(W_1 h_{ic}^{(1)})...))$ are the logit vectors, which make up the columns of the logit matrix $Z \in \mathbb{R}^{K \times Kn}$, using the same ordering as in the feature matrices. In the main text we set $\sigma = \mathrm{id}$ throughout, and leave consideration of homogeneous networks to further work.

## C. Further Results for Hadamard Initialization

Here we provide an additional result under the Hadamard initialization framework discussed in Section 4.1.

### C.1. Minimal Rank Logit Matrices for Hadamard Initialization

We derive a rank lower bound for logit matrices in this Hadamard setting that are compatible with a continuously decreasing loss along gradient flow. This hence informs us how low rank we can expect the logit matrix to become whilst still fitting the data in the Hadamard context.

**Proposition C.1.** *Let $K = 2^m$ for $m \in \mathbb{N}$, and let $\Phi$ be the Sylvester-Hadamard matrix of size $K$. The lowest rank achievable for a matrix of the form $Z = \Phi \mathrm{diag}(0, a)\Phi^T \in \mathbb{R}^{K \times K}$ satisfying $Z_{cc} - Z_{c'c} > 0$ for all $c' \neq c$, where $a \in \mathbb{R}_{\geq 0}^{K-1}$, is $m = \log_2(K)$.*

*Proof.* Denote the $i^{\text{th}}$ column of $\Phi$ by $\phi_i$, using zero indexing $i = 0, ..., K - 1$. Let

$$S = \{u \in \{1, ..., K - 1\} : a_u \neq 0\}, \tag{8}$$

so that $S$ is the support of $a$, and note that $\mathrm{rank}(Z) = |S|$. We can rewrite the entries of $Z = \Phi\mathrm{diag}(0, a)\Phi^T$ as

$$Z_{ij} = \sum_{u,v=0}^{K-1} (\phi_u)_i \mathrm{diag}(0, a)_{uv} (\phi_v)_j$$

$$= \sum_{u,v=1}^{K-1} (\phi_u)_i a_u \delta_{uv} (\phi_v)_j = \sum_{u=1}^{K-1} a_u (\phi_u)_i (\phi_u)_j.$$

Using that $\Phi$ is symmetric, this becomes

$$Z_{ij} = \sum_{u=1}^{K-1} a_u (\phi_i)_u (\phi_j)_u = \sum_{u=1}^{K-1} a_u (\phi_i \circ \phi_j)_u,$$

where $\circ$ denotes the Hadamard (entrywise) product. Using Lemma D.5, the columns satisfy $\phi_i \circ \phi_j = \phi_{i \oplus j}$, where $\oplus$ is the bitwise XOR. Hence

$$Z_{ij} = \sum_{u=1}^{K-1} a_u (\phi_{i \oplus j})_u.$$

In particular, when $i = j$ we have $Z_{ii} = \sum_u a_u$, while as $j$ ranges over $j \neq i$, the value $i \oplus j$ ranges over $\{1, ..., K - 1\}$, so the off-diagonal entries in column $i$ are exactly

$$\left\{ \sum_{u=1}^{K-1} a_u (\phi_k)_u : k = 1, ..., K - 1 \right\}.$$

Therefore, the condition that $Z_{cc} > Z_{c'c}$ for all $c' \neq c$ is equivalent to

$$\sum_{u=1}^{K} a_u > \sum_{u=1}^{K-1} a_u (\phi_k)_u, \quad \text{for all } k \in \{1, ..., K - 1\}.$$

Now identify the indices $\{1, ..., K - 1\}$ with elements of $\mathbb{F}_2^m$ via binary strings, recalling that $K = 2^m$. By Lemma D.6, we have $\Phi_{ij} = (-1)^{i \cdot j}$, where $i \cdot j$ is the mod-2 dot product. Thus

$$\sum_{u=1}^{K-1} a_u (\phi_k)_u = \sum_{u=1}^{K-1} a_u (-1)^{k \cdot u}.$$

We begin by proving the lower bound $|S| \geq m$ by contradiction. Suppose $|S| < m$. View $S \subset \mathbb{F}_2^m$ and let

$$V = \text{span}_{\mathbb{F}_2}(S) = \left\{ \sum_{j=1}^{|S|} \lambda_j s_j : s_j \in S, \lambda_j \in \mathbb{F}_2 \right\}. \tag{9}$$

Since $S$ has fewer than $m$ elements, $|V| \leq 2^{|S|} < 2^m$, and so $V \neq \mathbb{F}_2^m$. consequently $\dim(V) \leq m - 1$, and the orthogonal complement $V^\perp$ is non-empty:

$$\exists v^* \neq 0 : v^* \in V^\perp = \{x \in \mathbb{F}_2^m : x \cdot v = 0, \forall v \in V\}.$$

now note that the index $v^* \in V^\perp$ satisfies for all $u \in S \subset V$ that $v^* \cdot u = 0$, and hence $(-1)^{v^* \cdot u} = 1$. Consequently

$$a \cdot \phi_{v^*} = \sum_{u \in S} a_u (-1)^{v^* \cdot u} = \sum_{u \in S} a_u = \sum_{u=1}^{K-1} a_u.$$

This shows that for $k = v^* \neq 0$, the corresponding off-diagonal value equals the diagonal value, i.e. $Z_{cc} = Z_{c'c}$ for some $c' \neq c$, contradicting the strict inequality. Hence we must have $|S| \geq m$.

To prove that the minimal size of $S$ is exactly $m$, choose $S$ to be the standard basis of $\mathbb{F}_2^m$, and set $a_i = 1$ for all $i \in S$. then for any index $v \neq 0$ we have:

$$\sum_{u=1}^{K-1} a_u (-1)^{v \cdot u} = \sum_{r=1}^{m} (-1)^{v_r} = m - 2|\{r : v_r = 1\}| < m = \sum_{u=1}^{K-1} a_u,$$

where $v_r$ is the $r^{\text{th}}$ element of the binary representation of $v$. since $v \neq 0$, the set $\{r : v_r = 1\}$ must be non-empty, and so we must have $Z_{cc} - Z_{c'c} > 0$ for all $c' \neq c$. This completes the proof. $\square$

## D. Proofs

### D.1. Proof of Theorem 3.1

We consider the objective described in Eq. (3). We initially focus on the DNC solution, as described in Definition 2.1. In particular, since this definition specifies that the normalized matrices are balanced, and the asymptotic problem characterizes the normalized matrices, we consider balanced matrices throughout the proof. Hence DNC is given by

$$W_L = Q\Sigma U_L^T, \quad H_1 = U_1 \Sigma' V^T, \quad W_l = U_{l+1} \Sigma'' U_l^T, \quad l = 1, ..., L-1, \tag{10}$$

where $U_L, ..., U_1 \in \mathbb{R}^{d \times d}$ are any orthogonal matrices, $\Sigma \in \mathbb{R}^{K \times d}$, $\Sigma' \in \mathbb{R}^{d \times Kn}$, $\Sigma'' \in \mathbb{R}^{d \times d}$ all have their top $K \times K$ block given by $\alpha \text{diag}(1, 1, ..., 1, 0)$, for some $\alpha \in \mathbb{R}$. The matrices $Q \in \mathbb{R}^{K \times K}$ and $V \in \mathbb{R}^{Kn \times Kn}$ are orthogonal such that

$$Q \text{diag}(1, 1, ..., 1, 0) V^T = \frac{1}{\sqrt{n}} S \otimes 1_n^T,$$

and hence $Z = \frac{1}{\sqrt{n}} \alpha^{L+1} S \otimes 1_n^T$.

Note as a consequence we have for all $i = 1, ..., n$ and $c \neq c' = 1, .., K$

$$(z_{ic})_c - (z_{ic})_{c'} = \frac{1}{\sqrt{n}} \alpha^{L+1} \left[ \frac{K-1}{K} + \frac{1}{K} \right] = \frac{1}{\sqrt{n}} \alpha^{L+1},$$

so the constraint of Eq. (3) is simply $\alpha \geq (\sqrt{n})^{\frac{1}{L+1}}$.

The objective value is then

$$\|H_1\|_F^2 + \sum_{l=1}^{L} \|W_l\|_F^2 = (L+1)(K-1)\alpha^2 \geq (L+1)(K-1)n^{\frac{1}{L+1}}, \tag{11}$$

and this gives the objective value achieved by the DNC solution.

We shall now show that the cross-polytope solution outperforms DNC for the given range of $K$ and $L$. First we shall assume $K$ is even, we will cover the odd case after. The cross-polytope uses a similarly balanced construction to Eq. (10), and is given by

$$W_L = \tilde{Q}\tilde{\Sigma}\tilde{U}_L^T, \quad H_1 = \tilde{U}_1\tilde{\Sigma}'\tilde{V}^T, \quad W_l = \tilde{U}_{l+1}\tilde{\Sigma}''\tilde{U}_l^T, \quad l = 1, ..., L-1, \tag{12}$$

where $\tilde{U}_L, ..., \tilde{U}_1 \in \mathbb{R}^{d\times d}$ are any orthogonal matrices, $\tilde{\Sigma} \in \mathbb{R}^{K\times d}$, $\tilde{\Sigma}' \in \mathbb{R}^{d\times Kn}$, $\tilde{\Sigma}'' \in \mathbb{R}^{d\times d}$ all have their top $K \times K$ block given by $\beta\mathrm{diag}(1_{\frac{K}{2}}, 0_{\frac{K}{2}})$, for some $\beta \in \mathbb{R}$. The matrices $\tilde{Q} \in \mathbb{R}^{K\times K}$ and $\tilde{V} \in \mathbb{R}^{Kn\times Kn}$ are orthogonal such that

$$\tilde{Q}\mathrm{diag}(1_{\frac{K}{2}}, 0_{\frac{K}{2}})\tilde{V}^T = \frac{1}{2\sqrt{n}}C \otimes 1_n^T, \tag{13}$$

where $C \in \mathbb{R}^{K\times K}$ is given in block form by

$$C = \begin{bmatrix} X & 0 & ... & 0 \\ 0 & X & ... & 0 \\ ... & ... & ... & ... \\ 0 & 0 & ... & X \end{bmatrix}, \quad \text{where } X = \begin{bmatrix} 1 & -1 \\ -1 & 1 \end{bmatrix}. \tag{14}$$

Hence $Z = \frac{1}{2\sqrt{n}}\beta^{L+1}C \otimes 1_n^T$.

This gives that the constraint condition of the objective in Eq. (3) is

$$(z_{ic})_c - (z_{ic})_{c'} = \begin{cases} \frac{1}{\sqrt{n}}\beta^{L+1} & \text{if } c \text{ and } c' \text{ are paired in the cross-polytope,} \\ \frac{1}{2\sqrt{n}}\beta^{L+1} & \text{otherwise.} \end{cases}$$

Hence the constraint is satisfied when $\beta \geq (2\sqrt{n})^{\frac{1}{L+1}}$.

The objective value is then

$$\|H_1\|_F^2 + \sum_{l=1}^{L} \|W_l\|_F^2 = (L+1)\frac{K}{2}\beta^2 \geq (L+1)\frac{K}{2}(2\sqrt{n})^{\frac{2}{L+1}}. \tag{15}$$

This outperforms the NC objective value stated in Eq. (11) when

$$(L+1)\frac{K}{2}(2\sqrt{n})^{\frac{2}{L+1}} < (L+1)(K-1)n^{\frac{1}{L+1}},$$

which is equivalent to

$$2^{\frac{2}{L+1}} < 2\left(1 - \frac{1}{K}\right),$$

and this inequality holds when $L = 2, K \geq 6$, or $L \geq 3, K \geq 4$. This covers all even $K$ in this range.

It remains to cover the odd $K$. Here take the same solution as the even case specified in Eq. (12) and Eq. (13), but update the singular values so that $\tilde{\Sigma}, \tilde{\Sigma}', \tilde{\Sigma}''$ have their top $K \times K$ block given by $\beta\mathrm{diag}(1_{\frac{K-1}{2}}, 2^{-\frac{1}{L+1}}, 0_{\frac{K-1}{2}})$, and the orthogonal matrices $\tilde{Q}, \tilde{V}$ so that

$$\tilde{Q}\mathrm{diag}(1_{\frac{K-1}{2}}, \frac{1}{2}, 0_{\frac{K-1}{2}})\tilde{V}^T = \frac{1}{2\sqrt{n}}\bar{C} \otimes 1_n^T,$$

where

$$\bar{C} = \begin{bmatrix} C & 0 \\ 0 & 1 \end{bmatrix}, \quad \text{where } C \in \mathbb{R}^{K-1\times K-1} \text{ has entries as before.}$$

Consequently $Z = \frac{1}{2\sqrt{n}}\beta^{L+1}\bar{C} \otimes 1_n^T$.

The condition $(z_{ic})_c - (z_{ic})_{c'} \geq 1$ again reduces to $\beta \geq (2\sqrt{n})^{\frac{1}{L+1}}$, and now the objective takes value

$$\|H_1\|_F^2 + \sum_{l=1}^{L} \|W_l\|_F^2 = (L+1)\left[\frac{K-1}{2}\beta^2 + 2^{-\frac{2}{L+1}}\beta^2\right] \geq (L+1)(2\sqrt{n})^{\frac{2}{L+1}}\left[\frac{K-1}{2} + 2^{-\frac{2}{L+1}}\right], \tag{16}$$

and this outperforms the DNC objective value of Eq. (11) when

$$(L+1)n^{\frac{1}{L+1}}\left[\frac{K-1}{2}2^{\frac{2}{L+1}}+1\right] \le (L+1)(K-1)n^{\frac{1}{L+1}},$$

which is equivalent to

$$2^{\frac{2}{L+1}} \le 2\left(1-\frac{1}{K-1}\right).$$

This holds when $L = 2, K \ge 6$ or $L \ge 3, K \ge 5$.

Consequently we have that the final values of $L$ and $K$ for which DNC is suboptimal are given by $L = 2, K \ge 6$ and $L \ge 3, K \ge 4$.

It remains to prove local optimality of the DNC solution.

**Proof of Local Optimality:** Denote the set of parameters by $\Theta = (W_L, \ldots, W_1, H_1)$, and the max-margin objective by $E(\Theta) = \|H_1\|_F^2 + \sum_{l=1}^{L} \|W_l\|_F^2$. The logit matrix is a continuous mapping $Z(\Theta) = W_L \ldots W_1 H_1$. By Lemma D.7, for any $\Theta$, the objective is bounded below as follows:

$$E(\Theta) \ge (L+1)\|Z(\Theta)\|_{S_p}^p =: f(Z), \qquad p = \frac{2}{L+1}.$$

Moreover, for $\Theta^*$ the parameters of the DNC solution, for which $Z(\Theta^*) = Z_* = S \otimes 1_n^\top$, we have $E(\Theta^*) = (L+1)\|Z_*\|_{S_p}^p = f(Z_*)$.

To show that $\Theta^*$ is a local minimum of $E(\Theta)$, it suffices to prove that $Z_*$ is a strict local minimum of $f(Z)$ over the feasible set $\mathcal{Z}_{\text{feas}} = \{Z \mid g_{icc'}(Z) \ge 0\}$, where $g_{icc'}(Z) = (Z_{ic})_c - (Z_{ic})_{c'} - 1$. If this were true, then for any $\Theta$ sufficiently close to $\Theta^*$ such that $Z(\Theta) \ne Z_*$ and $Z(\Theta) \in \mathcal{Z}_{\text{feas}}$, we have $E(\Theta) \ge f(Z(\Theta)) > f(Z_*) = O(\Theta^*)$.

Let $r = K - 1$, and recall this is the rank of $Z_*$ which has $r$ identical singular values equal to $\sqrt{n}$. Define

$$f_{\text{smooth}}(Z) = (L+1) \sum_{i=1}^{r} \sigma_i(Z)^p, \tag{17}$$

over the top-$r$ singular values of $Z$. Because these top $r$ singular values of $Z_*$ are separated from the zero singular value by a strictly positive spectral gap, this function is twice continuously-differentiable in a neighborhood of $Z_*$. Specifically, its gradient

$$\nabla f_{\text{smooth}}(Z) = 2 U \text{diag}(\sigma_1^{p-1}, \ldots, \sigma_r^{p-1}) V^T, \quad \text{where } Z = U\text{diag}(\sigma_1, \ldots, \sigma_r)V^T,$$

at $Z_*$, evaluates to

$$G := \nabla_Z f_{\text{smooth}}(Z_*) = 2n^{-L/(L+1)}Z_*.$$

The constraints $g_{icc'}(Z)$ defining $\mathcal{Z}_{\text{feas}}$ are linear in $Z$. Also, at $Z_*$, all constraints are tight ($g_{icc'}(Z_*) = 0$) and, for any feasible $Z_k \in \mathcal{Z}_{\text{feas}}$, we have $g_{icc'}(Z_k) \ge 0$. By linearity, this implies:

$$\langle \nabla g_{icc'}(Z), Z_k - Z_* \rangle = g_{icc'}(Z_k) - g_{icc'}(Z_*) \ge 0. \tag{18}$$

Here, since

$$g_{icc'}(Z) = (e_c - e_{c'})^\top Z e_{ci} - 1 \implies \nabla_Z g_{icc'}(Z) = (e_c - e_{c'})e_{ci}^\top =: \nabla_Z g_{icc'}.$$

Thus, it is easy to check that

$$G = 2n^{-L/(L+1)}Z_* = \lambda \sum_{i,c,c' \ne c} \nabla g_{icc'}, \qquad \text{for } \lambda := \frac{2}{K}n^{-L/(L+1)} > 0. \tag{19}$$

Combining the above displays, it follows that for *any* feasible $Z_k$:

$$\langle G, Z_k - Z_* \rangle = \lambda \sum_{i,c,c' \ne c} \langle \nabla g_{icc'}, Z_k - Z_* \rangle \ge 0. \tag{20}$$

Now, assume for contradiction that $Z_*$ is *not* a *strict* local minimum of $f(Z)$ over $\mathcal{Z}_{\text{feas}}$. Then, for every $\epsilon > 0$, there exists at least one feasible point $Z \neq Z_*$ within an $\epsilon$-neighborhood of $Z_*$ that does not strictly increase the objective. By taking $\epsilon_k = 1/k$, we can explicitly construct an infinite sequence $(Z_k)_{k=1}^{\infty}$ converging to $Z_*$ such that for every $k \geq 1$: (1) $Z_k \in \mathcal{Z}_{\text{feas}}$, (2) $Z_k \neq Z_*$, and, (3) $f(Z_k) \leq f(Z_*)$.

Let $t_k = \|Z_k - Z_*\|_F$, so $t_k \to 0$, and $t_k > 0$ for all $k$. Define normalized directions $\Delta_k = (Z_k - Z_*)/t_k$. Since $\|\Delta_k\|_F = 1$ for all $k$, we can extract a convergent subsequence $\Delta_k \to \bar{\Delta}$ with $\|\bar{\Delta}\|_F = 1$. Without loss of generality, relabel the subsequence indices as $k$ so that the properties $f(Z_k) \leq f(Z_*)$ and $t_k > 0$ continue to hold for all $k$ in the subsequence.

Because $f_{\text{smooth}}$ is twice continuously differentiable in a neighborhood of $Z_*$, its Hessian $\nabla^2 f_{\text{smooth}}$ is continuous. On any sufficiently small compact ball centered at $Z_*$, the operator norm of the Hessian is bounded by a finite maximum $M$. For sufficiently large $k$, all matrices $Z_k$ in the sequence lie within this ball. In this case, by Taylor's theorem, there exists a uniform constant $C = M/2 > 0$ independent of $k$ such that for all for all large $k$:

$$f_{\text{smooth}}(Z_k) \geq f_{\text{smooth}}(Z_*) + t_k \langle G, \Delta_k \rangle - C t_k^2 \tag{21}$$

Because of Eq. (20) and $t_k > 0$, it holds that $\langle G, \Delta_k \rangle \geq 0$ for all $k$, which implies

$$\langle G, \bar{\Delta} \rangle \geq 0 . \tag{22}$$

We split the analysis into two cases based on $\bar{\Delta}$:

**Case 1:** $\langle G, \bar{\Delta} \rangle > 0$.

Let $\eta = \langle G, \bar{\Delta} \rangle > 0$. Since $f(Z) \geq f_{\text{smooth}}(Z)$ and $f(Z_*) = f_{\text{smooth}}(Z_*)$, by applying Eq. (21), for all sufficiently large $k$:

$$f(Z_k) - f(Z_*) \geq f_{\text{smooth}}(Z_k) - f_{\text{smooth}}(Z_*) \geq t_k \langle G, \Delta_k \rangle - C t_k^2 = t_k \left( \langle G, \Delta_k \rangle - C t_k \right)$$

Since $t_k \to 0$ and $C$ is a constant independent of $k$, the term $C t_k \to 0$. Meanwhile, because the inner product is a continuous linear functional and $\Delta_k \to \bar{\Delta}$, we have $\lim_{k \to \infty} \langle G, \Delta_k \rangle = \eta > 0$. Therefore, for sufficiently large $k$, we simultaneously have $\langle G, \Delta_k \rangle > \eta/2$ and $C t_k < \eta/2$. For all such $k$, the term in the parentheses is strictly positive, yielding $f(Z_k) - f(Z_*) > 0$, contradicting our assumption that $f(Z_k) \leq f(Z_*)$.

**Case 2:** $\langle G, \bar{\Delta} \rangle = 0$.

In this case, by Eq. (19),

$$0 = \langle G, \bar{\Delta} \rangle = \lambda \sum_{i,c,c' \neq c} \langle \nabla g_{icc'}, \bar{\Delta} \rangle \implies \langle \nabla g_{icc'}, \bar{\Delta} \rangle = 0, \forall i, c, c' \neq c ,$$

where for the implication we used that $\lambda > 0$ and from Eq. (18) that $\langle \nabla g_{icc'}, \bar{\Delta} \rangle \geq 0$ for all $i, c, c' \neq c$.

In fact, this gives

$$\langle \nabla g_{icc'}, \bar{\Delta} \rangle = \bar{\Delta}_{c,ic} - \bar{\Delta}_{c',ic} = 0 \implies \bar{\Delta}_{c,ic} = \bar{\Delta}_{c',ic}, \quad \forall i, c, c' \neq c \implies \bar{\Delta} = 1_K v^\top ,$$

for some $v$. Since $\|\bar{\Delta}\|_F = 1$, we know $v \neq 0$.

We will compute the singular values of

$$M_t := Z_* + t \bar{\Delta} = Z_* + t \, 1_K v^\top .$$

It suffices to work with the eigenvalues of

$$M_t^\top M_t = Z_*^\top Z_* + K t^2 v v^\top ,$$

where we used that $1_K^\top Z_* = 0$.

Now decompose

$$v = v_\| + v_0, \qquad v_\| \in \text{row}(Z_*), \quad v_0 \in \ker(Z_*).$$

Because all $K - 1$ nonzero singular values of $Z_*$ are equal to $\sqrt{n}$, $Z_*^\top Z_*$ acts as $nI$ on $\text{row}(Z_*)$ and as $0$ on $\ker(Z_*)$. Hence

$$Z_*^\top Z_* v_\| = n v_\|, \qquad Z_*^\top Z_* v_0 = 0.$$

We distinguish two subcases.

*Subcase 2a: $v_0 \neq 0$.*

We will show that there exists constant $c_1 > 0$ such that for all small enough $t$

$$\sigma_K(M_t) \geq c_1 t. \tag{23}$$

If this were true then we argue as follows: Write

$$Z_k = M_{t_k} + t_k E_k, \qquad E_k := \Delta_k - \bar{\Delta}, \qquad \|E_k\|_F \to 0.$$

By Weyl's inequality,

$$\sigma_K(Z_k) \geq \sigma_K(M_{t_k}) - t_k \|E_k\|_{\mathrm{op}}.$$

Since $\|E_k\|_{\mathrm{op}} \leq \|E_k\|_F \to 0$, for all sufficiently large $k$, by Eq. (23):

$$\sigma_K(Z_k) \geq \frac{c_1}{2} t_k.$$

Using Taylor of the smooth part (21),

$$f_{\mathrm{smooth}}(Z_k) - f_{\mathrm{smooth}}(Z_*) \geq -C t_k^2,$$

from which we conclude that

$$f(Z_k) - f(Z_*) \geq (L+1)\sigma_K(Z_k)^p - C t_k^2 \geq (L+1)\left(\frac{c_1}{2}\right)^p t_k^p - C t_k^2.$$

Because $p = \frac{2}{L+1} < 1$ when $L \geq 2$, the term $t_k^p$ dominates $t_k^2$ as $k \to \infty$. Hence $f(Z_k) > f(Z_*)$ for all sufficiently large $k$, contradicting the assumption $f(Z_k) \leq f(Z_*)$.

It remains proving Eq. (23). We first handle separately the degenerate case $v_\| = 0$. Since $vv^\top$ acts on $\ker(Z_*)$ while $Z_*^\top Z_*$ acts as $nI$ on $\mathrm{row}(Z_*)$ and vanishes on $\ker(Z_*)$, the non-zero eigenvalues of $M_t^\top M_t$ are

$$n \quad \text{with multiplicity } K-1, \qquad Kt^2\|v\|_2^2 \quad \text{with multiplicity } 1.$$

Therefore, for small enough $t < \sqrt{n/(K\|v\|^2)}$

$$\sigma_K(M_t) \geq c_1 t$$

for $c_1 = \sqrt{K}\,\|v\|_2$.

Assume now that both $v_\| \neq 0$ and $v_0 \neq 0$. Define orthonormal basis of $\mathrm{span}\{v_\|, v_0\}$:

$$e_1 := \frac{v_\|}{\|v_\|\|_2}, \qquad e_0 := \frac{v_0}{\|v_0\|_2}.$$

In this basis, the restriction of $M_t^\top M_t$ to $\mathrm{span}\{v_\|, v_0\}$ is

$$C_t = \begin{bmatrix} n + Kt^2\|v_\|\|_2^2 & Kt^2\|v_\|\|_2\,\|v_0\|_2 \\ Kt^2\|v_\|\|_2\,\|v_0\|_2 & Kt^2\|v_0\|_2^2 \end{bmatrix}.$$

Indeed, $Z_*^\top Z_*$ contributes $n$ on the $e_1$ direction and $0$ on the $e_0$ direction, while $vv^\top = (v_\| + v_0)(v_\| + v_0)^\top$, has the above $2 \times 2$ coordinate matrix in the basis $(e_1, e_0)$.

All remaining eigenvalues of $M_t^\top M_t$ are unchanged: they are equal to $n$ on the orthogonal complement of $e_1$ inside $\mathrm{row}(Z_*)$, and equal to $0$ on the orthogonal complement of $e_0$ inside $\ker(Z_*)$. Thus the only nontrivial eigenvalues are those of $C_t$.

Let $0 \leq \mu_-(t) \leq \mu_+(t)$ denote the eigenvalues of the positive semidefinite matrix $C_t$. Then,

$$\mu_-(t)\mu_+(t) = \det(C_t) = \left(n + Kt^2\|v_\|\|_2^2\right)Kt^2\|v_0\|_2^2 - K^2t^4\|v_\|\|_2^2\|v_0\|_2^2 = Knt^2\|v_0\|_2^2,$$

and

$$\mu_+(t) \leq \operatorname{tr}(C_t) = n + Kt^2\|v\|_2^2.$$

Combining,

$$\mu_-(t) \geq \frac{Knt^2\|v_0\|_2^2}{n + Kt^2\|v\|_2^2}.$$

Moreover,

$$\mu_-(t) \leq \operatorname{tr}(C_t)/2 = n/2 + Kt^2\|v\|^2/2.$$

For all sufficiently small $t > 0$, we have $n + Kt^2\|v\|_2^2 \leq 3n/2$, and hence

$$n > \frac{3n}{4} \geq \mu_-(t) \geq \frac{2K}{3}\|v_0\|_2^2\, t^2.$$

Thus, again, for all sufficiently small $t$, $\sigma_K(Z_k) = \sqrt{\mu_-(t_k)}$ and Eq. (23) holds.

*Subcase 2b: $v_0 = 0$.*

In this case, $v = v_\| \in \operatorname{row}(Z_*)$. Then

$$M_t^\top M_t = Z_*^\top Z_* + Kt^2 vv^\top.$$

Since $v$ lies in the row space of $Z_*$, one eigenvalue of $Z_*^\top Z_*$ equal to $n$ is perturbed to

$$n + Kt^2\|v\|_2^2,$$

while all other nonzero eigenvalues remain equal to $n$, and the zero eigenvalues remain unchanged. In particular, $\sigma_K(M_t) = 0$, and therefore

$$f(M_t) = f_{\text{smooth}}(M_t).$$

Define

$$\bar{g}(t) := f_{\text{smooth}}(Z_* + t\bar{\Delta}) = f(M_t).$$

Then

$$\bar{g}(t) - \bar{g}(0) = (L+1)\Big[\big(n + Kt^2\|v\|_2^2\big)^{p/2} - n^{p/2}\Big].$$

A Taylor expansion at $t = 0$ yields

$$\bar{g}(t) - \bar{g}(0) = \eta t^2 + O(t^4), \quad \eta := n^{p/2-1}K\|v\|_2^2 > 0.$$

Since also

$$\bar{g}'(0) = \langle G, \bar{\Delta}\rangle = 0,$$

it follows that

$$\bar{g}''(0) = 2\eta > 0.$$

Now define, for each $k$,

$$g_k(t) := f_{\text{smooth}}(Z_* + t\Delta_k).$$

By Taylor, there exists $s_k \in (0, t_k)$ such that

$$f_{\text{smooth}}(Z_k) - f_{\text{smooth}}(Z_*) = g_k(t_k) - g_k(0) = t_k g_k'(0) + \frac{t_k^2}{2}g_k''(s_k) = t_k\langle G, \Delta_k\rangle + \frac{t_k^2}{2}g_k''(s_k).$$

Here,

$$g_k''(t) = \nabla^2 f_{\text{smooth}}(Z_* + t\Delta_k)[\Delta_k, \Delta_k].$$

Because $s_k \to 0$ and $\Delta_k \to \bar{\Delta}$, and because $\nabla^2 f_{\text{smooth}}$ is continuous, we obtain

$$g_k''(s_k) \to \nabla^2 f_{\text{smooth}}(Z_*)[\bar{\Delta}, \bar{\Delta}] = \bar{g}''(0) = 2\eta.$$

Hence, for all sufficiently large $k$,

$$g_k''(s_k) \geq \eta.$$

Using also $\langle G, \Delta_k \rangle \geq 0$, we conclude that

$$f_{\text{smooth}}(Z_k) - f_{\text{smooth}}(Z_*) \geq \frac{\eta}{2} t_k^2 > 0$$

for all sufficiently large $k$.

Since $f(Z_k) \geq f_{\text{smooth}}(Z_k)$ and $f(Z_*) = f_{\text{smooth}}(Z_*)$, it follows that

$$f(Z_k) - f(Z_*) > 0,$$

contradicting the assumption $f(Z_k) \leq f(Z_*)$.

Thus both subcases lead to a contradiction, and Case 2 cannot occur.

### D.2. Proof of Theorem 3.2

Recall the constrained optimization problem stated in Eq. (3). Since here we are only interested in the global minimum of the constrained optimization problem, we can apply Lemma D.7 to reduce to the following alternative problem that shares its global minimizer

$$\min_Z \left\{ \|Z\|_{S_{\frac{2}{L+1}}}^{\frac{2}{L+1}} \right\} \quad \text{s.t. } (z_{ic})_c - (z_{ic})_{c'} \geq 1, \quad c' \neq c = 1, ..., K, \quad i = 1, ..., n. \tag{24}$$

Where the Schatten quasi norm is defined in Eq. (59). We now note that we assume the NC1 property, and so $Z = Z' \otimes 1_n^T$ for some $Z' \in \mathbb{R}^{K \times K}$. Since $Z$ and $Z'$ have the same singular values up to a scale of $\sqrt{n}$, we can further reduce the problem in this case to

$$\min_{Z'} \left\{ \|Z'\|_{S_{\frac{2}{L+1}}}^{\frac{2}{L+1}} \right\} \quad \text{s.t. } Z'_{cc} - Z'_{c'c} \geq 1, \quad c' \neq c = 1, ..., K. \tag{25}$$

Now expand the Schatten-quasi norm term in the $L \to \infty$ limit

$$\|Z'\|_{S_{\frac{2}{L+1}}}^{\frac{2}{L+1}} = \sum_{i=1}^{\text{rank}(Z')} \sigma_i^{\frac{2}{L+1}} = \sum_{i=1}^{\text{rank}(Z')} e^{\frac{2}{L+1} \log(\sigma_i)}$$

$$= \text{rank}(Z') + \frac{2}{L+1} \sum_{i=1}^{\text{rank}(Z')} \log(\sigma_i) + O\left(\frac{1}{L^2}\right),$$

where $\sigma_1, ..., \sigma_{\text{rank}(Z')}$ are the non-zero singular values of $Z'$. Hence we have the objective of Eq. (25) becomes

$$\min_{Z'} \left\{ \text{rank}(Z') + \frac{2}{L+1} \sum_{i=1}^{\text{rank}(Z')} \log(\sigma_i) + O\left(\frac{1}{L^2}\right) \right\} \quad \text{s.t. } Z'_{cc} - Z'_{c'c} \geq 1, \quad c' \neq c = 1, ..., K. \tag{26}$$

The global minimum in the $L \to \infty$ limit must be minimal rank such that the constraint is satisfied. We first remark that the constraint can be satisfied by matrices of rank two. The construction in the theorem statement suffices, since if $Z' = \beta X^T X$, where $X \in \mathbb{R}^{2 \times K}$ has each column vector of unit norm, and no vector repeated then $Z'_{cc} = \beta x_c^T x_c = \beta$, $Z'_{c'c} = \beta x_{c'}^T x_c < \beta$, and then setting $\beta > \max_{c,c' \neq c} (1 - x_{c'}^T x_c)^{-1}$ produces a rank two matrix that satisfies the constraint.

We also quote Lemma D.2, which states that, for $K > 2$, $Z'$ can only satisfy the condition if $\text{rank}(Z') \geq 2$, hence the minimal rank in our setting must be exactly two, and we reduce our objective value from Eq. (26) to

$$\min_{Z'} \left\{ \frac{2}{L+1} \sum_{i=1}^{\text{rank}(Z')} \log(\sigma_i) + O\left(\frac{1}{L^2}\right) \right\} \quad \text{s.t. } Z'_{cc} - Z'_{c'c} \geq 1, \quad c' \neq c = 1, ..., K, \text{ and } \text{rank}(Z') = 2.$$

We now use the technical assumption that $Z'$ is positive semi-definite, so that we can write $Z' = X^T X$ for $X \in \mathbb{R}^{2 \times K}$. Noting that minimizing $\sum_{i=1}^{\text{rank}(Z')} \log(\sigma_i)$ is the same as minimizing $\sigma_1 \sigma_2 = \det(XX^T)$, and that $Z'_{c'c} = x_{c'}^T x_c$, this reduces for large $L$ to

$$\min_{X \in \mathbb{R}^{2 \times K}} \left\{ \det(XX^T) \right\} \quad \text{s.t. } x_c^T x_c - x_{c'}^T x_c \geq 1, \quad c' \neq c = 1, ..., K,$$

or stated entirely in terms of the column vectors

$$\min_{x_i \in \mathbb{R}^2, i=1,...,K} \left\{ \det\left(\sum_i x_i x_i^T\right) \right\} \quad \text{s.t. } x_c^T x_c - x_{c'}^T x_c \geq 1, \quad c' \neq c = 1, ..., K. \tag{27}$$

We now use that the features have equal norm, meaning $\|x_i\| = r$ for all $i = 1, ..., K$. Writing $x_i = (r\cos(\theta_i), r\sin(\theta_i))$, and using the shorthand $c_i = \cos(\theta_i)$, $s_i = \sin(\theta_i)$ we have:

$$\sum_i x_i x_i^T = \sum_i \begin{bmatrix} rc_i & rs_i \end{bmatrix} \begin{bmatrix} rc_i \\ rs_i \end{bmatrix} = \sum_i \begin{bmatrix} r^2 c_i^2 & r^2 c_i s_i \\ r^2 c_i s_i & r^2 s_i^2 \end{bmatrix},$$

and hence

$$\det\left(\sum_i x_i x_i^T\right) = r^4 \sum_{i,j=1}^K [c_i^2 s_j^2 - c_i c_j s_i s_j].$$

Note that the two terms cancel when $i = j$. Combining the terms $(i,j)$ and $(j,i)$, this then becomes:

$$r^4 \sum_{i<j} [c_i^2 s_j^2 + c_j^2 s_i^2 - 2c_i c_j s_i s_j] = r^4 \sum_{i<j} (c_i s_j - c_j s_i)^2,$$

and hence

$$\det\left(\sum_i x_i x_i^T\right) = r^4 \sum_{i<j} \sin^2(\theta_i - \theta_j).$$

The constraint of Eq. (27) becomes

$$x_i^T x_i - x_i^T x_j = r^2 - r^2[c_i c_j + s_i s_j] = r^2(1 - \cos(\theta_i - \theta_j)) \geq 1,$$

so the optimization problem of Eq. (27) has reduced to

$$\min_{r,\theta_1,...,\theta_K} \left\{ r^4 \sum_{i<j} \sin^2(\theta_i - \theta_j) \right\} \quad \text{s.t. } r^2(1 - \cos(\theta_i - \theta_j)) \geq 1, \quad j \neq i = 1, ..., K. \tag{28}$$

Note that our set of constraints can only be satisfied when

$$r^2 \geq \max_{i,j\neq i}(1 - \cos(\theta_i - \theta_j))^{-1}.$$

Let $\delta = \min_{i,j\neq i}\{|\theta_i - \theta_j|\}$, where the angular difference is measured modulo $2\pi$, so as to capture the wrap around value as well. By the pigeonhole principle $\delta \leq \frac{2\pi}{K}$, and $\delta > 0$ else the constraint is not satisfied. Given that $K > 2$, we have our constraints satisfied when

$$r^2 \geq (1 - \cos(\delta))^{-1}.$$

Since the objective is monotonic increasing in $r$, when considering global minima we can set $r$ to this minimal value, and our objective of Eq. (28) is now

$$\min_{\theta_1,...,\theta_K} \left\{ \frac{\sum_{i<j} \sin^2(\theta_i - \theta_j)}{(1 - \cos(\delta))^2} \right\} \quad \text{s.t. } \delta = \min_{i,j\neq i}\{|\theta_i - \theta_j|\}. \tag{29}$$

Note that this has a rotational symmetry, $\theta_i \to \theta_i + \alpha$ leaves the optimal value unchanged.

We now aim to further simplify the expression in the numerator. using that for any angle $\phi$ we have $2\sin^2(\phi) = 1 - \cos(2\phi)$, we can rewrite the objective as

$$\frac{\frac{1}{4}K(K-1) - \frac{1}{2}\sum_{i<j}(\cos 2(\theta_i - \theta_j))}{(1 - \cos(\delta))^2}.$$

Next use that

$$\cos 2(\theta_u - \theta_v)) = \text{Re}(e^{2i(\theta_u - \theta_v)}) = \text{Re}(z_u \bar{z}_v),$$

where $z_u = e^{2i\theta_u}$, Re denotes the real part, and a bar denotes the complex conjugate. Also note that

$$\left| \sum_u z_u \right|^2 = \sum_{u,v} z_u \bar{z}_v = K + \sum_{u<v} (z_u \bar{z}_v + z_v \bar{z}_u) = K + 2 \sum_{u<v} \mathrm{Re}(z_u \bar{z}_v),$$

and hence we can write the objective as

$$\frac{\frac{1}{4}K^2 - \frac{1}{4} | \sum_u e^{2i\theta_u} |^2}{(1 - \cos(\delta))^2}.$$

We now use the rotational symmetry, choosing to transform $\theta_i \to \theta_i + \alpha$, where $\alpha$ is such that

$$\sum_u \sin(2\theta_u + 2\alpha) = 0.$$

Doing this, and redefining $\theta_u$, we have the reduction:

$$\left| \sum_u e^{2i\theta_u} \right|^2 = \left[ \sum_u \cos(2\theta_u) \right]^2,$$

and so our objective of Eq. (29) is now:

$$\min_{\theta_1,\ldots,\theta_K} \left\{ \frac{\frac{1}{4}K^2 - \frac{1}{4} [\sum_u \cos(2\theta_u)]^2}{(1 - \cos(\delta))^2} \right\}, \tag{30}$$

with the constraint that the gaps are at least $\delta$.

We reduce this to a problem solely in $\delta$ by minimizing the numerator subject to the condition. This is equivalent to:

$$\max_{\theta_1,\ldots,\theta_k} \left\{ \sum_u \cos(2\theta_u) \right\}, \tag{31}$$

subject to the smallest gap being $\delta$. Here we use Lemma D.3, which states that for $K$ even, or $K$ odd and $\delta \in (0, \frac{2\pi}{K+1}]$, the maximum value of this sum, subject to the constraint, is given by

$$\frac{\sin\left( \lceil \frac{K}{2} \rceil \delta \right) + \sin\left( \lfloor \frac{K}{2} \rfloor \delta \right)}{\sin(\delta)}.$$

We will return to the odd $K$, $\delta \in (\frac{2\pi}{K+1}, \frac{2\pi}{K}]$ case later.

Our full optimization problem of Eq. (30) becomes:

$$\min_{\delta \in (0, \frac{2\pi}{K}]} \left\{ \frac{K^2 \sin^2(\delta) - (\sin\left( \lceil \frac{K}{2} \rceil \delta \right) + \sin\left( \lfloor \frac{K}{2} \rfloor \delta \right))^2}{4(\sin(\delta))^2(1 - \cos(\delta))^2} \right\}. \tag{32}$$

From here we focus on the case where $K = 2m$ for some integer $m$, the proof for the odd case for this objective is similar. The goal is to show this is minimized at $\delta = \frac{2\pi}{K}$, and hence the vectors are uniformly spaced on a circle. Writing $K = 2m$, we have

$$\min_{\delta \in (0, \frac{\pi}{m}]} \left\{ \frac{m^2 \sin^2(\delta) - \sin^2(m\delta)}{\sin^2(\delta)(1 - \cos(\delta))^2} \right\}. \tag{33}$$

Define the function

$$F(\delta) = \frac{\sin(m\delta)}{\sin(\delta)} = \frac{e^{im\delta} - e^{-im\delta}}{e^{i\delta} - e^{i\delta}} = \sum_{u=0}^{m-1} e^{(2u-(m-1))i\delta}.$$

Taking a square

$$F^2(\delta) = \sum_{u,v=0}^{m-1} e^{-2(m-1)i\delta} e^{(2u+2v)i\delta} = m + 2 \sum_{u=1}^{m-1} (m - u) \cos(2u\delta).$$

So we can rewrite our objective as

$$\min_{\delta \in (0, \frac{\pi}{m}]} \left\{ \frac{m^2 - m - 2\sum_{u=1}^{m-1}(m-u)\cos(2u\delta)}{(1 - \cos(\delta))^2} \right\}. \tag{34}$$

Now use that

$$2\sum_{j=1}^{m-1}(m-j) = m^2 - m,$$

to further reduce to

$$\min_{\delta \in (0, \frac{\pi}{m}]} \left\{ 2\sum_{u=1}^{m-1}(m-u)\frac{(1 - \cos(2u\delta))}{(1 - \cos(\delta))^2} \right\}.$$

It is clear that it is sufficient for the objective to be monotonic decreasing on its domain, and a sufficient condition for this to occur is that each of the functions:

$$f_u(\delta) = \frac{1 - \cos(2u\delta)}{(1 - \cos(\delta))^2}, \quad u = 1, .., m - 1, \tag{35}$$

is monotonic decreasing on the range $(0, \frac{\pi}{m}]$.

Using standard trigonometric identities, and defining $\delta = 2t$, this reduces to

$$f_u(t) = \frac{2\sin^2(2ut)}{4\sin^4(t)}, \quad u = 1, ..., m - 1, \quad t \in \left(0, \frac{\pi}{2m}\right].$$

Dropping a multiplicative factor, and using a square root, which doesn't change monotonicity, we can redefine our functions to

$$f_u(t) = \frac{\sin(2ut)}{\sin^2(t)}, \quad u = 1, ..., m - 1, \quad t \in \left(0, \frac{\pi}{2m}\right].$$

Taking a derivative, the condition for monotonic decreasing is equivalent to

$$u\cot(2ut) < \cot(t), \quad u = 1, ..., m - 1, \quad t \in \left(0, \frac{\pi}{2m}\right].$$

Note the right hand side is always positive on the range, so this inequality holds trivially if the left hand side is negative. This occurs if $2ut > \pi/2$. Otherwise, we use the fact that the function $x\cot(x)$ is strictly decreasing on $x \in (0, \pi/2]$, and hence, since $2ut > t$, we have

$$2ut\cot(2ut) < t\cot(t),$$

so that

$$u\cot(2ut) \le 2u\cot(2ut) < t\cot(t),$$

So each of these functions is monotonic decreasing on the given range.

Hence our objective is monotonic decreasing in the minimum gap $\delta$, and so the best objective occurs when $\delta$ is maximized, which corresponds with the Gram vectors $x_1, ..., x_K$ being uniformly spaced on a circle in some order.

The remaining sub-case is $K$ odd, and $\delta \in (\frac{2\pi}{K+1}, \frac{2\pi}{K}]$. Lemma D.3 also provides the maximum of the cosine sum in Eq. (31) in this case, giving

$$\max_{\theta_1, ..., \theta_k \text{ s.t. gaps} \ge \delta} \left\{ \sum_u \cos(2\theta_u) \right\} = \frac{|\sin(K\delta)|}{\sin(\delta)}.$$

Following the same steps above, it only remains to prove that the following objective

$$\min_{\delta \in (\frac{2\pi}{K+1}, \frac{2\pi}{K}]} \left\{ \frac{K^2 - \frac{\sin^2(K\delta)}{\sin(\delta)}}{4(1 - \cos(\delta))^2} \right\}, \tag{36}$$

takes its minimal value at $\delta = \frac{2\pi}{K}$.

To show this, write $u(\delta) = \sin(K\delta)/\sin(\delta)$ and denote the objective function in Eq. (36) by $f(\delta)$. Also, let $\gamma = 2\pi - K\delta$, so that $\gamma \in [0, \frac{2\pi}{K+1})$. We note that $\gamma \leq \delta$, and $\delta + \gamma \leq \frac{4\pi}{K+1} \leq \pi$. Hence

$$0 \leq \gamma \leq \delta \leq \pi - \gamma \implies \sin(\gamma) \leq \sin(\delta).$$

and so

$$u(\delta) = \frac{-\sin(\gamma)}{\sin(\delta)} \in [-1, 0],$$

and

$$\frac{f'(\delta)}{f(\delta)} = \frac{-2uu'}{K^2 - u^2} - \frac{2\sin(\delta)}{1 - \cos(\delta)}.$$

Note since $-u \in [0, 1]$, we have $K^2 - u^2 \geq K^2 - 1$. Also

$$u'(\delta) = \frac{1}{\sin^2(\delta)}[K\cos(K\delta)\sin(\delta) - \sin(K\delta)\cos(\delta)].$$

We then use that $\cos(K\delta) \leq 1$, $-\sin(K\delta) \leq \sin(\delta)$ and $\cos(\delta) \leq 1$, which gives

$$u'(\delta) \leq \frac{K\sin(\delta) + \sin(\delta)}{\sin^2(\delta)} = \frac{K+1}{\sin(\delta)}.$$

Also note that since $\delta \leq \frac{2\pi}{K} \leq \frac{2\pi}{3}$, we have

$$\frac{\sin(\delta)}{1 - \cos(\delta)} = \frac{1 + \cos(\delta)}{\sin(\delta)} \geq \frac{1}{2\sin(\delta)},$$

since the inequality reduces to $1 + 2\cos(\delta) \geq 0$, which is true on our range of $\delta$.

Also note it is the case that $K^2 = 1 \geq 2K + 2$ for $K \geq 3$. Combining all of this gives

$$\frac{-2uu'}{K^2 - u^2} \leq \frac{2u'}{K^2 - 1} \leq \frac{2(K+1)}{(K^2 - 1)\sin(\delta)} \leq \frac{1}{\sin(\delta)} \leq \frac{2\sin(\delta)}{1 - \cos(\delta)},$$

and hence

$$\frac{f'(\delta)}{f(\delta)} \leq 0.$$

Finally, noting that $f(\delta)$ is always greater than zero gives the result that $f$ is decreasing on this range, and hence the minimal value occurs at $\delta = \frac{2\pi}{K}$.

### D.3. Proof of Theorem 4.2

We begin by considering the deep UFM with linear activations as defined in Eq. (1), denoting the logit matrix as $Z = W_L...W_1H_1$. We will denote $W_0 := H_1$ to make equations more compact.

We have the following elementary calculations:

$$\frac{\partial Z_{xy}}{\partial (W_l)_{ab}} = (A_{l+1})_{xa}(H_l)_{by},$$

$$\frac{\partial g(Z)}{\partial Z} = P - Y,$$

where we have defined the following quantities:

$$A_{l+1} = W_L...W_{l+1}, \quad A_{L+1} = I_K, \quad H_l = W_{l-1}...W_1H_1, \quad H_0 = I_{Kn}, \tag{37}$$

$$Y = I_K \otimes 1_n^T,$$

$$P_{ij} = \frac{\exp(Z_{ij})}{\sum_k \exp(Z_{kj})},$$

from which it is clear that the equations of gradient flow are

$$\frac{dW_l}{dt} = A_{l+1}^T (Y - P) H_l^T. \tag{38}$$

We shall now specialize to Hadamard initialization. Recall we initialize with $W_L = U D_L R_L^T$, $W_l = R_{l+1} D_l R_l^T$ for $l = 1, ..., L - 1$. $H_1 = R_1 D_0 V^T$, where $U = \frac{1}{\sqrt{K}} \Phi$, $V = U \otimes Q$, where $\Phi$ is the $K \times K$ Sylvester Hadamard matrix, and $Q$ is a right singular matrix of $1_n^T$, meaning $1_n^T = \sqrt{n} e_1^{(n)T} Q^T$, where $e_1^{(n)}$ is the first standard basis vector in $\mathbb{R}^n$. Also each $D_l$ has its first $K$ diagonal entries given by $\alpha_0^{(l)}, ..., \alpha_{K-1}^{(l)}$, with all other entries being zero, so that the initializations are in the form of a singular value decomposition.

As a consequence, we have at initialization that $Z = U D_Z V^T$, where the first $K$ diagonal entries of $D_Z$ are given by $a_i = \prod_{l=0}^{L} \alpha_i^{(l)}$ for $i = 0, ..., K - 1$.

We can write this as

$$D_Z = [D_a, 0_{K \times K(n-1)}] = D_a \otimes e_1^{(n)T},$$

where $D_a = \mathrm{diag}(a_0, ..., a_{K-1}) \in \mathbb{R}^K$. Hence

$$Z = U(D_a \otimes e_1^{(n)T})(U \otimes Q)^T = U[(D_a U^T) \otimes (e_1^{(n)T} Q^T)],$$

then using the SVD of $1_n^T$

$$Z = \frac{1}{\sqrt{n}} U[D_a U^T \otimes 1_n^T],$$

and using that this has repeated columns

$$Z = (U D_{\tilde{a}} U^T) \otimes 1_n^T,$$

where we have defined $\tilde{a} = \frac{1}{\sqrt{n}} a$ and $D_{\tilde{a}} = \mathrm{diag}(\tilde{a}_0, ..., \tilde{a}_{K-1})$. We now apply Lemma D.4, and use that there are repeated columns, to get the form of the softmax matrix

$$P(Z) = (U D_{\tilde{\nu}} U^T) \otimes 1_n^T,$$

where

$$\tilde{\nu}_i = \frac{\sum_j \Phi_{ij} \exp(\frac{1}{K} (\Phi \tilde{a})_j)}{\sum_j \exp(\frac{1}{K} (\Phi \tilde{a})_j)}. \tag{39}$$

Noting that $Y = \sqrt{n} U[I_K, 0_{K \times K(n-1)}] V^T$, and that under our initialization

$$A_{l+1} = U D_L ... D_{l+1} R_{l+1}^T, \quad H_l = R_l D_{l-1} ... D_0 V^T,$$

we have our gradient flow equations specified in Eq. (38) become

$$\frac{dW_l}{dt} = \sqrt{n} R_{l+1} \left[ \prod_{r=l+1}^{L} D_r^T \right] \mathrm{diag}(1_K - \tilde{\nu}, 0_{K \times K(n-1)}) \left[ \prod_{r=0}^{l-1} D_r^T \right] R_l^T.$$

We see that if $W_l$ has singular vectors given by $R_{l+1}$ and $R_l$ at initialization, then the derivative has the same singular vectors. This implies that this property is preserved for all future times if it holds at initialization.

Consequently, only the singular values of $W_l$ evolve, and the singular vectors remain fixed. These singular values evolve under

$$\frac{dD_l}{dt} = \sqrt{n} \left[ \prod_{r=l+1}^{L} D_r^T \right] \mathrm{diag}(1_K - \tilde{\nu}, 0_{K \times K(n-1)}) \left[ \prod_{r=0}^{l-1} D_r^T \right].$$

Stated more succinctly in terms of the individual values:

$$\frac{d\alpha_i^{(l)}}{dt} = \sqrt{n}(1 - \tilde{\nu}_i) \prod_{r \neq l} \alpha_i^{(r)}. \tag{40}$$

We now note that

$$1 - \tilde{\nu}_i = \frac{\sum_j (1 - \Phi_{ij}) \exp(\frac{1}{K}(\Phi \tilde{a})_j)}{\sum_j \exp(\frac{1}{K}(\Phi \tilde{a})_j)}.$$

For $i = 0$, we clearly have $1 - \tilde{\nu}_i = 0$ since the first column and row of $\Phi$ are given by $1_K$. Hence $\frac{d}{dt}(\alpha_0^{(l)}) = 0$ for $l = 0, ..., L$. For the other cases, dividing the numerator and denominator by $\exp(\frac{1}{K} \sum_j \tilde{a}_j)$ gives

$$1 - \tilde{\nu}_i = \frac{\sum_{j=1}^{K-1} \Psi_{ij} \exp(-\frac{1}{K\sqrt{n}}(\Psi a)_j)}{1 + \sum_{j=1}^{K-1} \exp(-\frac{1}{K\sqrt{n}}(\Psi a)_j)},$$

where $\Psi \in \mathbb{R}^{K-1 \times K-1}$ is the matrix formed by deleting the first column and row of $1_K 1_{K-1}^T - \Phi$, and we used that $\tilde{a} = \frac{1}{\sqrt{n}} a$. Hence we can write our gradient flow as

$$\frac{d\alpha_i^{(l)}}{dt} = \frac{\sqrt{n}}{D} b_i \prod_{r \neq l} \alpha_i^{(r)}, \quad i = 1, ..., K-1, \quad \frac{d\alpha_0^{(l)}}{dt} = 0, \tag{41}$$

where

$$b_i = \sum_{j=1}^{K-1} \Psi_{ij} e^{-\frac{1}{K\sqrt{n}}(\Psi a)_j}, \quad D = 1 + \sum_{j=1}^{K-1} e^{-\frac{1}{K\sqrt{n}}(\Psi a)_j},$$

with $a_i = \prod_{l=0}^{L} \alpha_i^{(l)}$.

It is straightforward to verify that for all $l, r = 0, ..., L+1$, the quantities $(\alpha_i^{(l)})^2 - (\alpha_i^{(r)})^2$ are conserved along the flow given in Eq. (41). From small initialization such conserved quantities would be approximately zero. Here we set them exactly to zero, in which case we have

$$\alpha_i^{(l)} = \alpha_i^{(r)} \quad \forall l, r = 0, ..., L, \tag{42}$$

where we have used that singular values are non-negative. This is known as the balancedness condition. Hence we can drop the $l$ index from these singular values, writing them as $\alpha_i$. This reduces us to only $K - 1$ evolving variables, one for each $i = 1, ..., K-1$. We define the logit singular values to be

$$a_i = \prod_{l=0}^{L} \alpha_i^{(l)} = \alpha_i^{L+1}.$$

Transforming to such variables captures the full equations. Using Eq. (41), these quantities evolve as:

$$\frac{da_i}{dt} = (L+1)\alpha_i^L \frac{d\alpha_i}{dt} = \sqrt{n}(L+1)\frac{b_i}{D}\alpha_i^{2L} = \sqrt{n}(L+1)\frac{b_i}{D}a_i^{\frac{2L}{L+1}},$$

where here

$$b_i = \sum_{j=1}^{K-1} \Psi_{ij} e^{-\frac{1}{K\sqrt{n}}(\Psi a)_j}, \quad D = 1 + \sum_{j=1}^{K-1} e^{-\frac{1}{K\sqrt{n}}(\Psi a)_j},$$

Absorbing constants into the $a$ and $t$ variables via the transform

$$a' = \frac{1}{K\sqrt{n}}a, \quad t' = \frac{1}{K}(L+1)(K\sqrt{n})^{\frac{2L}{L+1}}t,$$

and dropping the primes then gives the final reduction stated in Eq. (6).

$$\frac{da_i}{dt} = \frac{1}{D}b_i a_i^{\frac{2L}{L+1}},$$

with $b_i = \sum_{j=1}^{K-1} \Psi_{ij} e^{-(\Psi a)_j}$ and $D = 1 + \sum_{j=1}^{K-1} e^{-(\Psi a)_j}$.

## D.4. Proof of Theorem 4.3

We write our singular values in the following form, so as to linearize around the DNC solution

$$a_i = \mu + \mu\epsilon_i, \quad \sum_i \epsilon_i = 0, \quad \mu = \frac{1}{K-1}\sum_i a_i = \frac{1}{K-1}\|a\|_1, \tag{43}$$

and let $\sum_i |\epsilon_i| = \|\epsilon\|_1 \ll 1$. This gives that

$$(\Psi a)_i = K\mu + \mu(\Psi\epsilon)_i.$$

Consequently

$$D = 1 + e^{-K\mu}\sum_j (1 - \mu(\Psi\epsilon)_j) + O(\|\epsilon\|_1^2).$$

Then using that $\sum_i \Psi_{ij} = K$, this reduces to

$$D = 1 + (K-1)e^{-K\mu} + O(\|\epsilon\|_1^2). \tag{44}$$

Similarly

$$b_i = e^{-K\mu}\sum_j \Psi_{ij}[1 - \mu(\Psi\epsilon)_j] + O(\|\epsilon\|_1^2)$$

$$= e^{-K\mu}[K - \mu(\Psi^2\epsilon)_i] + O(\|\epsilon\|_1^2).$$

Then using that $\Psi^2 = K(I_{K-1} + 1_{K-1}1_{K-1}^T)$ gives

$$b_i = Ke^{-K\mu}(1 - \mu\epsilon_i) + O(\|\epsilon\|_1^2). \tag{45}$$

Hence, inputting Eq. (44) and (45) into the gradient flow equation (6) the derivative is given by

$$\frac{da_i}{dt} = \frac{Ke^{-K\mu}}{1+(K-1)e^{-K\mu}}(1-\mu\epsilon_i)(\mu+\mu\epsilon_i)^{\frac{2L}{L+1}} + O(\|\epsilon\|_1^2)$$

$$= \frac{Ke^{-K\mu}\mu^{\frac{2L}{L+1}}}{1+(K-1)e^{-K\mu}}(1-\mu\epsilon_i)\left(1 + \frac{2L}{L+1}\epsilon_i\right) + O(\|\epsilon\|_1^2)$$

$$= \underbrace{\frac{Ke^{-K\mu}\mu^{\frac{2L}{L+1}}}{1+(K-1)e^{-K\mu}}}_{C(\mu)}\left[1 + \underbrace{\left(\frac{2L}{L+1} - \mu\right)}_{B(\mu)}\epsilon_i\right] + O(\|\epsilon\|_1^2),$$

where we have defined two constants $C(\mu)$ and $B(\mu)$. Now consider how the ratio of normalized logit values changes in this regime

$$\frac{d}{dt}\left(\frac{\hat{a}_i}{\hat{a}_j}\right) = \frac{1}{a_j^2}\left(a_j\frac{da_i}{dt} - a_i\frac{da_j}{dt}\right)$$

$$= \frac{C(\mu)}{a_j^2}[(\mu+\mu\epsilon_j)(1+B(\mu)\epsilon_i) - (\mu+\mu\epsilon_i)(1+B(\mu)\epsilon_j)] + O(\|\epsilon\|_1^2)$$

$$= \frac{\mu C(\mu)}{a_j^2}(B(\mu)-1)(\epsilon_i - \epsilon_j) + O(\|\epsilon\|_1^2).$$

Then using the expression for $B(\mu)$ this becomes

$$\frac{d}{dt}\left(\frac{\hat{a}_i}{\hat{a}_j}\right) = \frac{\mu C(\mu)}{a_j^2}\left[\frac{L-1}{L+1} - \mu\right](\epsilon_i - \epsilon_j) + O(\|\epsilon\|_1^2).$$

Noting that the sign of $\hat{a}_i - \hat{a}_j$ is the same as the sign of $\epsilon_i - \epsilon_j$, we see that in the regime linearized around DNC whether two distinct modes grow towards each other depends on the sign of $\frac{L-1}{L+1} - \mu$. In particular for $\frac{L-1}{L+1} > \mu$, DNC is unstable at

all scales, whilst for $\frac{L-1}{L+1} < \mu$ it is stable at all scales. This recovers the bound in the theorem statement when we recall that $\|a\|_1 = (K-1)\mu$. We also note this bound is trivial for $L = 1$, and so DNC is stable throughout the full loss surface in this case.

We now show an alternative low-rank structure that is stable above a threshold in the norm of the logit singular values. Define the vector $c \in \mathbb{R}^{K-1}$ that has value one in odd indices and value 0 in even indices. Write the singular values in the following form:

$$a = \mu c + \mu\epsilon, \quad \mu = \frac{2}{K}\|a\|_1, \quad \sum_i \epsilon_i = 0, \tag{46}$$

and consider the regime where $\|\epsilon\|_1 = \sum_i |\epsilon_i| \ll 1$. Using Lemma D.9, we have that the corresponding $b$ vector is given by:

$$b_i = \begin{cases} Ke^{-\frac{K}{2}\mu} + 2(e^{-K\mu} - e^{\frac{K}{2}\mu})(1 - \mu(\Psi\epsilon)_1) - Ke^{-\frac{K}{2}\mu}\mu\epsilon_i + O(\|\epsilon\|_1^2), & \text{if } i \text{ is odd,} \\ Ke^{-\frac{K}{2}\mu} - Ke^{-\frac{K}{2}\mu}\mu\epsilon_i + O(\|\epsilon\|_1^2), & \text{if } i \text{ is even.} \end{cases}$$

We also have

$$a_i^{\frac{2L}{L+1}} = \begin{cases} \mu^{\frac{2L}{L+1}}(1 + \frac{2L}{L+1}\epsilon_i) + O(\|\epsilon\|_1^2), & \text{if } i \text{ is odd,} \\ \mu^{\frac{2L}{L+1}}\epsilon_i^{\frac{2L}{L+1}}, & \text{if } i \text{ is even.} \end{cases}$$

Hence the derivatives of the logit singular values are given by

$$\frac{da_i}{dt} = \begin{cases} \frac{1}{D}\mu^{\frac{2L}{L+1}}e^{-\frac{K}{2}\mu}[K(1 + \frac{2L}{L+1}\epsilon_i - \mu\epsilon_i) + 2(e^{-\frac{K}{2}\mu} - 1)(1 + \frac{2L}{L+1}\epsilon_i - \mu(\Psi\epsilon)_1)] + O(\|\epsilon\|_1^2), & \text{if } i \text{ is odd,} \\ \frac{1}{D}Ke^{-\frac{K}{2}\mu}\mu^{\frac{2L}{L+1}}\epsilon_i^{\frac{2L}{L+1}} + O(\|\epsilon\|_1^2), & \text{if } i \text{ is even.} \end{cases} \tag{47}$$

We now can compare how the relative size of the modes grows. First let $i$ be odd, and $j$ be even, then we have

$$\frac{d\hat{a}_i}{d\hat{a}_j} = \frac{1}{a_j^2}\left[a_j\frac{da_i}{dt} - a_i\frac{da_j}{dt}\right]$$

$$= \frac{e^{-\frac{K}{2}\mu}\mu^{1+\frac{2L}{L+1}}}{Da_j^2}\left[(K - 2 + 2e^{-\frac{K}{2}\mu})\epsilon_j - Ke^{-\frac{K}{2}\mu}\epsilon_j^{\frac{2L}{L+1}} + O(\|\epsilon\|_1^2)\right]. \tag{48}$$

When $L > 1$ the second term in the brackets is higher order, and so to leading order this is given by

$$= \frac{e^{-\frac{K}{2}\mu}\mu^{1+\frac{2L}{L+1}}}{Da_j^2}\left[(K - 2 + 2e^{-\frac{K}{2}\mu})\epsilon_j + O(\|\epsilon\|_1^{\frac{2L}{L+1}})\right].$$

We see that when $\|\epsilon\|_1$ is small enough, the dominant term is given by $\epsilon_j$ up to a positive factor. Note, that $\epsilon_j > 0$. This is since $a_j = \mu\epsilon_j$, and we initialize the singular values as positive, from which they must remain so. Hence the derivative $\frac{d}{dt}(\hat{a}_i/\hat{a}_j)$ is positive, and the odd modes grow relative to the even modes, when $L > 1$. It is also simple to see from Eq. (48) that the opposite is the case for $L = 1$.

Next let $i$ and $j$ both be odd, after simplifying the expression we find in this case the relative size grows as

$$\frac{d\hat{a}_i}{d\hat{a}_j} = \frac{\mu^{1+\frac{2L}{L+1}}e^{-\frac{K}{2}\mu}}{Da_j^2}\left[\left(K(\mu - \frac{L-1}{L+1}) + 2\frac{L-1}{L+1}(1 - e^{-\frac{K}{2}\mu})\right)(\epsilon_j - \epsilon_i) + O(\|\epsilon\|_1^2)\right].$$

When $\|\epsilon\|_1$ is small enough, a sufficient condition for the leading order term to have the same sign as $\epsilon_j - \epsilon_i$ is $\mu > \frac{L-1}{L+1}$.

Hence we see that the modes close to zero in the structure continue to become small relative to the large modes, and the large modes are drawn to equality, implying $\hat{a}$ is moving towards $\frac{2}{K}c$ when $L > 1$ and $\mu > \frac{L-1}{L+1}$, implying this structure is stable throughout this region.

Lastly, we demonstrate the KL divergence can monotonically increase to infinity. To demonstrate this, consider an initialization of the logit singular values of the form

$$a = [\gamma, \delta, \gamma, ..., \delta, \gamma], \quad \gamma, \delta > 0, \tag{49}$$

so that there are $\frac{K}{2}$ appearances of $\gamma$ and $\frac{K}{2} - 1$ appearances of $\delta$. Using Lemma D.8, we have that the corresponding $b_i$ is given by

$$b_i = \begin{cases} 2e^{-K\gamma} + (K-2)e^{-\frac{K}{2}(\gamma+\delta)}, & \text{if } i \text{ is odd,} \\ Ke^{-\frac{K}{2}(\gamma+\delta)}, & \text{if } i \text{ is even.} \end{cases}$$

Consequently, the logit singular value evolution reduces to

$$\frac{da_i}{dt} = \begin{cases} \frac{1}{D}\gamma^{\frac{2L}{L+1}}\left(2e^{-K\gamma} + (K-2)e^{-\frac{K}{2}(\gamma+\delta)}\right), & \text{if } i \text{ is odd,} \\ \frac{1}{D}\delta^{\frac{2L}{L+1}}\left(Ke^{-\frac{K}{2}(\gamma+\delta)}\right), & \text{if } i \text{ is even.} \end{cases} \tag{50}$$

We see that if $a$ is initialized in the form of Eq. (49), it remains in this form under gradient flow. Now we consider how the relative size of the two distinct values changes

$$\frac{d}{dt}\left(\frac{\gamma}{\delta}\right) = \frac{1}{\delta^2}\left(\delta\frac{d\gamma}{dt} - \gamma\frac{d\delta}{dt}\right)$$

$$= \frac{1}{D\delta^2}\left[\delta\gamma^{\frac{2L}{L+1}}\left(2e^{-K\gamma} + (K-2)e^{-\frac{K}{2}(\gamma+\delta)}\right) - \gamma\delta^{\frac{2L}{L+1}}Ke^{-\frac{K}{2}(\gamma+\delta)}\right]$$

$$= \frac{\gamma e^{-K\gamma}}{D\delta}\left[\gamma^{\frac{L-1}{L+1}}\left(2 + (K-2)e^{\frac{K}{2}(\gamma-\delta)}\right) - K\delta^{\frac{L-1}{L+1}}e^{\frac{K}{2}(\gamma-\delta)}\right].$$

This derivative is positive when

$$\gamma^{\frac{L-1}{L+1}}\left(2 + (K-2)e^{\frac{K}{2}(\gamma-\delta)}\right) > K\delta^{\frac{L-1}{L+1}}e^{\frac{K}{2}(\gamma-\delta)},$$

for which a sufficient condition is that

$$(K-2)\gamma^{\frac{L-1}{L+1}}e^{\frac{K}{2}(\gamma-\delta)} > K\delta^{\frac{L-1}{L+1}}e^{\frac{K}{2}(\gamma-\delta)},$$

which can be rewritten as

$$\frac{\gamma}{\delta} > \left(\frac{K}{K-2}\right)^{\frac{L+1}{L-1}}. \tag{51}$$

We see that if $\gamma/\delta$ is above the threshold given in Eq. (51), then $\frac{d}{dt}(\gamma/\delta)$ is positive, and so $\gamma/\delta$ will monotonically increase under gradient flow, and so remains above the threshold. Consequently if we initialize in the region specified by this threshold, then the normalized logit singular values $\hat{a}$ must tend monotonically towards the vector $\frac{2}{K}(1,0,1,...,0,1)$. This is since the normalized singular values, for even and odd index are given by

$$a_{\text{odd}} = \frac{1}{\frac{K}{2} + \left(\frac{K}{2} - 1\right)\frac{\delta}{\gamma}}, \quad a_{\text{even}} = \frac{1}{\frac{K}{2}\frac{\gamma}{\delta} + \left(\frac{K}{2} - 1\right)},$$

which clearly tend to $\frac{2}{K}$ and $0$ respectively. To note that this monotonic approach must guarantee we tend to the limit point, rather than stalling at some other structure, recall in Lemma D.1 it was shown that the direction must tend to a KKT point, meaning a critical direction under the dynamics at infinity. It is simple to observe that the same condition in Eq. (51) also guarantees that the only direction for which $\frac{d}{dt}(\hat{a}) = 0$ is the limit point, and so it is the only KKT point in this portion of the space and so must be to where the path converges.

This guarantees that the KL divergence tends to infinity since

$$D_{\text{KL}}\left(\frac{1}{K-1}1_{K-1} \,\|\, \hat{a}\right) = -\log(K-1) - \frac{1}{K-1}\sum_i \log(\hat{a}_i), \tag{52}$$

and clearly if any $\hat{a}_i$ tends to zero this diverges to infinity, which we have shown occurs for the described initialization.

### D.5. Proof of Theorem 4.4

Recall our dynamics are given by

$$\frac{da_i}{dt} = \frac{1}{D} b_i a_i^{\frac{2L}{L+1}}, \quad b_i = \sum_{j=1}^{K-1} \Psi_{ij} e^{-(\Psi a)_j}, \quad D = 1 + \sum_{j=1}^{K-1} e^{-(\Psi a)_j}.$$

Suppose that $\|a\|_1 = \sum_i a_i = \epsilon \ll 1$. Expanding the exponential gives

$$D = 1 + \sum_j [1 - (\Psi a)_j] + O(\epsilon^2) = K - K\epsilon + O(\epsilon^2), \tag{53}$$

where we used that $\sum_j \Psi_{ij} = K$, which is inherited from the properties of the Sylvester Hadamard matrix. Similarly

$$b_i = \sum_j \Psi_{ij}(1 - (\Psi a)_j) + O(\epsilon^2) = K - (\Psi^2 a)_i + O(\epsilon^2).$$

We then use that $\Psi^2 = K(I_{K-1} + 1_{K-1}1_{K-1}^T)$, so that

$$b_i = K - Ka_i - K\epsilon + O(\epsilon^2). \tag{54}$$

Inputting Eq. (53) and (54) into the expression of the derivative gives:

$$\frac{da_i}{dt} = a_i^{\frac{2L}{L+1}} [K - Ka_i - K\epsilon + O(\epsilon^2)][K(1 - \epsilon) + O(\epsilon^2)]^{-1}$$

$$= a_i^{\frac{2L}{L+1}} [1 - a_i - \epsilon + O(\epsilon^2)][1 + \epsilon + O(\epsilon^2)]$$

$$= a_i^{\frac{2L}{L+1}} (1 - a_i) + O\left(\epsilon^{2+\frac{2L}{L+1}}\right).$$

We now consider how the ratio of normalized singular values changes in this linearized regime. This is given by

$$\frac{d}{dt}\left(\frac{\hat{a}_i}{\hat{a}_j}\right) = \frac{d}{dt}\left(\frac{a_i}{a_j}\right) = \frac{a_i}{a_j}\left[a_i^{\frac{L-1}{L+1}}(1 - a_i) - a_j^{\frac{L-1}{L+1}}(1 - a_j) + O\left(\epsilon^{2+\frac{L-1}{L+1}}\right)\right]. \tag{55}$$

When $L = 1$, this reduces to

$$\frac{d}{dt}\left(\frac{\hat{a}_i}{\hat{a}_j}\right) = -\frac{a_i}{a_j \|a\|_1}[\hat{a}_i - \hat{a}_j + O(\epsilon)].$$

When $\epsilon$ small enough, this clearly has opposite sign to $\hat{a}_i - \hat{a}_j$. Consequently the ratio $\hat{a}_i/\hat{a}_j$ must monotonically approach 1 in the linearized regime, meaning any two modes that are different must approach each other. Clearly under such dynamics, the only stable structure is when $\hat{a}$ is the uniform vector, meaning DNC is the only stable structure under the linearized dynamics for $L = 1$.

When $L \geq 2$ and $\|a\|_1$ is small as in the linearized regime, we have

$$\hat{a}_i > \hat{a}_j \implies a_i^{\frac{L-1}{L+1}}(1 - a_i) > a_j^{\frac{L-1}{L+1}}(1 - a_j).$$

Consequently the sign of the derivative in Eq. (55) is the same as that of $\hat{a}_i - \hat{a}_j$ when $\epsilon$ is small enough. This implies that any two non-equal singular values must grow further apart. Thus, DNC is critical, but not stable, in the linearized regime for $L \geq 2$.

### D.6. Proof of Theorem 4.5

Recall our optimization problem, given in Eq. (1). It was shown in Eq. (38) that the gradient flow equations for the parameter matrices are given by (using the shorthand $W_0 := H_1$)

$$\frac{dW_l}{dt} = A_{l+1}^T(Y - P)H_l^T, \quad \text{for } l = 0, ..., L,$$

where

$$A_{l+1} = W_L...W_{l+1}, \quad \text{for } l = 0, ..., L-1, \quad A_{L+1} = I_K,$$

$$H_l = W_{l-1}...W_1 H_1, \quad \text{for } l = 1, ..., L, \quad H_0 = I_{Kn},$$

and $P$ is the softmax output of the network

$$P_{ij} = \frac{\exp(Z_{ij})}{\sum_u \exp(Z_{uj})}.$$

We can consider how the logit matrix $Z = W_L...W_1 H_1$ is updated directly

$$\frac{dZ}{dt} = \sum_{l=0}^{L} W_L...W_{l+1} \frac{dW_l}{dt} W_{l-1}...H_1$$

$$= \sum_{l=0}^{L} A_{l+1} A_{l+1}^T (Y - P) H_l^T H_l. \tag{56}$$

Now recall that we initialize our parameter matrices as $W_l(0) = \epsilon B_l$, $l = 0, ..., L$, where each $B_l$ has entries sampled i.i.d. from a Gaussian distribution with mean 0 and variance $1/d$. We now quote Lemma D.11, which states that we have the following convergence in probability as $d \to \infty$:

$$A_{l+1} A_{l+1}^T \to \epsilon^{2(L-l)} I_K, \quad H_l^T H_l \to \epsilon^{2l} I_{Kn}.$$

Consequently we have

$$\frac{dZ}{dt}(0) \to (L+1)\epsilon^{2L}(Y - P(0)).$$

Now we note that as $\epsilon \to 0$, we have

$$P_{ij} = \frac{\exp(Z_{ij})}{\sum_u \exp(Z_{uj})} \to \frac{1}{K},$$

and hence

$$P \to \frac{1}{K} 1_K 1_{Kn}^T,$$

so that

$$Y - P \to I_K \otimes 1_n^T - \frac{1}{K} 1_K 1_{Kn}^T = S \otimes 1_n^T.$$

Hence in these joint limits we have the following convergence in probability

$$\frac{dZ}{dt}(0) \to (L+1)\epsilon^{2L} S \otimes 1_n^T.$$

### D.7. Supporting Lemmas

**Lemma D.1.** *Consider GF on the deep UFM (Eq. 1) with network width $d \geq K$. If there exists $t_0$ such that $\mathcal{L}(t_0) < \log(2)$, then any limit point of the normalized parameters $(\hat{W}_L, ..., \hat{W}_1, \hat{H}_1)$ lies in the direction of a Karush-Kuhn-Tucker (KKT) point of the constrained optimization problem*

$$\min_{W_L,...,W_1,H_1} \|H_1\|_F^2 + \sum_{l=1}^{L} \|W_l\|_F^2$$

$$s.t. \ (z_{ic})_c - (z_{ic})_{c'} \geq 1, \quad c' \neq c = 1, ..., K, \quad i = 1, ..., n,$$

*where $z_{ic}$ are the columns in class order of the logit matrix $Z = W_L...W_1 H_1$.*

*Proof.* Note the arguments given in the proof of Theorem 3.2 of (Ji et al., 2022), though for the $L = 1$ UFM, can be applied directly to produce the constrained optimization problem above from the gradient flow dynamics on the deep UFM. Alternatively, the arguments in the proof of Theorem 4.4 of (Lyu & Li, 2019), as described in their main text and their App. G, can be applied to this model since the deep UFM is an example of a homogeneous network, from which the result follows as a corollary of their main theorem. $\square$

**Lemma D.2.** *Let $K > 2$, and $Z' \in \mathbb{R}^{K \times K}$ satisfy $Z'_{cc} - Z'_{c'c} \geq 1$ for $c, c' \neq c = 1, ..., K$. Then $\text{rank}(Z') \geq 2$.*

*Proof.* Clearly the condition does not hold if $\text{rank}(Z') = 0$. If $\text{rank}(Z') = 1$, then

$$Z' = uv^T, \quad \text{for } u, v \in \mathbb{R}^K.$$

Hence our condition becomes

$$u_c v_c - u_{c'} v_c \geq 1.$$

Clearly $v_c$ is always non-zero, else the condition is not satisfied. if $v_c$ is positive then

$$u_c \geq u_{c'} + \frac{1}{v_c} > u_{c'},$$

and so $u_c$ must be the unique largest element of the vector $u$. If $v_c$ is negative then

$$u_c < u_c - \frac{1}{v_{c'}} \leq u_{c'},$$

and so $u_c$ must be the unique smallest element of the vector $u$.

Hence each entry of $u$ must be either the largest or the smallest element of the vector in order for $Z'$ to satisfy the condition. However at most two entries can be either the unique largest or the unique smallest entry. Given that $K > 2$ this condition can hence never be satisfied, and so we must have $\text{rank}(Z') \geq 2$. $\qquad\square$

**Lemma D.3.** *Consider the following optimization problem for $K \geq 3$*

$$\max_{\theta_1, ..., \theta_K \in (0, 2\pi]} \left\{ \sum_{u=1}^K \cos(2\theta_u) \right\},$$

*subject to the condition that all angles are separated by at least $\delta \in (0, \frac{2\pi}{K}]$, including the wrap around value. The attained value of this objective when $K$ is even, or $K$ is odd and $\delta \in (0, \frac{2\pi}{K+1}]$, is given by*

$$\frac{\sin\left(\lfloor \frac{K}{2} \rfloor \delta\right) + \sin\left(\lceil \frac{K}{2} \rceil \delta\right)}{\sin(\delta)},$$

*whilst for odd $K$ and $\delta \in (\frac{2\pi}{K+1}, \frac{2\pi}{K})$, it is given by*

$$\frac{|\sin(K\delta)|}{\sin(\delta)}.$$

*Proof.* Note we can redefine the domain to be $(-\frac{\pi}{2}, \frac{3\pi}{2}]$ without changing the value of the maximum. Define the regions $C_0 = (-\frac{\pi}{2}, \frac{\pi}{2}]$ and $C_\pi = (\frac{\pi}{2}, \frac{3\pi}{2}]$. We begin by considering a configuration of $\theta_1, ..., \theta_K$ with $r$ points in $C_0$, denoted $x_1 < ... < x_r$, and $s = K - r$ points in $C_\pi$, denoted $y_1 < ... < y_s$.

Suppose that the configuration has a choice of $j$ such that $x_{j+1} - x_j > \delta$. Then clearly there exists some value $\epsilon$ such that we can move the furthest of $x_j, x_{j+1}$ by $\epsilon$ in the direction of zero without breaking feasibility. This must improve the objective, since $\cos(2\theta)$ increases as you approach 0 within the set $C_0$. Hence in order for such a configuration to be optimal, we must have $x_{j+1} - x_j = \delta$ for $j = 1, ..., r - 1$. The exact same reasoning holds for the $y$'s within the set $C_\pi$. Hence any optimal configuration must have

$$x_j = \alpha + \left(j - \frac{r+1}{2}\right)\delta, \qquad y_j = \pi + \beta + \left(j - \frac{s+1}{2}\right)\delta, \tag{57}$$

where we have written this so that the two clusters are centered at $\alpha, \pi + \beta$ respectively. We have ensured such a configuration has feasibility within $C_0$ and $C_\pi$. To ensure feasibility between the two subsets, we consider the edge points of $C_0, C_\pi$. The

leftmost point in $C_0$ is $\alpha - \frac{r-1}{2}\delta$ and the rightmost point in $C_\pi$ is $\beta + \frac{s-1}{2}\delta$. Demanding the difference, after adding $2\pi$ to get the wrap around value, is at least $\delta$ gives

$$\alpha - \beta \geq -\left(\pi - \frac{K}{2}\delta\right).$$

Doing the same on the other side gives

$$\alpha - \beta \leq \pi - \frac{K}{2}\delta,$$

and so the condition for feasibility at the edge points can be stated as

$$|\beta - \alpha| \leq \pi - \frac{K}{2}\delta.$$

Returning to the objective value, the contribution from the points in $C_0$ is given by

$$\sum_{j=1}^{r} \cos(2\alpha + 2j - (r+1)\delta) = \text{Re}\left(e^{2i\alpha - (r+1)i\delta} \sum_{j=1}^{r} e^{2ij\delta}\right)$$

$$= \text{Re}\left(e^{2i\alpha} e^{-(r+1)i\delta} \frac{e^{2i\delta}\left(e^{2ri\delta} - 1\right)}{e^{2i\delta} - 1}\right) = \text{Re}\left(e^{2i\alpha} \frac{\sin(r\delta)}{\sin(\delta)}\right).$$

The contribution from the points in $C_\pi$ can be calculated similarly, giving a total objective value of

$$\text{Re}\left(e^{2i\alpha}\frac{\sin(r\delta)}{\sin(\delta)} + e^{2i\beta}\frac{\sin(s\delta)}{\sin(\delta)}\right). \tag{58}$$

Note this is clearly upper bounded by the modulus, where a global rotation allows for attainment of this bound, so we can rewrite this as

$$= \left|\frac{\sin(r\delta)}{\sin(\delta)} + e^{2i(\beta - \alpha)}\frac{\sin(s\delta)}{\sin(\delta)}\right|.$$

Define the quantity $\nu = 2(\beta - \alpha)$. Clearly if both $\sin(r\delta)$ and $\sin(s\delta)$ are non-negative, then the optimal value is $\nu = 0$. We call this Case 1, and it has objective value

$$\frac{\sin(r\delta) + \sin(s\delta)}{\sin(\delta)}.$$

Note that at most one of $\sin(r\delta)$ and $\sin(s\delta)$ is negative, since we require $r\delta \geq \pi$ or $s\delta \geq \pi$ for this to happen, which since $\delta \geq \frac{2\pi}{K}$, requires as a necessary condition $r$ or $s$ is greater than $\frac{K}{2}$, which can only happen for at most one. We call the case where one of them is negative Case 2. In this case our objective value is

$$\frac{1}{\sin(\delta)}\left(\sin^2(r\delta) + \sin^2(s\delta) + 2\sin(r\delta)\sin(s\delta)\cos(\nu)\right)^{\frac{1}{2}}.$$

We must have $\delta > \frac{\pi}{K}$ for this to happen, since $r, s \leq K$, then from the feasibility condition we have $|\nu| \leq \pi$. Hence, the objective is maximized when $|\nu|$ is at its upper bound, and so $|\nu| = 2\pi - K\delta$. Then using standard trigonometric identities, this case reduces to an objective value of

$$\frac{|\sin(K\delta)|}{\sin(\delta)}.$$

We have now calculated the best objective value for fixed $r$ conditional on the sign of the sine terms, it remains to optimize $r$. First we optimize among Case 1:

$$\sin(r\delta) + \sin(s\delta) = 2\sin\left(\frac{K}{2}\delta\right)\cos\left(\frac{2r - K}{2}\delta\right).$$

Since we have $0 < \frac{K}{2}\delta \leq \pi$, the sin term is non-negative. We also have $-\pi \leq \frac{2r-K}{2}\delta \leq \pi$, and so the cosine term is maximized when $2r - K$ is closest to zero, meaning $r = \lfloor\frac{K}{2}\rfloor$. This solution: $r = \lfloor\frac{K}{2}\rfloor$, $s = \lceil\frac{K}{2}\rceil$ maintains non-negativity

of both $\sin(r\delta)$ and $\sin(s\delta)$ provided $\lceil \frac{K}{2} \rceil \delta \leq \pi$. Assuming this is the case, we show Case 2 cannot beat Case 1. The Case 2 objective value is

$$|\sin(K\delta)| = 2\sin\left(\frac{K}{2}\delta\right)\left|\cos\left(\frac{K\delta}{2}\right)\right|.$$

For even $K$ we have

$$2\sin\left(\frac{K}{2}\delta\right)\left|\cos\left(\frac{K\delta}{2}\right)\right| \leq 2\sin\left(\frac{K}{2}\delta\right),$$

and so Case 1 beats Case 2 for even $K$. For odd $K$ when $\lceil \frac{K}{2} \rceil \delta \leq \pi$ we have

$$2\sin\left(\frac{K}{2}\delta\right)\left|\cos\left(\frac{K\delta}{2}\right)\right| \leq 2\sin\left(\frac{K}{2}\delta\right)\cos\left(\frac{1}{2}\delta\right) = \sin\left(\left\lfloor\frac{K}{2}\right\rfloor\delta\right) + \sin\left(\left\lceil\frac{K}{2}\right\rceil\delta\right),$$

where we used that $0 \leq \frac{1}{2}\delta < \frac{K}{2}\delta \leq \lceil\frac{K}{2}\rceil\delta \leq \pi$. Hence for all cases where $r = \lfloor\frac{K}{2}\rfloor\delta$ leads to positive $\sin(r\delta)$ and $\sin(s\delta)$, we have that the best objective value is

$$\frac{1}{\sin(\delta)}\left(\sin\left(\left\lfloor\frac{K}{2}\right\rfloor\delta\right) + \sin\left(\left\lceil\frac{K}{2}\right\rceil\delta\right)\right).$$

The only case where we do not get both sine terms positive is when $\lceil\frac{K}{2}\rceil\delta > \pi$, which occurs when $K$ is odd and $\delta \in \left(\frac{2\pi}{K+1}, \frac{2\pi}{K}\right]$. In this case any split produces a negative sine term, and so the maximum must instead occur for Case 2, meaning we get the alternative optimal value of

$$\frac{1}{\sin(\delta)}|\sin(K\delta)|.$$

$\square$

**Lemma D.4 (From Theorem 1 of Garrod et al. (2025)).** *Let $K = 2^m$ for $m \in \mathbb{N}$. Suppose $U = \frac{1}{\sqrt{K}}\Phi$, where $\Phi$ is the $K \times K$ Sylvester Hadamard matrix. Denote the columns of $U$ by $u_i$, $i = 1, \ldots, K$. For a matrix of the form $Z = \sum_i a_i u_i u_i^T$, the corresponding softmax matrix is*

$$P(Z) = \sum_i \nu_i u_i u_i^T,$$

*where*

$$\nu_i = \frac{\sum_j \Phi_{ij} e^{\frac{1}{K}(\Phi a)_j}}{\sum_j e^{\frac{1}{K}(\Phi a)_j}}.$$

**Lemma D.5 (From Lemma 3.6 Horadam (2007)).** *Let $K = 2^m$ for $m \in \mathbb{N}$. The columns of the $K \times K$ Sylvester Hadamard matrix $\Phi$, denoted by $\phi_i$ for $i = 0, \ldots, K-1$, satisfy*

$$\phi_i \circ \phi_j = \phi_{i \oplus j}$$

*where $\circ$ denotes the Hadamard product, and $\oplus$ denotes the bitwise XOR.*

**Lemma D.6 (From Exercise 1.12 of O'Donnell (2014)).** *Let $K = 2^m$ for $m \in \mathbb{N}$. The entries of the $K \times K$ Sylvester Hadamard matrix $\Phi$, are given by*

$$\Phi_{ij} = (-1)^{i \cdot j}$$

*where $\cdot$ denotes the mod-2 dot product, acting on the binary representations of $i$ and $j$.*

*Proof.* Since O'Donnell (2014) does not feature a proof we include a proof by induction on $m \in \mathbb{N}$ here for completeness.

First for $m = 1$, and denoting the Sylvester Hadamard matrix by $\Phi_K$ to make clear the dimension, we have

$$(\Phi_2) = \begin{cases} -1 & \text{if } i = j = 1 \\ 1 & \text{otherwise} \end{cases} = (-1)^{i \cdot j}.$$

Hence, the result holds for $m = 1$. Now take it true for $m \in \mathbb{N}$, and consider the $m + 1$ case. We can write $i, j \in \mathbb{F}_2^{m+1}$ as $i = [i'; x]$, $j = [j'; y]$, where we have concatenated the extra index so that $i', j' \in \mathbb{F}_2^m$ and $x, y \in \mathbb{F}_2$. Now using the properties of the Hadamard matrix construction, we have:

$$(\Phi_{2^{m+1}})_{ij} = \begin{cases} -(\Phi_{2^m})_{i'j'} & \text{if } x = y = 1 \\ (\Phi_{2^m})_{i'j'} & \text{otherwise} \end{cases} = (\Phi_{2^m})_{i'j'}(-1)^{x \cdot y} = (-1)^{i' \cdot j'}(-1)^{x \cdot y} = (-1)^{i \cdot j},$$

Which gives the result.

$\square$

**Lemma D.7** (**From Corollary 3 of** Shang et al. (2020)). *Let $W_L \in \mathbb{R}^{K \times d}$, $W_{L-1}, ..., W_1 \in \mathbb{R}^{d \times d}$, $H_1 \in \mathbb{R}^{d \times Kn}$, where $d \geq K$. The minimization of the sum of their Frobenius norms squared, subject to a known product, is given by*

$$\min_{W_L, ..., W_1, H_1 \text{ such that } Z = W_L ... W_1 H_1} \left\{ \|H_1\|_F^2 + \sum_{l=1}^{L} \|W_l\|_F^2 \right\} = (L+1) \|Z\|_{S_{\frac{2}{L+1}}}^{\frac{2}{L+1}},$$

*where $\| \cdot \|_{S_p}^p$ denotes the Schatten-quasi norm with parameter $p$, given by*

$$\|M\|_{S_p}^p = \sum_{i=1}^{rank(M)} \sigma_i^p, \tag{59}$$

*where $\sigma_1, ..., \sigma_{rank(M)}$ are the non-zero singular values of the matrix $M$.*

**Lemma D.8.** *Let $K = 2^m$ for $m \in \mathbb{N}$, and let $a \in \mathbb{R}^{K-1}$ be of the form $a = [\gamma, \delta, ..., \gamma, \delta]$, so that $\gamma$ appears in $\frac{K}{2}$ entries and $\delta$ appears in $\frac{K}{2} - 1$ entries, with the two alternating, and we have $\gamma, \delta > 0$. Then*

$$(\Psi e^{-\Psi a})_i = \begin{cases} 2e^{-K\gamma} + (K-2)e^{-\frac{K}{2}(\gamma+\delta)}, & \text{if } i \text{ is odd}, \\ Ke^{-\frac{K}{2}(\gamma+\delta)}, & \text{if } i \text{ is even}. \end{cases}$$

*Proof.* begin by writing the vector $a$ as $a = \delta 1_{K-1} + (\gamma - \delta)c$, where the vector $c$ has 1 in entries with odd indices, and 0 for even indices. Since $c$ corresponds with the first column of $\Psi$, up to a factor of two, the properties of the underlying Hadamard matrix give that

$$\Psi c = \frac{K}{2} e_1 + \frac{K}{2} 1_{K-1},$$

where $e_1$ is the first basis vector in $\mathbb{R}^{K-1}$. Consequently

$$\Psi a = \frac{K}{2}(\gamma + \delta)1 + \frac{K}{2}(\gamma - \delta)e_1,$$

and hence

$$\exp(-(\Psi a)_1) = \underbrace{e^{-K\gamma}}_{\alpha}, \quad \exp(-(\Psi a)_i) = \underbrace{e^{-\frac{K}{2}(\gamma+\delta)}}_{\beta}, \quad i = 2, ..., K - 1,$$

where we have denote these quantities by $\alpha$ and $\beta$. This can then be written as

$$\exp(-\Psi a) = (\alpha - \beta)e_1 + \beta 1_{K-1}.$$

Again noting that the first column of $\Psi$ is $2c$, we have that $\Psi e_1 = 2c$, and so

$$\Psi \exp(-\Psi a) = 2(\alpha - \beta)c + K\beta 1_{K-1}.$$

Using the explicit forms of $\alpha$ and $\beta$ then gives the result. $\square$

**Lemma D.9.** *Let $K = 2^m$ for $m \in \mathbb{N}$. Denote by the vector $c \in \mathbb{R}^{K-1}$ the vector that takes value one in its odd indices, and zero in its even indices. if $a \in \mathbb{R}^{K-1}$ can be written in the form*

$$a = \mu c + \mu \epsilon, \quad \mu = \frac{2}{K} \|a\|_1 \in \mathbb{R}, \quad \epsilon \in \mathbb{R}^{K-1}, \quad \sum_i \epsilon_i = 0,$$

*then to linear order in $\|\epsilon\|_1$, we have*

$$\Psi \exp(-\Psi a) = K e^{\frac{-K\mu}{2}} 1_{K-1} - K e^{-\frac{K\mu}{2}} \mu \epsilon + 2 \left( e^{-K\mu} - e^{-\frac{K\mu}{2}} \right) (1 - \mu(\Psi\epsilon)_1) c + O\left(\|\epsilon\|_1^2\right).$$

*Proof.* Using that $\Psi c = \frac{K}{2}(e_1 + 1_{K-1})$, where $e_1$ is the first basis vector in $\mathbb{R}^{K-1}$, we have

$$\Psi a = \frac{K\mu}{2}(e_1 + 1_{K-1}) + \mu(\Psi\epsilon).$$

Hence

$$(\exp(-\Psi a))_i = \begin{cases} e^{-K\mu} e^{-\mu(\Psi\epsilon)_1}, & \text{if } i = 1, \\ e^{-\frac{K}{2}\mu} e^{-\mu(\Psi\epsilon)_i}, & \text{otherwise.} \end{cases},$$

which we can write as

$$(\exp(-\Psi a))_i = \left[ e^{-\frac{K}{2}\mu} 1_{K-1} + (e^{-K\mu} - e^{-\frac{K}{2}\mu}) e_1 \right]_i e^{-\mu(\Psi\epsilon)_i}.$$

Now expand the exponential to linear order in $\|\epsilon\|_1$

$$= \left[ e^{-\frac{K}{2}\mu} 1_{K-1} + (e^{-K\mu} - e^{-\frac{K}{2}\mu}) e_1 \right]_i - e^{-\frac{K}{2}\mu} \mu(\Psi\epsilon)_i - (e^{-K\mu} - e^{-\frac{K}{2}\mu}) \mu(e_1)_i (\Psi\epsilon)_i + O(\|\epsilon\|_1^2).$$

Next use that $(e_1)_i (\Psi\epsilon)_i = (\Psi\epsilon)_1 (e_1)_i$, and group the $e_1$ terms, giving

$$\exp(-\Psi a) = e^{-\frac{K}{2}\mu} 1_{K-1} - e^{-\frac{K}{2}\mu} \mu(\Psi\epsilon) + (e^{-K\mu} - e^{-\frac{K}{2}\mu})(1 - \mu(\Psi\epsilon)_1) e_1 + O(\|\epsilon\|_1^2).$$

Then using that $\Psi 1_{K-1} = K 1_{K-1}$, $\Psi e_1 = 2c$ and $\Psi^2 \epsilon = K\epsilon$, gives that

$$\Psi \exp(-\Psi a) = K e^{-\frac{K}{2}\mu} 1_{K-1} - K e^{-\frac{K}{2}\mu} \mu \epsilon + 2(e^{-K\mu} - e^{-\frac{K}{2}\mu})(1 - \mu(\Psi\epsilon)_1) c + O(\|\epsilon\|_1^2).$$

$\square$

**Lemma D.10.** *Let $X \in \mathbb{R}^{d \times v}$ be a random matrix independent of $W \in \mathbb{R}^{d \times d}$, where the entries of $W$ are i.i.d. $N(0, 1/d)$. If we have the following convergence in probability in the $d \to \infty$ limit, where $v$ is fixed*

$$X^T X \to I_v,$$

*then we also have*

$$X^T W^T W X \to I_v.$$

*Proof.* Note it is simple to show that $\mathbb{E}[W^T W] = I_d$, and from this we clearly have

$$\mathbb{E}[X^T W^T W X | X] = X^T X.$$

We can now calculate the conditional variances. Write the columns of $X$ as $x_1, ..., x_m$, then

$$(X^T W^T W X)_{ij} = (W x_i)^T (W x_j) = \sum_{u=1}^{d} (w_u^T x_i)(w_u^T x_j),$$

where we denote the rows of $W$ by $w_u^T$.

Conditional on $X$, the rows $w_r$ are independent Gaussian vectors, so the summands are independent. Hence:

$$\text{Var}[(X^T W^T W X)_{ij} | X] = d \, \text{Var}[(w_1^T x_i)(w_1^T x_j) | X].$$

Denote the variables $A = w_1^T x_i, B = w_1^T x_j$. Now using standard properties of centered Gaussians we have

$$\text{Var}(A) = \frac{1}{d} x_i^T x_i, \quad \text{Var}(B) = \frac{1}{d} x_j^T x_j, \quad \text{Cov}(A, B) = \frac{1}{d} x_i^T x_j,$$

along with the fact that for centered jointly Gaussian variables we have

$$\text{Var}(UV) = \text{Var}(U)\text{Var}(V) + \text{Cov}(U, V)^2,$$

gives

$$\text{Var}((X^T W^T W X)_{ij} | X) = \frac{1}{d} [(X^T X)_{ii} (X^T X)_{jj} + (X^T X)_{ij}^2].$$

Then since $v$ is fixed, and $X^T X \to I$ in probability, we have that the conditional variance converges to zero in probability as $d \to \infty$. Chebyshev's inequality applied entrywise then gives

$$X^T W^T W X - X^T X \to 0,$$

and therefore

$$X^T W^T W X \to I.$$

$\square$

**Lemma D.11.** *Let $K, n, L \in \mathbb{N}$ and let the matrices $W_L \in \mathbb{R}^{K \times d}$, $W_l \in \mathbb{R}^{d \times d}$, $W_0 \in \mathbb{R}^{d \times Kn}$ be initialized with entries sampled i.i.d. from a Gaussian with mean 0 and variance $1/d$. Define the matrices*

$$A_{l+1} = W_L...W_{l+1}, \quad l = 0, ..., L - 1, \quad H_l = W_{l-1}...W_1 H_1, \quad l = 1, ..., L.$$

*Then we have the following convergence in probability in the $d \to \infty$ limit, with $K, n, L$ fixed*

$$A_{l+1} A_{l+1}^T \to I_K, \quad H_l^T H_l \to I_{Kn}.$$

*Proof.* We shall show for $H_l$, the proof for $A_{l+1}$ is almost identical. We apply Lemma D.10 iteratively. It is simple to show $H_1 \in \mathbb{R}^{d \times Kn}$ satisfies $H_1^T H_1 \to I$ in probability. Then applying Lemma D.10 with $v = Kn$ and first with $X = H_1, W = W_1$, then $X = W_1 H_1, W = W_2$, so on, gives

$$H_l^T H_l = (W_{l-1}...W_1 H_1)^T (W_{l-1}...W_1 H_1) \to I_{Kn},$$

for $l = 1, ..., L$. $\square$

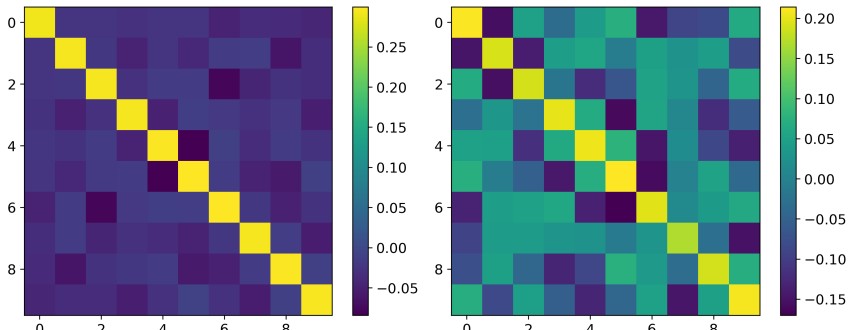

*Figure 7.* Demonstrations of different normalized mean logit matrices after training for $L = 3$, $K = 10$. **Left:** logits that are approximately DNC, with effective rank $8.9 \approx K - 1$. **Right:** logits of a low rank alternative, with effective rank 4.3.

## E. Further Numerical Experiments

### E.1. Main Text Experimental Details

In this section we provide further details for the plots of the main text.

Figure 1 is produced by computing the matrix $\tilde{Z}$ defined in Eq. (4) for hyperparameters $K = 4$, $L = 3$, and $n = 1$, for both the DNC solution (Definition 2.1) and a canonical low-rank solution (Eq. (14)). The angles are computed by taking dot products between columns, and the logit distances by taking the difference between diagonal and off-diagonal elements. In the low-rank case there are multiple values for the angular and logit distances; we consider only the smallest, since this is the one that determines the loss at large norm. These quantities are then plotted in the same hyperplane to enable a direct comparison of distances.

Figure 2 is produced by running deep UFM simulations from random initialization with $K = 10$, $n = 5$, $d = 100$, and a range of network depths $L$. We report the effective rank, a quantity that aims to approximate the hard rank in settings where singular values are not identically zero. The effective rank is defined as

$$R_{\text{eff}}(M) = \exp\left(-\sum_{i=1}^{\text{rank}(M)} p_i \log(p_i)\right), \quad \text{where } p_i = \frac{\sigma_i}{\sum_{j=1}^{\text{rank}(M)} \sigma_j}, \tag{60}$$

and $\sigma_1, ..., \sigma_{\text{rank}(M)}$ are the non-zero singular values of the matrix $M$.

For each value of $L$ we conduct five runs, and we plot the mean effective rank with one-standard-deviation error bars.

The first two plots of Figure 3 show experiments conducted for a range of class numbers $K$, with network depth $L = 10$, network width $d = K$, $n = 5$, and learning rate $0.08$. To aid convergence, the networks were initially trained with a small amount of regularization, $\lambda = 10^{-3}$, for $2 \times 10^5$ epochs, followed by a further $1 \times 10^5$ epochs with no regularization. The Gram factor $X$ is obtained using the SVD of the mean logit matrix. For the robustness experiment, we trained five models for which rank 2 was observed. Since $Z$ was symmetric positive semidefinite of rank 2, $X$ was computed via its singular value decomposition (note that $X$ is unique up to rotation). We then computed the angles between adjacent feature means and, across the five runs, reported the maximum and minimum angles observed in each case.

The final plot in Figure 3 shows the normalized margins for a range of class numbers and ranks, for $L = 4$ and $n = 5$, overlaid with the same quantity for $L = 1$ bottlenecked UFMs. To obtain a range of ranks, we used a variety of network widths and a new random seed for each run. Due to the non-benign loss landscape, different ranks can arise even for the same network width. Results were logged throughout training, and when the same rank appeared across multiple runs we report the average. To produce the bottlenecked cases, we trained an $L = 1$ UFM with network width $r = 2, \ldots, K - 1$ for each $K$. In both sets of experiments we used a small amount of regularization, $\lambda = 10^{-3}$, for the first $2 \times 10^5$ epochs, followed by unregularized training for $1 \times 10^5$ epochs. The normalized margins are the smallest gaps between the correct normalized logit value and any other logit value for the same datapoint, taken over the entire dataset.

Figure 4 is produced by evolving a vector $a$ under the Hadamard dynamics of Eq. (6) for $K = 16$ and $L = 2$. The vector is initialized with entries sampled uniformly from $[0, 1]$, and then renormalized to have $L_1$ norm $10^{-3}$. We show a single

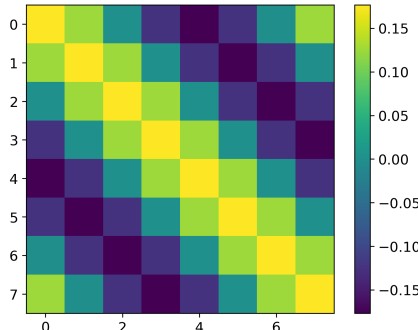

*Figure 8.* Plot of the normalized mean logit matrix at convergence of a rank two solution for $K = 8, L = 10$.

training run for simplicity, but the dynamics exhibit the same phenomenology under this setup across a large number of random initializations.

Figure 5 shows the impact of width on the early dynamics of $Z$. For the first plot, the network parameters are as described in Theorem 4.5, with $\epsilon = 0.01$, $K = 10$, $L = 3$, $n = 5$, and a range of widths $d$. The logit derivative is computed using Eq. (56). The reported metric is the normalized squared Frobenius distance,

$$M = \|\hat{Z} - \hat{S}\|_F^2. \tag{61}$$

For the second plot, we use the same parameters and metric, but evaluate the logit matrix after the logit norm has increased by a factor of ten. For the final plot, we report the effective rank of the mean logit matrix for a range of network widths with $L = 3$ and $K = 10$, after training for $2 \times 10^5$ epochs. In all experiments, we conduct five runs and report the mean with one-standard-deviation error bars.

Figure 6 shows the outcome of training a ResNet-20 with a ReLU head on the MNIST and CIFAR-10 datasets from random initialization. For the MNIST experiments, we subsample 5,000 examples per class to match the class balance of CIFAR-10. MNIST is trained for 4,000 epochs and CIFAR-10 for 5,000 epochs. In all cases, the network attains $\approx 100\%$ training accuracy early in training, so learning primarily occurs in the terminal phase (Papyan et al., 2020). The input data are preprocessed by subtracting the mean and dividing by the standard deviation. We use batch gradient descent with batch size 10,000 to approximate full-batch gradient descent, as assumed by the model. No explicit regularization is used in any experiment. In the first plot, we set $d = 50$ and vary the number of layers in the ReLU head between 1 and 8. In the second plot, we fix the number of layers in the ReLU head to $L = 3$ and vary the head width between 50 and 200. For both plots, we train five models at each parameter setting and report the mean with one-standard-deviation error bars. The final plot shows generalization performance versus the effective rank of the network across the forty five MNIST models from the previous two plots. We also include the line of best fit to illustrate the overall trend.

### E.2. Further Model Experiments

In Figure 7 we show examples of normalized mean logit matrices recovered during training of a deep UFM with $K = 10$, $L = 3$, $n = 5$, and network width $d = 100$. Even though the DNC solution clearly has a larger normalized margin over the data than the low-rank alternative, the network often converges to low-rank solutions such as the one shown.

In Figure 8 we show the result of training a deep UFM with $K = 8$, $L = 10$, $n = 5$, and $d = 8$. The classes were permuted to produce the symmetric structure shown. This corresponds to the uniformly separated circle solution described in Figure 3 of the main text.

### E.3. Further Hadamard Experiments

The left panel of Figure 9 demonstrates how the effective rank changes with depth in the Hadamard setting. In Appendix C.1 we show that, in this Hadamard setting, the minimal rank the logit matrix can achieve while remaining compatible with a continuously decreasing loss along gradient flow is $m = \log_2(K)$. The left panel shows that the low-rank bias becomes stronger as $L$ increases. Specifically, after long training times the effective rank decreases monotonically with depth: it

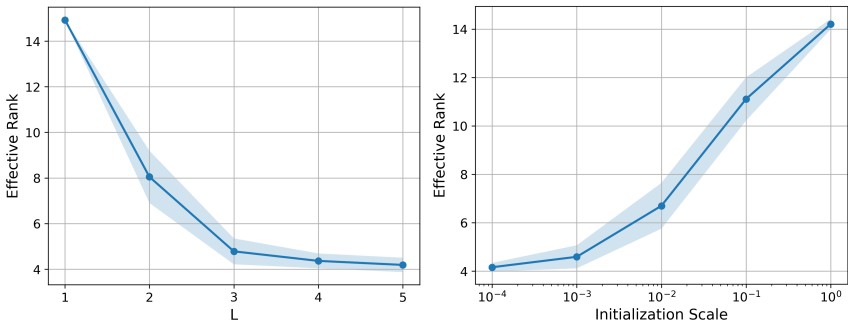

*Figure 9.* **Left:** Empirical mean of the effective rank of the mean logit matrix at $t = 10^3$ under the dynamics of Equation (6) with parameter $K = 16$ for various network depths $L$. The vector $a \in \mathbb{R}^{K-1}$ is initialized with entries uniformly sampled from $[0.5, 1]$, then normalized to have $L_1$ norm of $10^{-3}$. For each $L$ the average empirical effective rank over 10 runs is reported, along with one-standard-deviation error bars. **Right:** consideration of various initialization scales at $L = 3$. Same initialization set up, similarly average and one-standard-deviation error bars over 10 runs is reported.

starts at $K - 1 = 15$, corresponding to NC when $L = 1$, and eventually concentrates at the lower bound $\log_2(K) = 4$ for large $L$. This provides further evidence that increasing depth induces a low-rank bias in our model.

The description of the "rich-get-richer" effect in the linearized regime naturally raises the question of how the initialization scale might affect the rank of the converged solution. Since a larger initialization scale implies less time spent in the linearized regime, it may lead to a higher rank at convergence. This is explored in the right panel of Figure 9, where, as the theory predicts, the effective rank at convergence increases as the initialization scale is increased in the Hadamard regime.

Here we also provide an example of a gradient-flow trajectory along which the KL divergence provably diverges. This corresponds to the initialization considered in the proof of Theorem 4.3 in App. D.4. In Figure 10, we show the evolution of the normalized logit singular values from the initialization $a = (\gamma, \delta, \ldots, \delta, \gamma)$, with $\gamma = 0.2$ and $\delta = 0.1$. Note that DNC corresponds to the case $\gamma/\delta \to 1$, whereas the cross-polytope geometry—an alternative stable point of the loss landscape—corresponds to $\gamma/\delta \to \infty$.

In App. D.4, we show that if $\gamma/\delta$ satisfies

$$\frac{\gamma}{\delta} > \left( \frac{K}{K-2} \right)^{\frac{L+1}{L-1}},$$

then the ratio must continue to increase for all future times. The specific choice in the figure satisfies this inequality at initialization, and it is clear that the evolution tends toward the cross-polytope geometry rather than DNC. Note that the KL divergence increases monotonically throughout this trajectory. In fact, it must diverge in the limit $t \to \infty$, but it does so slowly due to the lazy dynamics that emerge at large norm (Garrod et al., 2025), as well as the fact that the KL divergence involves the logarithm of the singular values.

### E.4. Hadamard Initialization is Representative of Random Initialization

In Section 4, we used Hadamard initialization to study the finite-time dynamics of the deep UFM. Although this structured initialization yields analytically tractable equations, the resulting phenomenology is also representative of small random initialization. The evolution of the singular values under small random initialization is shown in Figure 11, which may be contrasted with the Hadamard case in Figure 4 of the main text. We present one illustrative example, although all experiments of this kind exhibited the same qualitative behavior. The two cases display remarkably similar phenomenology; most notably, both exhibit the "rich-get-richer" effect, which explains how low-rank bias emerges at the level of individual training trajectories.

This is a standard observation in the spectral initialization literature, where structured initializations often trace representative trajectories for a much broader class of initial conditions. This phenomenon was first studied by Saxe et al. (2013; 2019) and Gidel et al. (2019) in the MSE-loss setting, and later extended to the CE-loss setting by Garrod et al. (2025). Across these works, the dynamics consistently exhibit a two-stage evolution: in the initial phase, the singular vectors rapidly align with the spectral basis; after this transient phase, the singular value trajectories closely follow those obtained under spectral initialization. Our results provide evidence of the same observation. Early in training, while the singular values remain

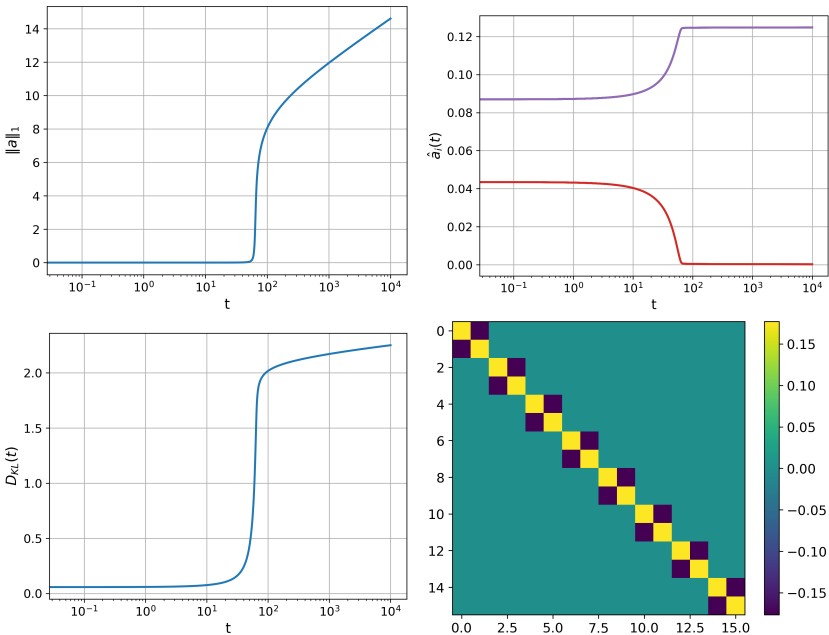

*Figure 10.* Evolution dynamics under Eq. (6) from the initialization described in Eq. (49) with $\gamma = 0.2$, $\delta = 0.1$ for parameters $L = 2, K = 16$. **Top Left**: Evolution of the logit norm $\|a\|_1 = \sum_i a_i$. **Top Right**: Evolution of the normalized singular values $\hat{a}_i$. **Bottom Left**: Evolution of the KL-divergence $D_{\mathrm{KL}}\left(\frac{1}{K-1}1_{K-1} \,\|\, \hat{a}\right)$, defined in Eq. (52). **Bottom Right**: mean logit matrix at the end of training, normalized to have unit Frobenius norm.

approximately stable, the singular vectors continue to evolve; once this stage has passed, the dynamics closely resemble those under Hadamard initialization, exhibiting the key phenomenology responsible for low-rank bias in the model.

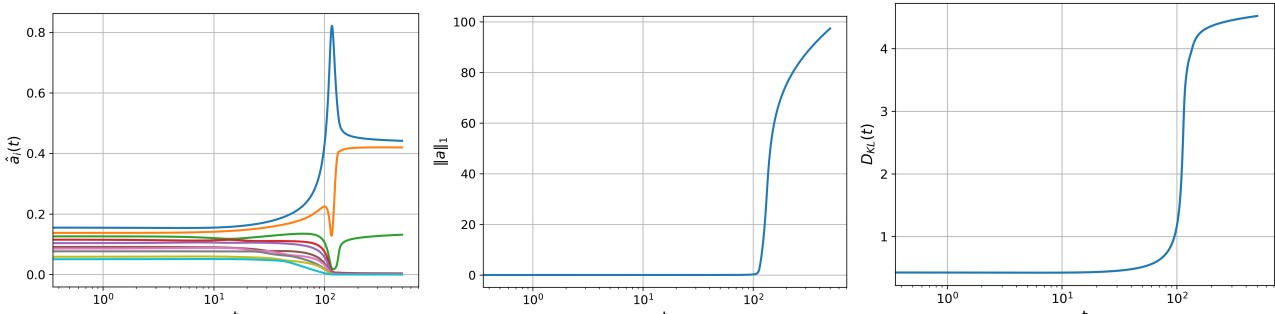

*Figure 11.* Experiments under Random initialization for the deep UFM. **Left:** Evolution of the normalized mean logit singular values $\hat{a}_i$ under gradient flow on the loss function of Eq. (1) for depth $L = 3$, width $d = 100$, $K = 10$ classes and $n = 5$ data points per class. Network was initialized with Gaussian normal entries of standard deviation 0.03. Note convergence to low-rank solution where many modes approach zero as also shown in the Hadamard case in Figure 4. **Middle:** Corresponding evolution of the mean logit nuclear norm. By the time norm increases and system exits the linearized regime, the low-rank structure has already taken hold, just as in the Hadamard case. **Right**: The corresponding KL divergence(definition in Eq. (52).).

