# OpenReview forum: "The Implicit Bias of Depth: From Neural Collapse to Softmax Codes"
_ICML.cc/2026/Conference — ICML 2026 regular_

### Official Review · Reviewer_3Hc2 · 2026-03-03

**Soundness:** 3
**Presentation:** 3
**Significance:** 3
**Originality:** 3
**Overall Recommendation:** 4
**Confidence:** 2

**Summary:**

The paper studies neural collapse (NC) in deep networks and shows that depth alone induces an implicit low-rank bias, creating alternative geometric structures that can outperform NC in terms of margin. While NC is the unique optimum in shallow networks, in deep networks it is not guaranteed. The authors rigorously analyze both the asymptotic landscape and the finite-time dynamics using the deep unconstrained features model, proving that NC is only locally stable after a certain norm threshold. Empirical experiments on MNIST and CIFAR-10 confirm the theory.

**Compliance With Llm Reviewing Policy:**

Affirmed.

**Ethical Review Concerns:**

I maintain a positive evaluation of this work. The authors have addressed my questions.

**Final Justification:**

I maintain my positive evaluation of this work because of its theoretical insights.

**Key Questions For Authors:**

1) How sensitive are your finite-time dynamics results to standard initialization schemes, and do you expect NC to remain unstable under typical settings?

2) Your analysis focuses on linear or simple ReLU heads, do you have any experiments on more complex architectures, such as ResNets with skip connections or networks with batch normalization?

**Limitations:**

yes

**Strengths And Weaknesses:**

$\textbf{Strengths}$

This paper provides rigorous and comprehensive analysis of implicit bias in deep networks, moving beyond shallow models to uncover how depth induces low-rank structures that compete with neural collapse. The deep UFM isolates depth-induced effects, the Hadamard-based framework clarifies finite-time dynamics, and experiments on MNIST and CIFAR-10 confirm that these insights hold in practical networks. The work also makes connections to generalization, initialization, and architectural factors and offers an explanation for when and why neural collapse emerges.


$\textbf{Weaknesses}$

While the paper is strong theoretically, some aspects are limited in practical generality. The Hadamard initialization, while mathematically convenient, is not representative of standard network initializations, so the instability of neural collapse observed in the analysis may be less pronounced in typical training. The focus on fully linear or simple ReLU heads leaves out more complex architectures, including skip connections, batch normalization, and nonlinear embeddings, which are common in modern networks. Additionally, the infinite-depth results, though elegant, may not fully capture the behavior of networks with realistic depth, leaving some uncertainty about how directly the theory maps to real-world training scenarios.

---

> ### Author Rebuttal · Authors · 2026-03-31
>
> We thank the reviewer for their thoughtful feedback. Below, we address each of the questions raised.
>
> **Hadamard initialization…not representative of standard network initializations…How sensitive are your finite-time dynamics results to standard initialization schemes. do you expect NC to remain unstable under typical settings?**
>
> We appreciate the opportunity to clarify. The reviewer is correct that Hadamard initialization differs from standard random schemes. However, we want to highlight two points in its defense.
>
> First, any gradient-flow trajectory, regardless of initialization, reveals structure in the loss landscape. The fact that non-DNC solutions are reachable from Hadamard initialization proves they exist as stable configurations, which is informative about our landscape independent of the initialization used to find them.
>
> Second, the finite-time phenomena we observe under Hadamard initialization also appear for small random initialization. We ran several small random initializations for K=10 and tracked the normalized singular values for comparison to Fig 4. We show one characteristic example here https://ibb.co/LDxGhcQz, but all runs displayed the same phenomenology: The Hadamard and random plots look similar, both showing the ‘rich get richer’ dynamic. In the revision, we will add several such plots to evidence that Hadamard initialization captures behaviour characteristic of random initialization. Based on this observation, we expect NC to remain unstable close to the origin under random initialization.
>
> More generally, numerical validation of similar phenomenology for spectral and small random initialization appears in Saxe et al. (2013) and Garrod et al. (2025). We will add this context to further justify Hadamard initialization as a modeling tool.
>
> **focus on linear or simple ReLU heads leaves out more complex architectures …. experiments on complex architectures, such as ResNets with skip connections or networks with batch normalization?**
>
> This is an important point. Our theoretical analysis uses assumptions for tractability, so empirical validation is essential. Accordingly, we provide experiments in Sec 5. We also note that prior work (25+ papers, see App. A) empirically validates UFM-based explanations across many deep learning phenomena, suggesting that the model captures aspects of modern network behavior.
>
> We have also realized that our theory can be extended to include residual connections. In linear models, residual connections can be interpreted as a change in initialization, since $h_{l+1} = h_l + W_{l+1} \sigma(h_l)$ reduces, under linear activation to $h_{l+1}=(I+W_l) h_l.$ Thus the residual connection can be absorbed into the weight initialization. One can then show that, from small initialization, the network evolves as though it were an $L=1$ UFM, and is therefore driven towards NC, countering the low-rank bias. We will  include this analysis in the revision so that residual connections are covered in our theory.
>
> On the empirical side, the feature map used in our real-data experiments is the ResNet-20, which includes both skip connections and batch norm. The context therefore where the network uses these architectural features at each layer actually corresponds to the L=1 datapoint on our graphs. We did not include this since we model L>1 in the theory, but will add this datapoint to the graphs for the revision. On MNIST for L=1, d=50, the effective rank averaged over five runs was 8.88, and the average test accuracy was 0.9877. This is consistent with our interpretation that skip connections promote the emergence of NC and improve generalization. We agree that a broader empirical study of these architectures would be valuable future work.
>
> **The infinite-depth results … may not capture the behavior of networks with realistic depth, leaving uncertainty about how the theory maps to real-world training scenarios**
>
> We clarify that the only infinite-depth result is Thm 3.2. We agree that, taken in isolation, this result may not directly apply to finite-depth settings. To address this, the experiment in Fig 3 shows that the d=2 softmax code already appears at depths as small as L=4.
>
> In fact, the main utility of Thm 3.2’s is to identify the d=2 softmax code. Once identified, its objective value can be computed analytically and compared with alternative structures. In particular, one can show that the d=2 softmax code can outperform NC on the max-margin problem in Eq. (3) for small L, for example K=4,5,6 and L=3. We will add this discussion in the revision to explain the finite-depth relevance of the result.
>
> That said, we agree that some ambiguity remains in how the theory maps to real networks. To address this, we will add a limitations section clarifying the boundaries of our modeling framework.
>
> Thank you again for your positive assessment and detailed feedback, which has helped us identify several places where the manuscript can be strengthened and clarified.

---

> > ### Author Rebuttal · Reviewer_3Hc2 · 2026-04-02
> >
> > Thank you for your time and effort, you have answered all my questions. I will keep my score.

---

### Official Review · Reviewer_hANK · 2026-03-12

**Soundness:** 3
**Presentation:** 3
**Significance:** 3
**Originality:** 4
**Overall Recommendation:** 5
**Confidence:** 4

**Summary:**

The paper studies the setting of learning a deep unconstrained feature model (followed by a deep linear network) with cross-entropy loss for a classification task without explicit ​$\ell_2$-regularization. The paper shows that, in contrast to the similar setting with only a single linear layer, network depth induces an implicit low-rank bias that favors a low-rank structure different from the Neural Collapse (NC) geometry. They further show that when the depth diverges to infinity, the logits converge to the softmax code structure. The analysis of the gradient-flow dynamics also explains why this low-rank structure emerges instead of the NC geometry.

**Compliance With Llm Reviewing Policy:**

Affirmed.

**Final Justification:**

I maintain a positive evaluation of this work because of its theoretical insights, which provide a clear picture of the deep UFM framework, from the loss landscape to the gradient flow dynamics.

**Key Questions For Authors:**

I would like to better understand several points in Section 3.2:

- "Crucially, $|| \tilde Z ||$ contributes to the overall scale of $Z$, so maximizing $|| \tilde Z ||$ is advantageous for reducing the loss": Could you please clarify this point?

- I understand the argument that the norm of $\tilde Z$ of the low-rank structure becomes smaller due to the effect of depth, but why do the authors say that "low-rank structures “propagate norm” more efficiently through repeated matrix multiplication"? I would like to better understand the point being made here.

- Discrimination versus confidence: "Since diverging logit norms drive softmax outputs toward one-hot vectors, larger $|| \tilde Z ||$ can be interpreted as higher confidence. Maximal angular separation, by contrast, promotes discrimination between classes."
However, if the features are highly separated, wouldn't the logits also move closer to a one-hot vector and thus be interpreted as high confidence as well?

**Limitations:**

yes

**Strengths And Weaknesses:**

**Strengths:**
- The paper investigates a well-defined theoretical problem that provides insight into the geometry of neural networks after training.
- The paper provides a clear picture (from loss landscape to gradient flow dynamic) for deep UFM framework of why deep neural collapse (DNC) is not the optimal structure, which covers comprehensively the properties of low-rank bias structure vs. DNC compared with previous works.
- I find that the convergence of softmax code of rank 2 in the limit of depth without the width constraint is interesting and novel.
- The writing is good, and related works are well discussed.

**Weaknesses:**
- Since UFM is still a framework based on strong assumptions, additional experiments demonstrating convergence to the analyzed geometry on real datasets and models, if possible, would be highly appreciated. I understand that this may be challenging, so this is not a major concern. For example, a version of Figure 3 evaluated on real data and networks would be helpful.

---

> ### Author Rebuttal · Authors · 2026-03-31
>
> We thank the reviewer for their thoughtful and constructive comments. We appreciate the care they took in reviewing our submission, and we respond to each weakness and question below.
>
> **experiments demonstrating convergence to the analyzed geometry on real datasets and models, if possible, would be highly appreciated. I understand that this may be challenging, so this is not a major concern. For example, a version of Figure 3 evaluated on real data and networks.**
>
> This is an excellent suggestion, and we agree that it would strengthen the paper. In particular, recovering analogues of the left and right panels of Figure 3 in real networks would be especially compelling. Due to time constraints, and given the number of models required for the right panel of Figure 3, we will not be able to produce these results before the discussion period ends. We will, however, prioritize this direction in the revised version of the paper.
>
> **"Crucially, $||\tilde{Z}||$ contributes to the overall scale of $Z$, so maximizing $||\tilde{Z}||$ is advantageous for reducing the loss": Could you please clarify this point?**
>
> When positive margins are achieved, the cross-entropy loss decreases monotonically with the norm of the logit matrix, since increasing the logit norm widens the margins in the logit matrix. It is then clear from Eq. (4) that increase $\|\tilde{Z}\|$ increases the norm of $Z$. Hence, the larger $||\tilde{Z}||$, the better the performance on the objective, and so the network is incentivised to maximize $||\tilde{Z}||$. To improve the clarity of the manuscript, we will provide this expanded explanation in a revision.
>
> **I understand the argument that the norm of $\tilde{Z}$ of the low-rank structure becomes smaller due to the effect of depth, but why do the authors say that "low-rank structures “propagate norm” more efficiently through repeated matrix multiplication"? I would like to better understand the point being made here.**
>
> We wish to emphasize that this is a basic linear-algebraic effect rather than something specific to neural networks. To help visualize this, consider two diagonal matrices with frobenius norm 1
>
> $M_2=$ diag$(1/\sqrt{2},1/\sqrt{2},0,...,0), $
>
> $M_1 =$ diag$(1,0,...,0).$
>
> Here $M_2$ has rank 2 and $M_1$ has rank 1. Raising to the $(L+1)$-th power gives
>
> $||M_1^{L+1}||_F = ||M_1||_F=1,$
>
> while
>
> $||M_2^{L+1}||_F = ||2^{-L/2} M_2||_F = 2^{-L/2},$
>
> which decays exponentially in $L$. The rank-1 matrix $M_1$ preserves its norm perfectly because all of its norm is concentrated in a single direction, whereas $M_2$ spreads its norm across two directions, each with a singular value below 1, causing exponential decay under repeated multiplication. This is what we mean by "low-rank structures propagate norm more efficiently": fewer directions means larger individual singular values for a given Frobenius norm, and hence less decay under matrix powers. In the  setting studied in the paper, the same intuition applies through the product of singular values across layers (Section 3.2). We agree this point was too compressed in the current draft and will clarify it in the revision.
>
> **Discrimination versus confidence: "Since diverging logit norms drive softmax outputs toward one-hot vectors, larger $||\tilde{Z}||$ can be interpreted as higher confidence. Maximal angular separation, by contrast, promotes discrimination between classes." However, if the features are highly separated, wouldn't the logits also move closer to a one-hot vector and thus be interpreted as high confidence as well?**
>
> This is a good point and we will revise the wording. The intended meaning was not that only the norm can affect confidence. Rather, once the normalized logits are fixed, the only role of the norm is to scale the logit gaps, thereby pushing the softmax output closer to a one-hot vector and increasing confidence. By contrast, the normalized logits determine how datapoints are positioned relative to classes, and therefore encode the discriminative structure itself. As the reviewer correctly notes, changes in the normalized logits can also increase confidence; our point was simply that all discriminative structure is encoded in $\tilde{Z}$, while the norm only controls confidence for a given discriminative geometry. Low-rank solutions exploit this by sacrificing quality in $\tilde{Z}$ (worse angular separation) to gain a larger $||Z||_F$ (higher confidence), which is what we mean by trading discrimination for confidence.
>
> Once again, we sincerely thank the reviewer for the encouraging assessment and valuable suggestions. Please let us know if there are any further questions.

---

> > ### Author Rebuttal · Reviewer_hANK · 2026-04-01
> >
> > I thank the authors for the detailed rebuttal. My concerns are solved, and I would like to maintain my positive score of this work.

---

### Official Review · Reviewer_TccT · 2026-03-14

**Soundness:** 3
**Presentation:** 3
**Significance:** 3
**Originality:** 3
**Overall Recommendation:** 4
**Confidence:** 4

**Summary:**

This paper studies the implicit bias induced by depth in the deep unconstrained feature model (deep UFM), which is trained with multiclass cross-entropy and no explicit regularization. The authors ask whether depth alone, through optimization dynamics and architecture, can disfavor deep neural collapse (DNC) and instead promote alternative low-rank structures. The paper argues that the answer is yes.

The paper makes three main claims.
* First, asymptotically, depth changes the implicit max-margin landscape: unlike the shallow case, DNC is no longer the unique optimal structure, and low-rank alternatives can be locally or globally preferred. The authors attribute this to a depth-induced low-rank bias: low-rank matrices propagate norm more efficiently through repeated matrix multiplication, allowing larger logits for fixed parameter norm.
* Second, under Hadamard-style spectral initialization, the paper derives reduced singular-value dynamics and shows that for deep models the DNC direction becomes unstable near the origin, while low-rank directions can become stable. This is interpreted through a “rich-get-richer” mechanism among singular values.
* Third, under standard random Gaussian initialization, the paper argues that increasing width biases the initial logit velocity toward the simplex-ETF direction associated with DNC, providing a possible explanation for why DNC is still often seen in practice despite the low-rank bias induced by depth.

The authors focus on a central concept: how depth alone reshapes implicit bias in classification models trained without explicit regularization. Overall, the authors outline the concept through a combination of asymptotic analysis, finite-time dynamical analysis, and supporting experiments on deep UFMs and ResNet-style models. The problem is interesting, timely, and relevant to ongoing work on neural collapse, implicit bias, and geometry of deep classifiers. The paper is also well connected to prior work on UFMs, DNC, softmax codes, and gradient-flow implicit bias

**Compliance With Llm Reviewing Policy:**

Affirmed.

**Final Justification:**

The original subdifferential-based argument in Appendix D.1 has been replaced with a different approach based on a smooth/nonsmooth decomposition and a perturbation argument. While still presented at a sketch level, the revised reasoning appears coherent and resolves the key issue I had with the original proof, especially through the use of the $t^p$ vs $t^2$ scaling argument for $p<1$.

The argument in Appendix D.2 has also been significantly improved and is now much clearer and more structured than before.

Overall, my main blocking concern was the correctness of the theoretical arguments. Based on the revised sketches, I no longer see a clear flaw in the reasoning. Although a fully detailed proof would still be desirable, I believe the current level of justification is sufficient for the purposes of the paper, and I am therefore comfortable updating my assessment accordingly.

**Key Questions For Authors:**

1. In Appendix D.1, can the authors provide a fully rigorous nonsmooth-analysis justification for the local optimality of the DNC solution for $f(Z)=(L+1)|Z|\_{S_p}^p$, $p=\frac{2}{L+1}<1$ at the rank-deficient point $Z\_\star=S\otimes 1_n^\top$? In particular, please specify the exact subdifferential notion being used, justify why the claimed matrix belongs to it, and justify the directional-derivative inequality used afterward. A convincing answer here would significantly improve my soundness assessment.

2. In Appendix D.2, can the authors formalize the optimization argument that the optimal 2D configuration must cluster around $0$ and $\pi$ with adjacent gaps equal to $\delta$, rather than presenting it only as geometric intuition? If this step can be made rigorous, it would strengthen confidence in Theorem 3.2.

3. In Theorem 4.5 / Appendix D.6, can the authors clarify the scaling of $\dot Z(0)$ with respect to the initialization scale $\epsilon$? My reading of Eq. (17) suggests each summand scales like $\epsilon^{2L}$, whereas the theorem concludes $\epsilon^{2L-2}$. If I am missing a factor, please make that explicit.

4. The manuscript reports that a lower effective rank of the mean logit matrix correlates with worse generalization in the ResNet-head experiments. Can the authors discuss more explicitly how this compares to prior low-rank simplicity-bias work, especially papers arguing that low-rank bias can help when aligned with the task structure? Clarifying the distinction between the different “rank objects” being measured would improve the significance of the empirical discussion.

**Limitations:**

I encourage the authors to discuss more on the following:
* the limited scope of the model assumptions (deep UFM, orthogonal-input equivalence, Hadamard initialization)
* the gap between the rigorous theory and the ResNet/ReLU experiments
* the fact that some results are asymptotic in depth, width, or initialization scale, and
* the possibility that low-rank bias may help or hurt, depending on the task structure, rather than being uniformly undesirable.

**Strengths And Weaknesses:**

This paper addresses an interesting and relevant question: whether depth alone, without explicit regularization, induces a bias away from neural collapse in deep UFMs trained with cross-entropy. I found the main question clearly stated, the overall motivation strong, and the empirical illustrations useful throughout the paper. In particular, I appreciated that the paper tries to connect three levels of analysis: asymptotic max-margin geometry, finite-time gradient-flow dynamics, and standard random initialization effects. The experiments are also well integrated with the narrative, especially the figures illustrating the decrease of effective rank with depth and the increase of effective rank with width.

On the positive side, the paper is generally well written. Apart from some notation issues, undefined symbols, and several local mistakes in the appendix, the narrative is reasonably easy to follow. The introduction and contributions sections do a good job of positioning the work relative to prior literature on UFM, DNC, and implicit bias. The authors also do a good job of repeatedly returning to the same conceptual message: depth can create a low-rank preference even without explicit regularization.

However, I also found several technical and attribution issues that materially affect my confidence in the soundness of some of the main claims. Some appendices are correct or close to correct, but several core proofs contain either local mathematical mistakes, missing rigor, or arguments that seem to rely on unjustified nonsmooth-analysis steps. In fact, I have substantial concerns about the technical soundness and rigor of several proofs, especially in Appendices D.1, D.2, D.4, D.5, and D.6.

## 1. The novelty claim around low-rank bias should be positioned more carefully

The paper repeatedly emphasizes statements like “we show that depth induces an implicit low-rank bias.” But the low-rank inductive bias arising from depth in deep linear and nonlinear networks is already well known. The paper itself cites related work in this direction, and there is existing literature showing that increasing depth can favor lower-rank structure through optimization and parametrization (see, for example, [1]).

So, in my view, the genuinely new part is not the broad statement that depth induces low-rank bias. Rather, the new part is the specific analysis in the deep-UFM + cross-entropy + no-explicit-regularization setting, together with the connection to DNC, softmax codes, and the width-vs-depth picture developed here. I would encourage the authors to state that distinction more carefully. Otherwise, the novelty can sound broader than what is actually established.

## 2. Section 5 should be positioned more carefully relative to prior empirical work

Section 5 is broadly consistent with prior empirical work on low-rank simplicity bias in deep networks, particularly papers such as Huh et al. (2023) [1], in that both report that increasing depth induces lower-rank structure at initialization and/or convergence.

However, the discussion of generalization should be positioned more carefully. Prior work often argues that low-rank bias can improve generalization when it matches the intrinsic task structure, while also noting that this same bias can become harmful when the task itself is high-rank or when depth is excessive. By contrast, the present paper reports that a lower effective rank of the mean logit matrix tends to correlate with worse generalization in its ResNet-head experiments.

This is not necessarily a contradiction. The two works are not measuring the same object:
* Prior work often studies the rank of embeddings, Gram matrices, or task-aligned representations,
* whereas the present paper studies the effective rank of the mean logit matrix.

Those are different quantities and can behave differently. I therefore do not think the paper is empirically incompatible with that prior literature, but I do think it should discuss this distinction explicitly and position the result more carefully.

## 3. Attribution around Definition 2.1 should be clarified

Before giving Definition 2.1, the paper states that it uses a “well-established equivalent characterization” attributed to Zhu et al. (2021) and Garrod & Keating (2024a). I think this attribution is not precise enough.

My understanding is:
* I could not find this exact equivalent characterization in Zhu et al. (2021), which seems to concern shallow NC/UFM rather than deep DNC in the form used here.
* In Garrod & Keating (2024a), the relevant statement appears to be obtained by combining a definition and a proposition, rather than being literally the same object as the present Definition 2.1.

More importantly, as written, Definition 2.1 is not literally the same as Garrod & Keating’s stronger canonical form. The latter imposes an aligned SVD-type structure on the unnormalized layers, whereas Definition 2.1 only constrains the normalized matrices subject to a balancedness condition. The implication from the stronger Garrod-style form to Definition 2.1 seems fine, but the converse only holds modulo layerwise positive rescaling. Appendix D.1 then appears to use the stronger canonical form, not merely Definition 2.1 as stated. So the paper should clarify exactly what is equivalent to what.

## 4. Some references to equations/definitions are unnecessarily hard to follow

* In Section 3.1 (“Depth Breaks the Benign Landscape”), the authors write that they apply the framework of Lyu & Li (2019) to the deep UFM “(Eq. 7).” But Eq. 7 appears much later in the appendix. Since Eq. 7 is the same model as Eq. 1 in the main text, the paper should simply refer to Eq. 1 there for readability.

* Similarly, Theorem 3.2 refers to logit matrices satisfying “NC1,” but NC1 is not explicitly defined before the theorem. The appendix later introduces DNC1 (feature collapse), which appears to correspond to the intended meaning. This should be defined in the main text before use.

## 5. Equations should be labeled more systematically

A simpler but important presentation issue: many equations are unlabeled, which makes it harder to check the proof and to write a precise review. For a theory-heavy paper, I strongly encourage the authors to label essentially all equations that they later use in derivations.

## 6. Appendix D.1 (Proof of Theorem 3.1)

### (i) Indexing and notation problems

There is a notational inconsistency in the parameterization of the “DNC solution.”
The text writes $W_l = U_l \Sigma'' U_{l-1}^\top$, $l=1,\dots,L-1$. But then for $l=1$, this gives $W_1 = U_1 \Sigma'' U_0^\top$
and $U_0$ is never defined. This matters because the claim is that the displayed factorization realizes Definition 2.1, whose balancedness condition uses $\hat W_0 := \hat H_1$. As written, the indexing does not cleanly verify that condition.

I believe the intended construction is recoverable by shifting the indices, for example $ W_l = U_{l+1}\Sigma'' U_l^\top$; $l=1,\dots,L-1$; together with $H_1 = U_1\Sigma' V^\top$ and $W_L = Q \Sigma U_L^\top.$

With that correction, the telescoping product and the balancedness argument appear to go through. But the appendix should be fixed, because the current statement is not rigorous as written.

This same issue seems to extend to the cross-polytope construction and the other explicit constructions in that proof.

### (ii) Local typographical mistakes inside the proof

I also noticed several local mistakes in D.1:

* Line 950: The paper writes $(z\_{ic })\_{c} - (z\_{ic })\_{c } = \cdots$ but this cannot be right. I believe the intended expression is $(z\_{ic })\_{c} - (z\_{ic })\_{c' } = \cdots$

*  Line 950: The text says “if $i$ and $j$ are paired.” I think what is meant is not that sample indices are paired, but that the true class $c$ and the competing class $c'$ are paired.

*  Line 1010: After “Now define $g_{icc'} = \dots$”, the condition appears as something like $c\neq c$, which is clearly a typo. It should be the condition $c\neq c'$.

* Line 1035: After the sentence “Also note that the matrix ...”, one displayed equation appears to be missing the gradient symbol. Without it, the equation becomes dimensionally inconsistent, since a scalar quantity is being equated to a matrix expression.

These are individually small issues, but they occur in a core proof.

### (iii) The subdifferential argument for local optimality is not fully justified

This is the most important technical issue in the paper for me.
After reduction to the balanced formulation, the proof studies $f(Z) = (L+1)|Z|_{S_p}^p$, $p=\frac{2}{L+1}$.

For $L\ge 2$, we have $p=\frac{2}{L+1}<1$. The proof then attempts to show that the DNC solution $Z_\star = S\otimes 1_n^\top$ satisfies a KKT-type stationarity condition by identifying $G = 2 n^{-L/(L+1)} Z_\star$ as an element of $\partial f(Z_\star)$.
The issue is that the justification appears to rely on the gradient formula for $f$ at matrices with positive singular values, and then implicitly passes to the limit as $Z\to Z_\star$. But $Z_\star$ is rank-deficient: its rank is $K-1$, so it has at least one singular value equal to zero.

To see why this is a real issue, consider a sequence of nearby matrices $Z_\varepsilon \to Z_\star$ such that the zero singular value is perturbed to a small positive value $\varepsilon>0$. For example, if the singular values of $Z_\star$ are $\sigma_1=\cdots=\sigma_{K-1}=\sqrt{n}, \quad \sigma_K=0$; consider matrices $Z_\varepsilon$ with singular values
$\sigma_1=\cdots=\sigma_{K-1}=\sqrt{n}$, $\sigma_K=\varepsilon$, where $\varepsilon\to 0$.

For matrices with strictly positive singular values, the gradient formula used in the proof takes the form $\nabla f(Z) = 2U \mathrm{diag}(\sigma_i^{p-1}) V^\top$. Now look at the coefficient corresponding to the smallest singular value, $\sigma_K^{p-1} = \varepsilon^{p-1}$.
Since $p<1$, we have $p-1<0$, therefore $\varepsilon^{p-1} = \frac{1}{\varepsilon^{1-p}} \longrightarrow \infty$ as $\varepsilon\to 0$.

So along such a sequence $Z_\varepsilon\to Z_\star$, the gradients $\nabla f(Z_\varepsilon)$ contain entries that diverge to infinity and hence do not converge to a finite matrix. But the appendix seems to use a limiting-gradient notion of subdifferential of the form
$M\in \partial f(Z_\star) \iff \exists Z_k\to Z_\star \text{ such that } \nabla f(Z_k)\to M$.

Under that definition, the proof must explicitly construct a special approximating sequence approaching $Z_\star$ for which the gradients converge to the claimed matrix $2 n^{-L/(L+1)} Z_\star$. No such construction is provided.

So the current proof does not, in my view, justify the claim $2 n^{-L/(L+1)} Z_\star \in \partial f(Z_\star)$. This is not a cosmetic issue: it is the key nonsmooth step in the local-optimality argument.

### (iv) Later use of a “standard subgradient property” is not automatic here

A related issue appears later when the proof invokes $f'(Z_\star;\Delta)\ge \langle G,\Delta\rangle$ as a “standard property of sub-gradients.” This is not automatic in the present setting.

Why? Because the function $f(Z)=(L+1)|Z|\_{S\_p}^p$ with $p<1$ is nonconvex. In fact, it is a Schatten quasi-norm power, not a convex norm. Also, near rank-deficient matrices, it is not locally Lipschitz in the same benign way one usually exploits in standard Clarke/Rockafellar convex-subgradient arguments.
So the inequality $f'(Z_\star;\Delta)\ge \langle G,\Delta\rangle$ cannot simply be cited as a generic convex-subgradient fact.

The authors need either:
* a precise theorem from nonsmooth analysis adapted to this nonconvex quasi-norm setting, together with verification of its assumptions,
* or a direct proof of the directional inequality in their specific case.

As written, neither is provided. I would therefore not accept this step without a precise justification.

## 7. Appendix D.2 (Proof of Theorem 3.2)

(i) At line 1102, the authors write $Z'\_{c'c}=x\_{c'}^\top x\_c$ inside the second equality, but the intended expression should clearly be $Z'\_{c'c}=\beta x\_{c'}^\top x\_c$

(ii) At line 1155, the paper says “Since the objective is monotonic decreasing in $r$ ...” I believe this is reversed. The objective appears to be increasing in $r$, so the minimization argument should instead conclude that one chooses the smallest feasible $r$.

(iii) The paper says, roughly, that the optimal configuration must have gaps equal to $\delta$ in the direction of whichever is closer between $0$ and $\pi$. I understand the intended reasoning to be:
* if a point is not blocked by another point at distance exactly $\delta$ in the direction of the nearer maximizer,
* then one could slide it slightly toward that preferred direction,
* improve the objective,
* and keep feasibility.

This is plausible and matches the usual packing/local-improvement intuition.  But the current text does not fully formalize that step. It is not enough to say “therefore at optimum every point must be jammed”; one must show that the proposed perturbation indeed preserves all constraints and improves the objective. This issue affects not just one sentence, but most of that paragraph. The result may still be true, but the proof should be written more rigorously.

## 8. Appendix D.3 (Proof of Theorem 4.2)

* In D.3, the matrices $P$ and $Y$ are used before being clearly defined in the proof. From context, I infer that $P_{ij}=\frac{e^{Z_{ij}}}{\sum_k e^{Z_{kj}}}$ and $Y = I_K \otimes 1_n^\top$. But these should be stated explicitly.

* Also, at line 1306, the authors write $H_1 = R_1^\top D_0 V^\top$ but I believe this should be $ H_1 = R_1 D_0 V^\top$

## 9. Appendix D.4 (Proof of Theorem 4.3)

(i) At line 1420, the proof writes $D = 1 + K e^{-K\mu} + \mathcal{O}(|\epsilon|_1^2)$. But from the preceding sums, the relevant index runs from $1$ to $K-1$, so I believe the correct expansion is $D = 1 + (K-1)e^{-K\mu} + \mathcal{O}(|\epsilon|_1^2)$. This mistake propagates into the denominator of $C(\mu)$. That said, I do not think this changes the final stability threshold, because this denominator only enters through a common positive prefactor $C(\mu)$, while the sign argument later depends on $B(\mu)-1$, not on this prefactor. So I think the conclusion may survive, but the displayed algebra should be corrected.

(ii) At line 1494, the sentence $a_j = \epsilon_j$ should, I believe, be $a_j = \mu_j \epsilon_j$

(iii) At line 1552, the proof contains “Theorem ??”: “To note that this monotonic approach must guarantee we tend to the limit point, rather than stalling at some other structure, recall in Theorem ?? ...” This should obviously be fixed.

(iv) Lemma D.8 seems to contain a sign error in the decomposition of $e^{-\Psi a}$. The current decomposition effectively behaves like $e^{-K\mu/2}1_{K-1} + \bigl(e^{-K\mu}+e^{-K\mu/2}\bigr)e_1$, which would give first coordinate $e^{-K\mu}+2e^{-K\mu/2}$.
But that is not the target coordinate. The correct decomposition should instead be $e^{-K\mu/2}1_{K-1} + \bigl(e^{-K\mu}-e^{-K\mu/2}\bigr)e_1$.
Indeed, then the first coordinate becomes $e^{-K\mu/2} + \bigl(e^{-K\mu}-e^{-K\mu/2}\bigr) = e^{-K\mu}$ which is the correct value.

With this correction, the subsequent Taylor expansion and the final expression of the lemma appear to go through.

## 10. Appendix D.5 (Proof of Theorem 4.4)

Eq. (16) is written as an exact equality, but earlier in the derivation, the proof approximates $da_i/dt$ and discards higher-order terms. Therefore, the resulting displayed equation should strictly include a remainder term. At a minimum, the paper should write it as an asymptotic relation or add an explicit error term inherited from the approximation.

This may not destroy the qualitative conclusion, but the equation should not be presented as exact.

## 11. Appendix D.6 (Proof of Theorem 4.5)

The overall argument of D.6 appears conceptually reasonable to me.

Starting from $Z = W_LW_{L-1}\cdots W_1H_1$, the authors differentiate under gradient flow and obtain $\dot Z = \sum_{l=0}^{L}
A_{l+1}A_{l+1}^\top (Y-P), H_l^\top H_l$, where $A_{l+1}=W_L\cdots W_{l+1}$ and $H_l=W_{l-1}\cdots W_0$.
Then, using Lemma D.9, they argue that the Gram factors become proportional to the identity in the large-width limit, so the direction of $\dot Z(0)$ is proportional to $Y-P(0)$. Since $Z(0)\to 0$ as $\epsilon\to 0$, the softmax becomes uniform and hence $Y-P(0)\to S\otimes 1_n^\top$.

That directional conclusion makes sense. However, I think there is a potential scaling inconsistency in $\epsilon$.
The weights are initialized as $ W_l(0)=\epsilon B_l$. Each weight contributes one factor of $\epsilon$.
Now in the $l$-th summand of Eq. (17), we have the factors $A_{l+1}A_{l+1}^\top$ and $H_l^\top H_l$.

Since $A_{l+1}=W_LW_{L-1}\cdots W_{l+1}$, it contains $L-l$ matrices, so $A_{l+1}A_{l+1}^\top \sim \epsilon^{2(L-l)}$.
Similarly, $H_l=W_{l-1}\cdots W_0$ contains $l$ matrices, so $H_l^\top H_l \sim \epsilon^{2l}$.
Therefore, each summand scales as $\epsilon^{2(L-l)}\epsilon^{2l} = \epsilon^{2L}$ independently of $l$.

Since there are $L+1$ summands in the sum, this suggests $\dot Z(0)\sim (L+1)\epsilon^{2L}(Y-P(0))$. By contrast, the proof concludes $
\dot Z(0)\sim (L+1)\epsilon^{2(L-1)}(Y-P(0))$. So the derivation seems off by two powers of $\epsilon$.

This discrepancy may not affect the directional conclusion of the theorem, which is the main conceptual point, but it does suggest that the scaling derivation should be checked carefully.

## 12. Lemma D.9

In the proof of Lemma D.9, the authors introduce vectors $g_i$ through $g_i=\sqrt{\frac{1}{d}},w_i$ and state that $ g_i\sim\mathcal N(0,I_d)$.
This is incorrect.
If $w_i\sim\mathcal N(0,I_d)$, then multiplying by a scalar rescales the covariance, so $g_i=\sqrt{\frac{1}{d}} w_i
\sim \mathcal N(0,\frac{1}{d}I_d)$, not $\mathcal N(0,I_d)$.

Equivalently, if the authors wanted a standard Gaussian, the correct normalization would be $g_i=\sqrt d,w_i$.
This matters because if $g_i\sim \mathcal N(0,\Sigma)$, then $\mathbb E[g_ig_i^\top]=\Sigma$.

So under the current definition, $\mathbb E[g_ig_i^\top]=\frac{1}{d}I_d$, whereas the proof at that point seems to proceed as if $
\mathbb E[g_ig_i^\top]=I_d$.

Likewise, identities of the form $\mathbb E[g_i^\top A g_i]=\operatorname{tr}(A)$ should instead become $ \mathbb E[g_i^\top A g_i]=\frac{1}{d}\operatorname{tr}(A)$.

The authors later introduce matrices of the form $W=\sqrt{\frac{1}{d}},\widetilde W, \quad \widetilde W_{ij}\sim\mathcal N(0,1)$, which is the standard random-matrix scaling, and the later variance computations appear more consistent with that normalization. So I suspect this is a statement-level error rather than a fatal breakdown of the proof. But it is still mathematically incorrect and should be fixed.

## 13. Supporting lemmas from prior work should be cited more precisely

When a result is imported from another paper, the manuscript should cite the exact theorem, proposition, or lemma being used, not just the entire paper. This is especially important in a proof-heavy submission. It would make verification much easier.

# References

[1] The Low-Rank Simplicity Bias in Deep Networks (https://openreview.net/pdf?id=bCiNWDmlY2)

---

> ### Author Rebuttal · Authors · 2026-03-31
>
> We thank you for the thoughtful review and greatly appreciate the effort devoted to evaluating our paper. Below, we address each point in turn.
>
> **1. Novelty claim around low-rank bias**
>
> We will ensure in the revision that vague statements such as “we show that depth induces a low-rank bias” are qualified to reflect our specific setting. We will also expand the literature review to include references such as [1] and clarify that prior work has identified depth-induced low-rank bias in other settings.
>
> **2. Relation to prior work**
>
> We agree that the discussion of low-rankness and generalization should be clarified, particularly in relation to Huh et al. (2023). We will also cite Jacot (2023), Implicit Bias of Large Depth Networks, which develops related ideas. Our view is that softmax codes correspond to rank-underestimating minima, which leads to worse generalization. At the same time, low-rankness is not always harmful: for example, DNC has lower rank than at initialization and has been linked to improved generalization. We will clarify this in Section 5.
>
> **3. Definition 2.1**
>
> We agree that citing Zhu et al. here reduces clarity, and we will remove the citation.
>
> The reviewer is correct that our Definition 2.1 is not identical to that of Garrod & Keating. Their definition imposes an aligned SVD structure on the unnormalized layers (exact balancedness), which does not arise under gradient flow unless it holds at initialization. Our Definition 2.1 instead operates on the normalized matrices, where balancedness holds asymptotically irrespective of initialization. Since App. D.1 analyzes the max-margin problem characterizing the limiting direction of these normalized parameters, the solutions can be taken to be balanced. This follows from Definition 2.1, not from the Garrod & Keating formulation. We will clarify this distinction in the revision.
>
> **4,5: Equation usage**
>
> We will revise these as suggested.
>
> **6(i), 6(ii), 7 (i), (ii), 8-12: Typographical errors**
>
> Thank you for identifying these issues. We have verified that each is localized and does not affect the validity of the theorems. We will implement all the corrections you suggested. Additional details are below where needed.
>
> * 8: We will define P, Y before line 1294.
> * 9 (iii): this should refer to Lem D.1.
> * 10: We will include the error term in the subsequent lines; it is lower order and does not affect the conclusion.
> * 11: This should be $ϵ^{2L}$. The main text will be changed. The directional conclusion is unaltered.
>
> **6 (iii),(iv),  incorrect argument for local optimality**
>
> Thank you for identifying this issue. You are correct that the original argument is invalid, we will remove it. Below we clarify the impact on our results.
>
> **DNC as a KKT point**: DNC remains a KKT point of the asymptotic max-margin problem. This follows from Lyu & Li (2019), who show that if a gradient-flow trajectory converges in direction, then its limit must be a KKT point. Since a solution initialized at DNC remains DNC under the dynamics, this yields a gradient-flow path whose directional limit is DNC.
>
> **Local minimality**: We believe we have an alternative proof, but due to rebuttal space constraints we will provide it in a follow-up comment.
>
> More importantly, local minimality was used to support that DNC is a plausible limit under random initialization. However, knowing that DNC is a KKT point is a sufficiently strong result for our purposes. Indeed, the analysis preceding Theorem 3.1 relies on Lemma D.1, which only guarantees convergence to a KKT point. This is not a technical weakness of the result: in CE training, there are settings where randomly initialized paths converge to saddle KKT points (Vardi et al., 2022). Stronger guarantees are only known in special cases, such as under Hadamard initialization for L=1 (Garrod et al.). However we discuss this case in detail, showing that DNC is locally attractive above a norm threshold, but so are other structures. Our numerical results in Figure 3 also show that, for the same value of L, training can recover a range of structures, including NC.
>
> **7 (iii): Non-rigorous step**
>
> We agree that this step is not formal. We will provide a formal proof in a follow-up comment.
>
> **13: Lemma citations**
>
> We will revise the manuscript:
>
> * Lem D.1 - in the proof we mention Thm. 3.2 of Ji et al. (2021), Thm. 4.4 of Lyu & Li (2019).
> * Lem D.4: Lem. 3.6 of Horadam (2012).
> * Lem D.5: Exercise 1.12 of O’Donnell (2014); we will also include a short inductive proof.
> * Lem D.6: Cor. 3 of Shang et al. (2020); we will also correct a typo.
>
> **Limitations**
>
> An explicit limitations section would strengthen the manuscript. We will add one addressing the four highlighted areas.
>
> Thank you again for your detailed and constructive feedback, which we will use to improve the clarity and soundness of the paper. We will provide the formal arguments for 6 and 7(iii) in a follow-up comment, and we welcome any further questions.

---

> > ### Author Rebuttal · Reviewer_TccT · 2026-04-03
> >
> > I thank the authors for the detailed and constructive rebuttal. I appreciate that many of my comments were taken seriously, and that the authors acknowledged several issues and proposed concrete revisions. In particular, I find the clarifications regarding the novelty claim, the positioning with respect to prior work, and the attribution around Definition 2.1 to be satisfactorily addressed. I also appreciate the confirmation and correction of the various typographical and local issues in the appendices, as well as the clarification of the scaling in Appendix D.6.
> >
> > However, my main remaining concerns relate to the **core theoretical arguments**, which were the primary reason for my initial score.
> >
> > In particular:
> > * The authors acknowledge that the original subdifferential-based argument for local optimality in Appendix D.1 was incorrect and will be removed. While I appreciate this transparency, the replacement argument for local minimality has not yet been provided in the rebuttal (it is deferred to a follow-up comment). As a result, I am currently unable to assess whether the revised argument fully resolves the issue.
> > * Similarly, the non-rigorous step in Appendix D.2 is acknowledged and will be formalized later, but the formal argument is not yet available for evaluation.
> >
> > These points concern central parts of the theoretical contribution, not peripheral details. While I understand that providing full revised proofs may be difficult within the rebuttal constraints, I am not in a position to verify the correctness of the new arguments at this stage.
> >
> > More broadly, my current hesitation is not about the overall direction of the paper, which I find interesting and promising, but about the rigor and completeness of the theoretical results in their revised form. Since these theoretical components are central to the paper’s claims, I cannot raise my score in good faith without being able to check the corrected arguments.
> >
> > ### Follow-up questions
> >
> > 1. For Appendix D.1: could the authors provide (even at a high level) the key idea of the alternative argument for local minimality, and clarify which nonsmooth framework or tools are being used?
> >
> > 2. For Appendix D.2: can the authors outline the structure of the formal argument that replaces the current geometric intuition (e.g., what is the precise perturbation argument or optimization framework used)?
> >
> > If these revised arguments are sound and can be clearly verified, they would significantly improve my confidence in the paper.

---

> > > ### Author Response · Authors · 2026-04-04
> > >
> > > We address 6,7 with streamlined proofs. We note that local minimality is stronger than we need for the paper as we can show KKT as addressed in rebuttal, if the below proof is too terse to be confirmed we are happy to reduce the claim to KKT to ensure soundness of the paper. Thank you for your substantial efforts.
> > >
> > > **7**
> > >
> > > We calculate the val of the objective in line 1196. Let $C_0=[-π/2,π/2]$, $C_{π}=[π/2,3π/2]$. Consider a config of $θ_1,...,θ_K$ with $r$ points in $C_0$, denoted $x_1<...<x_r$, and $s=K-r$ in $C_π$, denoted $y_1<...<y_s$.
> > >
> > > Suppose $∃j$ s.t. $x_{j+1}-x_j>δ$. Then $∃ϵ$ s.t. one can shift the furthest of $x_j,x_{j+1}$ from 0 towards 0 by $ϵ$, improving the objective since $\cos(2θ)$ increases as you approach 0 in $C_0$. Same reasoning holds for the $y$'s. Hence any optimal config has
> > >
> > > $x_j=α+(j-\frac{r+1}{2})δ$
> > >
> > > $y_j=π+β+(j-\frac{s+1}{2})δ$
> > >
> > > for some $α,β$. Feasibility is ensured within $C_0,C_π$. For feasibility between, by considering edgepoints in $C_0,C_π$ and demand both gaps be $≥δ$ we have condition:
> > >
> > > $|β-α|≤π-\frac{K}{2}δ$
> > >
> > > Return to our objective for configs of this form: writing in complex form and using geometric sum formula, objective val is
> > >
> > > $\textrm{Re}(e^{2iα}\frac{\sin(r δ)}{\sin(δ)}+e^{2iβ}\frac{\sin(sδ)}{\sin(δ)})$
> > >
> > > This is upper bounded by the modulus, where global rotation allows attainment of the bound. Hence, consider
> > >
> > > $=|\frac{\sin(rδ)}{\sin(δ)}+e^{2i(β-α)}\frac{\sin(sδ)}{\sin(δ)}|$
> > >
> > > Define $\nu=2(β-α)$. If $\sin(rδ)≥0$ and $\sin(sδ)≥0$ (case 1), then objective is max at $\nu=0$ and has val
> > >
> > > $\frac{\sin(rδ)+\sin(sδ)}{\sin(δ)}$
> > >
> > > If $\sin(rδ)$ or $\sin(sδ)$ is -ve (case 2), then objective is:
> > >
> > > $\frac{1}{\sin(δ)}(\sin^2(rδ)+\sin^2(sδ)+2 \sin(rδ)\sin(sδ)\cos(\nu))^{\frac{1}{2}}$
> > >
> > > Must have $δ>\frac{π}{K}$ for this to happen since $r,s≤ K$, then from feasibility we have $|\nu|≤π$. Hence, objective is max when $|\nu|$ at its upper bound $|\nu|=2π-Kδ$ (from feasibility). Then use trig identities to simplify objective val to
> > >
> > > $\frac{|\sin(Kδ)|}{\sin(δ)}$
> > >
> > > We have calculated the best objective val for fixed $r$ conditional on the sign of the sin terms, remains to optimize $r$. First optimize among case 1:
> > >
> > > $\sin(rδ)+\sin(sδ)=2\sin(\frac{K}{2}δ)\cos(\frac{2r-K}{2}δ)$
> > >
> > > we have $0<\frac{K}{2}δ≤π$, so sin term is non -ve, and $-π≤\frac{2r-K}{2}δ≤π$, so cos term is max when $2r-K$ is closest to 0, i.e. $r=⌊\frac{K}{2}⌋$.
> > >
> > > This solution $(r=⌊\frac{K}{2}⌋$, $s=⌈\frac{K}{2}⌉$) maintains non-negativity of both $\sin(rδ),\sin (sδ)$ provided $⌈\frac{K}{2}⌉δ≤π$.
> > >
> > > Assuming thats the case, we show Case 2 cannot beat Case 1:
> > >
> > > $|\sin(Kδ)|=2\sin(\frac{K}{2}δ)|\cos(\frac{Kδ}{2})|$
> > >
> > > even $K$:
> > >
> > > $2\sin(\frac{K}{2}δ)|\cos(\frac{Kδ}{2})|≤ 2\sin(\frac{K}{2}δ)$
> > >
> > > odd $K$:
> > >
> > > $2\sin(\frac{K}{2}δ)|\cos(\frac{Kδ}{2})|≤2\sin(\frac{K}{2}δ)\cos(\frac{1}{2}δ)=\sin(⌊\frac{K}{2}⌋δ)+\sin(⌈\frac{K}{2}⌉δ)$
> > >
> > > where used $0<δ/2<Kδ/2≤⌈\frac{K}{2}⌉δ≤π$.
> > >
> > > In all cases when $r=⌊\frac{K}{2}⌋$ leads to +ve $\sin(rδ)$, $\sin(sδ)$, we have
> > >
> > > $\max_{θ_1,...,θ_K,\textrm{s.t. gaps }δ}\sum_u \cos(2θ_u)=\frac{1}{\sin(δ)}(\sin(⌊\frac{K}{2}⌋δ)+\sin(⌈\frac{K}{2}⌉δ))$
> > >
> > > And our proof goes through as described in the previous text to minimize over $δ$.
> > >
> > > Remains to cover cases where one of the sine's is -ve. This happens when $⌈\frac{K}{2}⌉δ$ is $≥π$. This occurs when $K$ odd, and $\frac{2π}{K+1}<δ≤\frac{2π}{K}$.
> > >
> > > Remains to replicate the next steps of the previous text for this alternative minimizer, i.e. show for odd $K>2$:
> > >
> > > $\min_{x∈[\frac{2π}{K+1},\frac{2π}{K}]}\frac{K^2-u^2(x)}{(1-\cos(x))^2}$
> > >
> > > takes its min val at $x=\frac{2π}{K}$, where $u(x)=\sin(Kx)/\sin(x)$.
> > >
> > > To do this we show the derivative is negative on domain, through standard trig identities and inequalities.
> > >
> > > **6**
> > >
> > > To prove the local optimality of DNC matrix $Z_*$, assume for contradiction that there exists feasible sequence $Z_k→Z_*$ with $f(Z_k)≤f(Z_*)$. Let
> > >
> > > $t_k:=\|Z_k-Z_*\|_F,$
> > >
> > > $Δ_k:=\frac{Z_k-Z_*}{t_k},$
> > >
> > > so $t_k→0$, $\|Δ_k\|_F=1$, and after subselection $Δ_k→\bar{Δ}$. Write
> > >
> > > $f=f_1+σ_K^p,$
> > >
> > > where $f_1$ is the sum of the first K-1 singular-value terms. Since K-1 nonzero singular values of $Z_*$ are separated from zero, $f_1$ is $C^2$ near $Z_*$, so
> > >
> > > $f(Z_k)-f(Z_*)≥f_1(Z_k)-f_1(Z_*)≥t_k⟨G,Δ_k⟩-O(t_k^2),\quad G=∇f_1(Z_*)$
> > >
> > > Feasibility and structure of $Z_*$ imply $⟨G,Δ_k⟩≥0$. If $⟨G,\bar{Δ}⟩>0$, the linear term is eventually +ve, contradiction. If $⟨G,\bar{Δ}⟩=0$, then the active constraints force $\bar{Δ}=1_Kv^\top$. Now analyze $M_t:=Z_*+t\bar{Δ}$. If $v∈\mathrm{row}(Z_*)$, then rank stays $K-1$, one existing nonzero singular value increases by order $t^2$, and $C^2$ regularity of $f_1$ transfers this positive quadratic increase from the ray $M_t$ to the nearby directions $Δ_k$, again giving $f(Z_k)>f(Z_*)$. If $v$ has a nonzero component in $\ker(Z_*)$, then we show $σ_K(M_t)≥ct$ for constant $c$ by computing eigenvalues of $M_t^TM_t$; by Weyl, the same holds for $Z_k$, so
> > >
> > > $f(Z_k)-f(Z_*)≥-O(t_k^2)+c \ t_k^p>0$
> > >
> > > for large $k$, since $p<1$: contradiction again.

---

### Official Review · Reviewer_ZwZK · 2026-03-23

**Soundness:** 4
**Presentation:** 4
**Significance:** 3
**Originality:** 3
**Overall Recommendation:** 4
**Confidence:** 3

**Summary:**

The authors focus on how network depth alters the implicit bias of unregularized multiclass cross-entropy training, namely whether training converges to (deep) neural collapse or to other geometries, in the deep unconstrained feature model (UFM). Overall, the authors argue that depth induces an implicit low-rank bias: in the asymptotic max-margin/norm-minimization view of gradient flow, deep neural collapse remains a local optimum but becomes globally suboptimal for sufficiently deep/multi-class settings because lower-rank structures propagate norm more efficiently through matrix products, and in the infinite-depth limit the optimal solutions correspond to a previously defined configuration dubbed softmax codes; they complement this with a dynamical analysis (under a Hadamard/spectral initialization) showing neural collapse can be unstable for L>1, while large width (and random initialization) bias early dynamics back toward neural-collapse-like directions, and they support these predictions with experiments on standard networks.

**Compliance With Llm Reviewing Policy:**

Affirmed.

**Final Justification:**

I have decided to maintain my original score, as my overall opinion of the paper was not substantially changed.

**Key Questions For Authors:**

- Is the mechanism for the emergence of low rank solutions posited here the same as "greedy low rank learning" which people have observed in regression problems, e.g in deep linear networks?
- Has your observation that lower rank coincides with worse generalization been supported in other works? I find it surprising and counterintuitive.

**Limitations:**

Yes.

**Strengths And Weaknesses:**

Strengths. Though I have not reviewed the proofs, I believe the paper is provides a valuable technical contribution, combining a landscape analysis with a finite-time dynamical analysis to study how depth affects implicit bias in the unregularized deep UFM. It is also well written and well organized, with a clear narrative and strong positioning relative to prior NC/DNC and implicit-bias work, and it addresses an important concept which likely to be useful as a theoretical lens for follow-on research.

Weaknesses. The main weakness is the inherent limitation of the UFM itself: treating features as free per-sample variables abstracts away shared feature learning and does not define a complete test-time mapping, so the extent to which the mechanisms directly explain real deep networks remains unclear. I will note however that simplications such as the UFM are ubiquitous in the DL theory literature, and should not disqualify the paper.

---

> ### Author Rebuttal · Authors · 2026-03-31
>
> We thank the reviewer for their thoughtful feedback and for the time and care devoted to reviewing our submission. We address each highlighted weakness and question below.
>
> **The main weakness is the inherent limitation of the UFM itself … the extent to which the mechanisms directly explain real deep networks remains unclear.**
>
> We agree that the UFM is an idealized model whose limitations must be kept in mind. As the reviewer notes, UFM-like abstractions are well established in deep learning theory; we refer readers to Appendix A of Garrod & Keating (2024) for an extensive catalog of recent UFM-based work and empirical justification. Our contribution is to leverage this framework to identify and characterize mechanisms that would be difficult to isolate in more complex settings. We also provide experiments showing that the phenomena extend beyond the UFM. That said, studying the extent to which our theoretical characterizations and tools carry over to more realistic models remains an important direction for future work, and we will add a dedicated limitations discussion in the revised paper.
>
> **Is the mechanism … posited here the same as "greedy low rank learning"?**
>
> We assume the reviewer is referring to Li et al. (2020), which we cite in the paper. This is a very interesting question. We believe the two phenomena are related, but not identical. In the L=1 UFM, as described by Garrod et al. (2025), the dynamics resemble the “greedy low-rank learning” picture of Li et al., in which the network moves through saddles of increasing rank before reaching a NC solution. In our setting, however, this process terminates early because of low-rank bias, causing the network to settle at a rank-constrained optimum rather than continuing toward the NC solution. We do suspect that the dynamics prior to convergence to a low-rank solution may be similar. The reason for this difference is that, in linear network regression problems, achieving small loss requires the network to match the rank of the target matrix Y, so low-rank bias cannot manifest in the same way.
>
> **Has your observation that lower rank coincides with worse generalization been supported in other works?**
>
> Related ideas appear in Arthur Jacot’s work ‘Implicit Bias of Large Depth Networks: a Notion of Rank for Nonlinear Functions’, which we will cite in the revision. In our setting, lowering rank is not always harmful: NC itself is a rank (K-1) solution, which is often much lower than the rank at initialization, and is associated with strong generalization (Papyan et al., 2020). The key point is instead that some minima are too rank-constrained and thereby harm generalization, and depth can push optimization towards these minima.
>
> Thank you again for your thoughtful review. We would be happy to address any further questions.

---

> > ### Author Rebuttal · Reviewer_ZwZK · 2026-04-03
> >
> > I would like to thank the authors for their well written and substantive response. I am still somewhat concerned with the use of the UFM model -though it is certainly well established, I am skeptical of it as a proxy for real architectures. Having said that, this is cleary a solid paper. I am maintaining my score.

---

### Decision · Program_Chairs · 2026-04-30

**Decision:**

Accept (regular)

**Comment:**

This paper studies how network depth affects the implicit bias of unregularized cross-entropy training in the deep unconstrained feature model (UFM). It shows that increasing depth induces a low-rank bias that departs from Neural Collapse (NC), with deep NC becoming suboptimal in certain regimes, and characterizes the limiting behavior as converging to softmax code structures. The paper combines loss landscape analysis with gradient flow dynamics and supports its theoretical findings with empirical results.

Reviewers raised several concerns, which have been largely addressed during the rebuttal. In particular, while the UFM setting relies on simplifying assumptions and may not fully capture real network behavior, reviewers agreed that this is a standard abstraction in the literature. After rebuttal, all reviewers recommend acceptance. Therefore, I concur with this assessment and encourage the authors to revise the manuscript according to the discussion in the rebuttal for the final version.